# Coordinated Disentanglement with Iterative Mode Discovery Under Hidden Correlations

**Rong Hu** [1] [2]  **Ling Chen** [1] [2]

## Abstract

Disentangled representation learning is a powerful paradigm for robust attribute prediction. While recent methods address attribute correlations, hidden correlations remain underexplored, where data under the value of a certain attribute exhibit underlying modes correlated with other attributes. To preserve mode information and achieve disentanglement, we jointly discover modes and enforce mode-based conditional independence. Yet, the interdependency between these two modules may lead to error amplification under naive iterations. We propose Coordinated Disentanglement with Iterative mode Discovery (CoDID), an end-to-end framework featuring a dynamic architecture that adapts to evolving number of modes, and a coordination mechanism that mitigates error amplification via meta-optimization. Empirical results demonstrate the state-of-the-art performance on diverse tasks.

## 1. Introduction

Supervised disentangled representation learning (DRL) aims to learn the representation of each data attribute under label supervision, such that a representation only changes when its corresponding attribute changes, while being invariant to changes in other attributes (Bengio et al., 2013). While DRL methods usually assume independence between attributes, this assumption is often violated in real-world data, e.g., human activity and user identity (ID) attributes may be correlated due to personal habits (Figure 1(a)). Under attribute correlations, enforcing representation independence causes the loss of attribute information, whereas enforcing conditional representation independence via attribute-based

[1]State Key Laboratory of Blockchain and Data Security, Zhejiang University, Hangzhou, China [2]College of Computer Science and Technology, Zhejiang University, Hangzhou, China. Correspondence to: Ling Chen <lingchen@cs.zju.edu.cn>.

*Proceedings of the 43$^{rd}$ International Conference on Machine Learning*, Seoul, South Korea. PMLR 306, 2026. Copyright 2026 by the author(s).

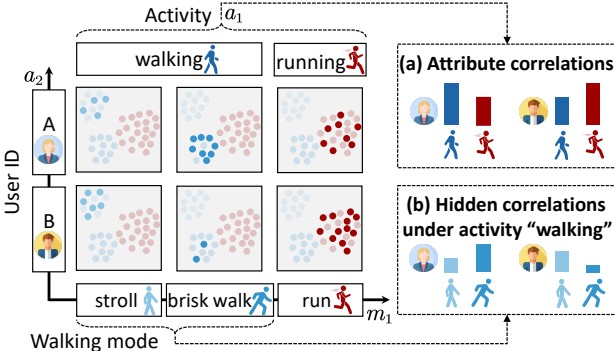

*Figure 1.* Human activity example. The distributions of (a) **"walking" / "running"** and (b) **"stroll" / "brisk walk"** differ between users, exhibiting attribute and hidden correlations, respectively. Since the "brisk walk" mode resembles "running", a model trained under such hidden correlations may over-encode user ID to better distinguish them, i.e., associating user B with "running" on these samples, as user B hardly performs "brisk walk". Under correlation shift, e.g., when user B frequently performs "brisk walk" at test time, this may lead to misclassification.

conditional mutual information minimization (A-CMI) can yield disentanglement (Funke et al., 2022).

Existing methods overlook the underlying variations of attributes: under a value of a certain attribute, variations related to this attribute may lead to a complex *multi-modal* data distribution, characterized by multiple high-density regions, with each referred to as a *mode*; these modes may be correlated with other attributes, referred to as *hidden correlations*. For example, variations of pace, stride, and posture may lead to different modes under "walking" activity, the casual "stroll" and energetic "brisk walk"; these modes may be correlated with user ID attribute due to subtle differences in personal habits (Figure 1(b)). In this case, minimizing attribute-based CMI may cause the loss of mode information, which is important for attribute prediction (Nie et al., 2020; Sugiyama, 2021; Li et al., 2017). The modes under different attribute values may form adjacent or interleaved cluster structures, where preserving such structures benefits attribute prediction, e.g., the high-intensity "brisk walk" mode resembles "running" activity, and encoding this easily confused mode helps distinguish walking from running.

To preserve mode information under hidden correlations, disentanglement can be achieved by minimizing mode-

based CMI. This requires mode labels for conditioning, which are often unavailable in real practice. A straightforward approach is to perform clustering on pre-trained representations to discover underlying modes, and then use cluster labels as proxies for mode labels in DRL. Since the pre-trained representations of the certain attribute are not disentangled, these representations might contain incomplete information about this attribute or redundant information about other attributes. Consequently, the clusters on these representations might misrepresent the true underlying modes, leading to clustering errors and compromising DRL.

Iteratively disentangling representations and discovering modes appears to be a promising solution, which can establish interactions between these two modules: disentangling representations could facilitate mode discovery, and the discovered modes could provide better guidance for minimizing mode-based CMI during disentanglement. However, the two modules might also compromise each other, potentially leading to catastrophic error amplification (Song et al., 2022): disentanglement based on inaccurate cluster labels might degrade the quality of representations, which in turn makes mode discovery more error-prone. In addition, prespecifying the number of clusters under each attribute value demands heavy computational overhead (Wang et al., 2018). While nonparametric clustering can reduce this overhead, abrupt changes in the number of clusters may cause training instability and increase the risk of error amplification.

To address the above challenges, we propose Coordinated Disentanglement with Iterative mode Discovery (CoDID) for supervised disentanglement under hidden correlations. CoDID pioneers *iterative* mode discovery and disentanglement under hidden correlations, featuring a theoretically grounded coordination mechanism that prevents error amplification. Our main contributions are:

- **Theoretical Insights**: We identify that error amplification primarily stems from *within-cluster correlation*, where a cluster contains data from multiple true modes that are correlated with other attributes. This correlation induces mode information loss during CMI minimization, hindering representation disentanglement; the resulting representations exhibit ambiguous mode boundaries, causing clustering errors and creating a detrimental cycle of error amplification.

- **Iterative Framework**: We propose an iterative framework of disentanglement and mode discovery, establishing interactions between them, which (1) performs iterative clustering to infer the number of modes and mode labels end-to-end, eliminating pre-specification overhead, and (2) minimizes CMI based on cluster labels to achieve disentanglement, using a dynamic architecture that seamlessly adapts to the evolving number of clusters, while maintaining training stability.

- **Meta-Coordination**: We propose a *meta-coordination mechanism*: (1) to disentangle representations while preserving mode information, it learns the weights of marginal representations in CMI minimization via meta-optimization; (2) to refine clustering and mitigate within-cluster correlation, it utilizes these weights to differentiate correlated modes within clusters and guide cluster splits, thus enabling mutual enhancement.

- **Extensive Evaluations**: Experiments on seven datasets demonstrate the superiority of CoDID for robust attribute prediction under correlation shift and out-of-distribution tasks, outperforming the best baseline by an average of 7.8% in accuracy and 7.9% in macro F1 score. Comprehensive investigations reveal the indicative patterns of the learned weights, and validate improved disentanglement and clustering.

## 2. Related Work

**Disentangled Representation Learning.** DRL methods can be roughly divided into unsupervised, weakly-supervised, and supervised DRL. Unsupervised DRL learns independent representation dimensions that each correspond to an unknown attribute by self-supervision, e.g., variational auto-encoding (Kim & Mnih, 2018; Chen et al., 2018). Yet, the feasibility of purely unsupervised disentanglement has been questioned (Locatello et al., 2019), which prompts DRL with weak supervision, e.g., similarity (Chen & Batmanghelich, 2020) or grouping information (Bouchacourt et al., 2018). Supervised DRL learns one multi-dimensional representation subspace for each labeled attribute (Qian et al., 2021; Yuan et al., 2021). Generally, DRL methods assume attribute independence and enforce representation independence between different attributes for disentanglement, e.g., MI minimization for supervised DRL (Yuan et al., 2021; Su et al., 2022). Under correlations, enforcing independence constraints hinders disentanglement, causing representation entanglement for unsupervised DRL (Träuble et al., 2021), and hurting representation informativeness for supervised DRL (Funke et al., 2022).

**DRL Under Correlations.** For DRL under attribute correlations, partial supervision (Träuble et al., 2021; Dittadi et al., 2021), relaxed independence constraints (Roth et al., 2023; Wang & Jordan, 2024), and graph modeling of attribute relations (Xie et al., 2024) have shown some success, but lack guarantees for disentanglement. Conditional independence constraints has been proven to be sufficient for DRL under correlations in linear regression cases (Funke et al., 2022), but requires supervision of the conditioning variables. For example, A-CMI (Funke et al., 2022) minimizes CMI based on known attribute labels; CMID (Dunion et al., 2023) minimizes CMI based on known history variables to bypass unknown current variables. Under mode-related hidden correlations, DRL can be achieved by minimizing CMI based

on estimated mode labels. Since mode discovery and disentanglement modules are mutually dependent, it is non-trivial to design an end-to-end method that can iteratively refine mode estimation for achieving DRL.

## 3. Methodology

### 3.1. Problem Definition

In supervised DRL, we are given data $\boldsymbol{x}$ with $Q$ labeled attributes $\boldsymbol{a} = (a_1, a_2, ..., a_Q) \in \mathcal{A}_1 \times \mathcal{A}_2 \times ... \times \mathcal{A}_Q$. When a certain attribute $a_q$ $(1 \leq q \leq Q)$ takes a value $a_q = \alpha$, the underlying variations related to $a_q$ might lead to a multi-modal data distribution $p(\boldsymbol{x}|a_k = \alpha)$, e.g., a Gaussian mixture. Each high-density region of the distribution is referred to as a *mode*. The number of modes under $a_q = \alpha$ is denoted as $K^{\mathrm{m}}(\alpha)$. The modes are indexed as $0, 1, ..., \sum_\alpha K^{\mathrm{m}}(\alpha)$. The categorical variable $m_q$ indicates which mode each sample comes from. Mutual information is used to measure correlations between variables, denoted as $I(\cdot\,;\cdot)$. This certain $a_q$ may exhibit various correlations with other attributes $a_{-q} = \{a_j\}_{j\neq q}$, including attribute correlations $I(a_q; a_{-q})$, and hidden correlations $I(m_q; a_{-q}|a_q) = \sum_\alpha p_{a_q}(a_q = \alpha)I(m_q; a_{-q}|a_q = \alpha)$ concerning the modes under each attribute value $\alpha$. Take human activity data as an example, the activity attribute and user ID attribute can be denoted as $a_1, a_2$, respectively, and the modes under each activity can be denoted as $m_1$, as shown in Figure 1. See the corresponding data generation process in Appendix B.

We focus on learning the disentangled representation $\boldsymbol{z}_q$ for this certain attribute $a_q$ under hidden correlations and attribute correlations, where representations $\boldsymbol{z} \in \mathbb{R}^{Q \times d_z}$ of dimension $d_z$ are obtained by the mapping $f(\boldsymbol{x}) = \boldsymbol{z} = (\boldsymbol{z}_1, \boldsymbol{z}_2, ..., \boldsymbol{z}_Q)$. For simplicity, we focus on the two-attribute case $(Q = 2, q = 1)$, which can be extended to multiple attributes $(Q > 2)$, as shown in Appendix E, F.5.

### 3.2. The Cause of Error Amplification

Mode-based CMI is a natural solution for handling hidden correlations with underlying modes. Yet, true mode labels are often unavailable. To estimate mode labels, one could perform clustering, and the obtained cluster labels $c_1$ can be used as proxies. In this case, the effectiveness of CMI minimization depends on whether clustering can provide proper guidance. Before introducing the iterative framework, we validate mode-based CMI minimization as the proper conditional independence constraint for DRL, and analyze the potential errors in using cluster labels as proxies.

**Motivating Mode-based CMI Minimization.** We show that mode-based CMI minimization is the proper independence constraint under hidden correlations with the following results: ❶ Attribute-based CMI minimization causes the loss of mode or attribute information under hidden correlations. ❷ Mode-based conditional independence $I(\boldsymbol{z}_1^1; \boldsymbol{z}_2^1|m_1) = 0$ is a property of the latent representations $\boldsymbol{z}_1^1, \boldsymbol{z}_2^1$ that generate the data. These results extend the propositions in (Funke et al., 2022) by introducing modes $m_1$ under $a_1$. See propositions in Appendix C.

**The Harm of Within-Cluster Correlations.** Consider the clustering error where data from multiple modes are mixed into one cluster. Due to the local patterns of hidden correlations, this cluster may exhibit correlations between modes and the other attribute. We define *within-cluster correlations* as the expectation of such correlations over clusters $c_1$, i.e., $I(m_1; a_2|c_1) = \sum_k I(m_1; a_2|c_1 = k)$. Under such correlations, clustering misguides CMI minimization and degrades representation informativeness, as formalized in Proposition 1. Specifically, representations $\boldsymbol{z}_1$ might lose mode information, which makes the mode boundaries more ambiguous, escalates clustering errors and within-cluster correlations, and creates a cycle of error amplification.

**Proposition 1.** *If* $I(m_1; a_2|c_1) > 0$, *then enforcing* $I(\boldsymbol{z}_1; \boldsymbol{z}_2|c_1) = 0$ *leads to at least one of* $I(\boldsymbol{z}_1; m_1) < H(m_1)$ *or* $I(\boldsymbol{z}_2; a_2) < H(a_2)$.

This proposition extends the results in (Funke et al., 2022), see Appendix D.1 for the proof. The MI between a representation and an attribute measures the amount of information the representation contains about the attribute, and $I(\boldsymbol{z}_1; m_1) < H(m_1)$, $I(\boldsymbol{z}_2; a_2) < H(a_2)$ indicate that $\boldsymbol{z}_1, \boldsymbol{z}_2$ lose mode/attribute information about $m_1, a_2$, respectively. The other type of clustering error, where data from one mode are divided into multiple clusters, does not degrade informativeness and can be mitigated as DRL progresses, as demonstrated in Appendix D.

**Interpretations.** Since $I(\boldsymbol{z}_1; \boldsymbol{z}_2|c_1) = 0$ if and only if $p(\boldsymbol{z}_1, \boldsymbol{z}_2|c_1) = p(\boldsymbol{z}_1|c_1)p(\boldsymbol{z}_2|c_1)$, minimizing CMI equals to aligning the joint distribution $p(\boldsymbol{z}_1, \boldsymbol{z}_2|c_1)$ and the marginal distribution $p(\boldsymbol{z}_1|c_1)p(\boldsymbol{z}_2|c_1)$ of representations $(\boldsymbol{z}_1, \boldsymbol{z}_2)$. For the corresponding labels $(m_1, a_2)$, within-cluster correlations indicate a mismatch between their joint and marginal distributions, i.e., $I(m_1; a_2|c_1) > 0 \rightarrow p(m_1; a_2|c_1) \neq p(m_1|c_1)p(a_2|c_1)$, as shown in Figure 2(a). As a result, to exactly align the joint and marginal distributions of representations, input data of different $(m_1, a_2)$ labels must be mapped to the same regions in the latent space, corrupting the mode/attribute patterns in the learned representations, as shown in Figure 2(b).

**Our Solution.** To preserve mode information, we introduce weighting for marginal representations in the CMI minimization objective (Figure 2(c)). By meta-optimization, weights are learned to maximize the informativeness of the resulting representations after CMI minimization. The informativeness of representations can only be preserved

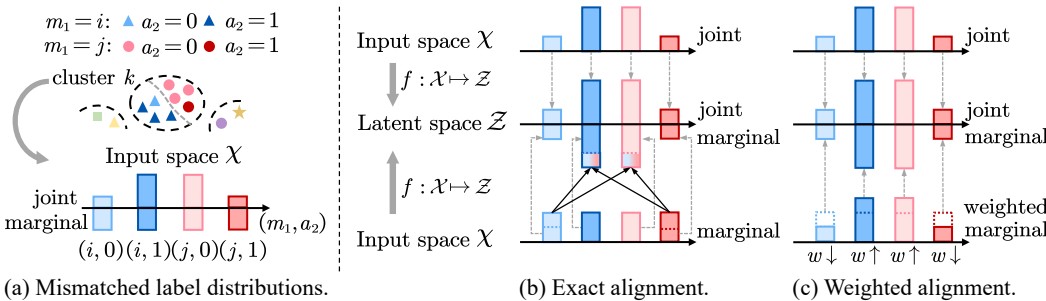

(a) Mismatched label distributions.      (b) Exact alignment.      (c) Weighted alignment.

*Figure 2.* **Within-cluster correlations**. (a): In cluster $k$, for labels $(m_1, a_2) = (i, 0), (i, 1), (j, 0), (j, 1)$, their joint distribution $p(m_1, a_2) = 12.5\%, 37.5\%, 37.5\%, 12.5\%$, respectively, which reflects mode-specific conditional probabilities of $a_2$ under within-cluster correlations; their marginal distribution $p(m_1)p(a_2) = 25\%$ for each label combination, as $p(m_1 = i) = p(m_1 = j) = p(a_2 = 0) = p(a_2 = 1) = 50\%$. (b): Exact alignment causes cross-label mapping due to the mismatched joint and marginal label distributions. (c): Weighted alignment can avoid this issue if the ideal mode-specific weights are assigned, i.e., $w = 0.5, 1.5$ for marginal representations with $a_2 = 0, 1$ in mode $i$, respectively, and $w = 1.5, 0.5$ for marginal representations with $a_2 = 0, 1$ in mode $j$, respectively.

when the weighted marginal label distribution matches the joint label distribution, i.e., when weights $w$ satisfy $w \cdot p(m_1|c_1)p(a_2|c_1) \propto p(m_1, a_2|c_1)$. This yields $w|_{a_2, m_1=i, c_1=k} : w|_{a_2, m_1=j, c_1=k} = p(a_2|m_1 = i, c_1 = k) : p(a_2|m_1 = j, c_1 = k)$, which shows that, for a given $a_2$ value in cluster $k$, the weights across modes reflect mode-specific conditional probabilities under within-cluster correlations. Such weight patterns naturally differentiate correlated modes, and can be used to guide cluster splits for mitigating within-cluster correlations.

### 3.3. Coordinated Disentanglement with Mode Discovery

The framework of CoDID is presented in Figure 3, which consists of disentanglement module for learning the disentangled representation of attribute $a_1$, mode discovery module for inferring the mode labels under each value of $a_1$, and meta-coordination mechanism for preventing error amplification. Our framework is architecture-agnostic and adaptable to various applications.

**Disentanglement module.** Representations are extracted from data by encoder $F$, i.e., $z = (z_1, z_2) = F(x)$, then passed to the following subnetworks: attribute predictors $C_1, C_2$ and mode predictor $C_1^m$ for supervised learning, where cluster labels are used as proxies for mode labels, decoder $R$ for learning to reconstruct data, and discriminator $D$ for minimizing weighted CMI based on cluster labels. When the number of clusters changes, only the layers that take cluster labels as input conditions or output related probabilities are reconfigured, while other layers remain unchanged (Appendix F.4), avoiding abrupt changes.

For weighted CMI minimization, representations are sampled from their joint distribution $p(z_1, z_2|c_1)$ and marginal distribution $p(z_1|c_1)p(z_2|c_1)$ by the following procedure: The samples in each mini-batch are partitioned by cluster labels $c_1$; in each cluster, the joint representations are formulated by concatenating $z_1, z_2$ from the same sample, and the marginal representations are formulated by concatenating

$z_1$ with the $z_2$ shuffled within this cluster. Then, representations and the cluster labels are passed to discriminator $D$ to align the distributions with adversarial training (Belghazi et al., 2018; Funke et al., 2022; Dunion et al., 2023).

**Mode discovery module.** Under each attribute value $a_1 = \alpha$, we perform non-parametric clustering on representations $z_1$ using a Dirichlet Process Gaussian Mixture Model (DPGMM) (Ronen et al., 2022), which adaptively determines the number of clusters $K^c(\alpha)$ and outputs cluster probabilities $r^c$. For each $z_1$, each dimension of vector $r^c$ indicates the probability of $z_1$ being assigned to a cluster under an attribute value. Cluster labels are obtained as $c_1 = \arg\max_k r_k^c$. Clustering results are iteratively refined after every $N_{cl}$ epochs of disentanglement module training.

To learn the number of clusters $K^c$, cluster split/merge operations are proposed and are accepted based on the Hastings ratio (Hastings, 1970). For split operations, each cluster is modeled with a 2-component GMM, which identifies potential subclusters and outputs subcluster probabilities $r^{sc} = (r_1^{sc}, r_2^{sc})$ for each $z_1$. $r^{sc}$ is sent into meta-coordination mechanism to calculate the refined subcluster probabilities, which are used in split proposals. See Appendix G for more details about DPGMM.

**Meta-Coordination Mechanism.** At the core of our framework lies the meta-coordination mechanism, which consists of weight net $W$ for preventing the harm of within-cluster correlations in disentanglement, and subclustering net $S$ for mitigating within-cluster correlations in mode discovery.

Weight net $W$ compute the weights $w$ of marginal representations $(z_1, z_2)$ using three inputs: (1) conditioning labels $(c_1, a_2)$, as the ideal weights with $w \cdot p(m_1|c_1)p(a_2|c_1) \propto p(m_1, a_2|c_1)$ are dependent on the cluster $c_1$ and attribute $a_2$; (2) the discrimination loss of marginal representations, as low similarity of marginal representations to joint representations may indicate underlying label distribution mismatch; (3) the prediction losses of $z_1$ and $z_2$ to assess representation quality, following (Shu et al., 2019; Cai et al.,

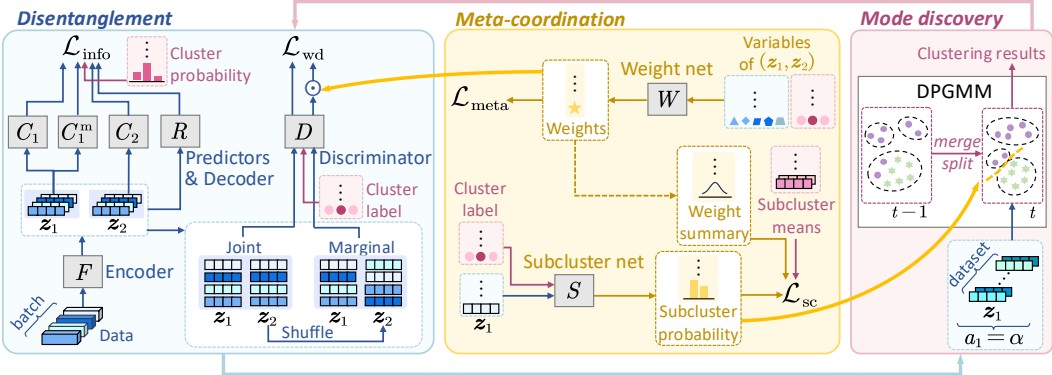

*Figure 3.* **The framework of CoDID**: mode discovery module produces clustering results for disentanglement; disentanglement module outputs representations for mode discovery. Clustering is performed under each $a_1$ value, $\alpha$. "$\odot$" denotes element-wise multiplication.

2020). Configured as a multi-layer perceptron, $W$ is capable of learning expressive weighting functions that adapt to data patterns while preserving theoretical constraints.

Subcluster net $S$ refines DPGMM subcluster probabilities $\boldsymbol{r}^{\mathrm{sc}}$ by computing subcluster probabilities $\hat{\boldsymbol{r}}^{\mathrm{sc}} = (\hat{r}_1^{\mathrm{sc}}, \hat{r}_2^{\mathrm{sc}})$ from representation $\boldsymbol{z}_1$ and its cluster label $c_1$. While DPGMM relies solely on representation distributions and may fail to identify meaningful subclusters when mode structures are ambiguous, $S$ leverages *weight summaries* to capture the mode-identifying patterns from the learned weights. This enables $S$ to differentiate correlated modes and assign them to distinct subclusters, enabling cluster splits that mitigate within-cluster correlations.

**Losses.** The losses for disentanglement module are formulated as follows. See Appendix F.5 for the training process.

(1) To learn informative representations, the *informative loss* $\mathcal{L}_{\mathrm{info}}$ combines *reconstruction loss* $\mathcal{L}_{\mathrm{re}}$ to retain data semantics (including attributes and underlying modes) (Zeng et al., 2023), and *attribute prediction loss* $\mathcal{L}_{\mathrm{ap}}$, *mode prediction loss* $\mathcal{L}_{\mathrm{mp}}$ to encode attribute and mode information:

$$\mathcal{L}_{\mathrm{ap}} = \mathbb{E}_{\boldsymbol{x}}[\sum_{i=1}^2 h_{\mathrm{ce}}(C_i(\boldsymbol{z}_i), a_i)] \tag{1}$$

$$\mathcal{L}_{\mathrm{mp}} = \mathbb{E}_{\boldsymbol{x}}[h_{\mathrm{kld}}(C_1^{\mathrm{m}}(\boldsymbol{z}_1), \boldsymbol{r}^{\mathrm{c}})] \tag{2}$$

$$\mathcal{L}_{\mathrm{re}} = \mathbb{E}_{\boldsymbol{x}}[h_{\mathrm{mse}}(R(\boldsymbol{z}_1, \boldsymbol{z}_2), \boldsymbol{x})] \tag{3}$$

$$\mathcal{L}_{\mathrm{info}} = \mathcal{L}_{\mathrm{ap}} + \lambda_{\mathrm{mp}} \cdot \mathcal{L}_{\mathrm{mp}} + \lambda_{\mathrm{re}} \cdot \mathcal{L}_{\mathrm{re}} \tag{4}$$

where $\lambda_{\mathrm{mp}}, \lambda_{\mathrm{re}}$ are loss coefficients, $h_{\mathrm{ce}}(\cdot), h_{\mathrm{mse}}(\cdot), h_{\mathrm{kld}}(\cdot)$ indicate cross-entropy, mean-squared error, and Kullback-Leibler Divergence, respectively, and cluster probabilities $\boldsymbol{r}^{\mathrm{c}}$ are used as soft proxies for mode labels.

(2) To enforce conditional independence for disentanglement, we formulate *weighted discrimination loss* $\mathcal{L}_{\mathrm{wd}}$. This loss evaluates the Jensen-Shannon Divergence between the joint and marginal representation distributions (Hjelm et al., 2019), with weights $w$ assigned to marginal representations:

$$\begin{aligned} \mathcal{L}_{\mathrm{wd}} = \mathbb{E}_{c_1}[\mathbb{E}_{(\boldsymbol{z}_1,\boldsymbol{z}_2)|c_1}[l_{\mathrm{bce}}(D(\boldsymbol{z}_1, \boldsymbol{z}_2, c_1), 1)] \\ + \mathbb{E}_{\boldsymbol{z}_1|c_1, \boldsymbol{z}_2|c_1}[w \cdot l_{\mathrm{bce}}(D(\boldsymbol{z}_1, \boldsymbol{z}_2, c_1), 0)]] \end{aligned} \tag{5}$$

where $h_{\mathrm{bce}}(\cdot)$ denotes binary cross-entropy. For a marginal representation $(\boldsymbol{z}_1, \boldsymbol{z}_2)$, we denote its attribute and mode prediction losses as $l_i^{\mathrm{ap}} = h_{\mathrm{ce}}(C_i(\boldsymbol{z}_i), a_i), i = 1, 2$, $l_1^{\mathrm{mp}} = h_{\mathrm{ce}}(C_1^{\mathrm{m}}(\boldsymbol{z}_1), \boldsymbol{r}_1)$, and its discrimination loss as $l^{\mathrm{d}} = l_{\mathrm{bce}}(D(\boldsymbol{z}_1, \boldsymbol{z}_2, c_1), 0)$. The weight is computed as:

$$w = W(l_1^{\mathrm{ap}}, l_2^{\mathrm{ap}}, l_1^{\mathrm{mp}}, l^{\mathrm{d}}, c_1, a_2) \tag{6}$$

(3) To learn proper weights that preserve representation informativeness during CMI minimization, we formulate *meta-informativeness loss* $\mathcal{L}_{\mathrm{meta}}$ based on meta-optimization, which uses one optimization process to tune another. For clarity, we only track the encoder parameters $\phi_{\mathrm{f}}$ and weight net parameters $\phi_{\mathrm{w}}$ in the computations. First, we optimize encoder parameters $\phi_{\mathrm{f}}$ w.r.t. weighted discrimination loss $\mathcal{L}_{\mathrm{wd}}$ with learning rate $\beta$, obtaining $\tilde{\phi}_{\mathrm{f}}$:

$$\tilde{\phi}_{\mathrm{f}}(\phi_{\mathrm{w}}) = \phi_{\mathrm{f}} - \beta \nabla_{\phi_{\mathrm{f}}} \mathcal{L}_{\mathrm{wd}}(\phi_{\mathrm{f}}, \phi_{\mathrm{w}}) \tag{7}$$

Then, we calculate $\mathcal{L}_{\mathrm{meta}}$ to evaluate the informativeness of representation $\tilde{\boldsymbol{z}}_1$ obtained after the optimization step w.r.t. CMI minimization, $(\tilde{\boldsymbol{z}}_1(\phi_{\mathrm{w}}), \tilde{\boldsymbol{z}}_2(\phi_{\mathrm{w}})) = F(\boldsymbol{x}; \tilde{\phi}_{\mathrm{f}}(\phi_{\mathrm{w}}))$. Attribute prediction loss and reconstruction loss are used to assess whether $\tilde{\boldsymbol{z}}_1$ retains the information for predicting attribute $a_1$ and for encoding mode structures in data distributions. $\mathcal{L}_{\mathrm{meta}}$ is formulated with coefficient $\lambda_{\mathrm{mr}}$ as:

$$\begin{aligned} \mathcal{L}_{\mathrm{meta}}(\phi_{\mathrm{w}}) = \mathbb{E}_{\boldsymbol{x}}[h_{\mathrm{ce}}(C_1(\tilde{\boldsymbol{z}}_1(\phi_{\mathrm{w}})), a_1) \\ + \lambda_{\mathrm{mr}} \cdot h_{\mathrm{mse}}(R(\tilde{\boldsymbol{z}}_1(\phi_{\mathrm{w}}), \tilde{\boldsymbol{z}}_2(\phi_{\mathrm{w}})), \boldsymbol{x})] \end{aligned} \tag{8}$$

(4) To obtain the refined subcluster probabilities, we train subcluster net $S$ to reproduce the subclustering of DPGMM. For a representation $\boldsymbol{z}_1$ with cluster label $c_1$ and subcluster means $\{\boldsymbol{\mu}_{c_1,j}^{\mathrm{sc}}\}_{j=1}^2$, $S$ learns to assign $\boldsymbol{z}_1$ to the subcluster whose center is closer to $\boldsymbol{z}_1$, aligning with DPGMM (Ronen et al., 2022). We formulate *isotropic loss* $\mathcal{L}_{\mathrm{iso}}$ as:

$$\mathcal{L}_{\mathrm{iso}} = \mathbb{E}_{\boldsymbol{x}}[\sum_{j=1}^2 \hat{r}_j^{\mathrm{sc}} \cdot ||\boldsymbol{z}_1 - \boldsymbol{\mu}_{c_1,j}^{\mathrm{sc}}||_{\ell_2}^2] \tag{9}$$

We formulate weight summaries for refining the outputs of subcluster net. Recall that the ideal weights have the

property $w|_{a_2,m_1=i,c_1=k} : w|_{a_2,m_1=j,c_1=k} = p(a_2|m_1 = i, c_1 = k) : p(a_2|m_1 = j, c_1 = k)$, i.e., in a cluster $c_1 = k$, marginal representations of the same $a_2$ label will have different weights across modes $m_1 = i, j$ under within-cluster correlations. To infer whether two $z_1$ samples are from the same mode, we first construct a weight summary $\mathcal{P}^{\text{w}}$ for each $z_1$ sample to record the weight distribution under each $a_2$ label: $\mathcal{P}^{\text{w}} = \{\mathcal{N}(\mu(\eta), \sigma^2(\eta))\}_{\eta \in \mathcal{A}_2}$, where we gather $z_2$ samples of label $a_2 = \eta$, calculate the weights of $(z_1, z_2)$, and fit these weights with a Gaussian distribution $w \sim \mathcal{N}(\mu(\eta), \sigma^2(\eta))$ for each $\eta$.

Under the same $a_2$ label, $z_1$ samples with more divergent weight distributions are more likely to originate from different modes, thus $S$ is encouraged to assign them to different subclusters via *weight alignment loss* $\mathcal{L}_{\text{wa}}$. This loss and the overall *subclustering loss* $\mathcal{L}_{\text{sc}}$ are formulated as follows, with coefficient $\lambda_{\text{wa}}$, Jensen-Shannon divergence $h_{\text{jsd}}(\cdot)$ and the sample-size weighted average $\bar{h}_{\text{jsd}}(\cdot)$ over $\eta$:

$$\mathcal{L}_{\text{wa}} = \mathbb{E}_{c_1}[\mathbb{E}_{z_{1,i}|c_1, z_{1,j}|c_1, z_2|c_1}[h_{\text{jsd}}(\hat{r}_i^{\text{sc}}, \hat{r}_j^{\text{sc}})/\bar{h}_{\text{jsd}}(\mathcal{P}_i^{\text{w}}, \mathcal{P}_j^{\text{w}})]]] \tag{10}$$

$$\mathcal{L}_{\text{sc}} = \mathcal{L}_{\text{iso}} + \lambda_{\text{wa}} \cdot \mathcal{L}_{\text{wa}} \tag{11}$$

## 4. Experiments

### 4.1. Experimental Setup

**Datasets.** We evaluate on toy data and seven real-world datasets, i.e., Colored MNIST (**CMNIST**) (Arjovsky et al., 2019a), Colored Fashion-MNIST (**CFashion-MNIST**) (Xiao et al., 2017), Canine-Background (**Canine-BG**) constructed from ImageNet (Deng et al., 2009), **UCI-HAR** (Anguita et al., 2013), **RealWorld** (Sztyler & Stuckenschmidt, 2016), **HHAR** (Stisen et al., 2015), and **MFD** (Lessmeier et al., 2016). Attributes $a_1, a_2$ represent the parity ("even", "odd") and color of digits on CMNIST, the style ("sporty", "chic") and color of clothing on CFashion-MNIST, the functional categories ("work", "pet") and image backgrounds ("indoors", "outdoors") of dogs on Canine-BG, fault type and operating condition on MFD, and activity and user ID on other human activity datasets. On CMNIST, CFashion-MNIST, and Canine-BG, we construct modes under each $a_1$ value as digits (e.g., "8" under parity "even"), clothing category (e.g., "sneaker" under style "sporty"), and dog breeds (e.g., "standard poodle" under functional category "pet"), respectively; on time series datasets, we assume natural underlying multi-modal data distributions. See Appendix F.1 for details.

**Evaluation Protocols.** We evaluate under introduced or natural correlations. *On CMNIST and CFashion-MNIST, and Canine-BG with constructed modes, we introduce correlations by sampling.* We train on correlated data and evaluate on three test sets with increasing train-test correlation shifts, test 1 (same correlations), test 2 (no correlations), and test 3

(anticorrelations). In Table 1, we train under both attribute and hidden correlations and report the results on test 3 (see Appendix H for full results). *On time series datasets, we assume natural correlations* (Wu et al., 2025). By leave-one-group-out validation based on $a_2$, training and test sets show data distribution shifts on disjoint $a_2$ values, e.g., different training/test users with distinct behaviors and various correlations with activities. See details in Appendix F.2.

**Metrics.** Our task is to learn the disentangled representation of $a_1$ for robust attribute prediction, evaluated by accuracy (Acc.) and macro F1 score (Mac. F1). This reflects disentanglement, as only disentangled representations can support robust prediction under various train-test shifts (Funke et al., 2022). Other disentanglement and clustering metrics are reported in Appendix A.1, A.4. For statistical tests, each experiment is repeated using 5 varying seeds.

**Baselines and Implementations.** The code of CoDID is available at GitHub[1]. We compare with common DRL and invariant learning methods (**MMD** (Lin et al., 2020), **DANN** (Ganin et al., 2016), **CORAL** (Sun & Saenko, 2016), **IRM** (Arjovsky et al., 2019b), **REx** (Krueger et al., 2021), **DTS** (Li et al., 2022), **IDE-VC** (Yuan et al., 2021), **MI** (Cheng et al., 2022), and **ID-FaceVC** (Rong & Liu, 2025)), and the DRL methods that address correlations (**A-CMI** (Funke et al., 2022), **HFS** (Roth et al., 2023), **DIOSC** (Oublal et al., 2024a), and **CODA** (Ou & Zhang, 2024)). For reference, we also include **BASE** with only supervised learning. Details of baselines, implementations, network architectures, and hyperparameters are in Appendix F.

### 4.2. Baseline Comparison

The baseline comparison results in Table 1 show that:

(1) CoDID consistently outperforms the best DRL baseline by an average of 7.8% accuracy and 7.9% macro F1 score across all datasets, and consistently outperforms BASE by an average of 10.3% accuracy and 10.4% macro F1 score across all datasets. This indicates that minimizing mode-based CMI can preserve important mode information, and that our iterative framework with meta-coordination mechanism can effectively prevent catastrophic error amplification, yielding reliable clustering and robust representations for attribute prediction. Under introduced correlations, the significant advantage on Canine-BG indicates better generalization on complex image data under attribute and hidden correlation shifts. Under natural correlations, the significant advantage on UCI-HAR and MFD demonstrates the practical value in improving generalization on real-world OOD data with underlying multi-modal distributions.

(2) In contrast, the baselines underperform BASE in some cases. Common DRL and invariant representation learning

---

[1] https://github.com/Rxannro/CoDID

*Table 1.* Baseline comparison (mean±std.). "*" indicates that CoDID is statistically superior to the baseline by pairwise t-test at a 95% significance level. The best results are **bold**. The runner-up results are underlined. The improvement over the best baseline is calculated.

| Method | CMNIST Acc. | CMNIST Mac. F1 | CFashion-MNIST Acc. | CFashion-MNIST Mac. F1 | Canine-BG Acc. | Canine-BG Mac. F1 | UCI-HAR Acc. | UCI-HAR Mac. F1 | Realworld Acc. | Realworld Mac. F1 | HHAR Acc. | HHAR Mac. F1 | MFD Acc. | MFD Mac. F1 |
|---|---|---|---|---|---|---|---|---|---|---|---|---|---|---|
| BASE | 79.2±0.9* | 78.3±0.9* | 77.8±1.5* | 77.1±1.5* | 55.6±2.0* | 54.6±2.3* | 71.2±2.8* | 69.7±3.6* | 64.6±1.4* | 65.4±1.4* | 80.8±1.6* | 80.9±2.0* | 72.7±1.6* | 76.3±0.9* |
| MMD | 57.3±2.7* | 41.2±7.0* | 63.5±3.6* | 62.7±3.7* | 54.1±6.7* | 52.1±9.4* | 70.3±3.7* | 66.2±3.5* | 66.0±1.9 | 65.2±2.3* | 80.9±1.2* | 80.5±1.7* | 78.2±1.9* | 79.1±1.6* |
| DANN | 58.3±2.9* | 56.9±2.6* | 63.8±2.5* | 62.5±2.8* | 29.8±4.4* | 29.6±4.8* | 67.8±3.1* | 65.1±3.0* | 66.0±1.7 | 65.1±2.0* | 77.1±1.8* | 76.7±1.5* | 74.0±2.1* | 76.3±1.4* |
| CORAL | 58.6±3.2* | 57.6±3.0* | 68.0±3.4* | 67.3±3.0* | 51.0±3.5* | 33.8±3.2* | 74.4±2.8* | 72.7±3.3* | 64.8±1.9* | 65.9±1.6* | 81.0±1.4* | 81.2±1.7* | 77.3±1.8* | 77.9±1.5* |
| IRM | 81.0±3.5* | 80.3±3.2* | 79.3±2.6 | 78.9±2.5* | 54.3±5.6* | 54.3±5.1* | 73.0±3.0* | 69.6±3.4* | 65.4±1.6* | 65.4±1.5* | 82.5±1.3* | 82.5±1.2* | 78.4±1.7* | 79.9±1.3* |
| REx | 81.7±3.3 | 80.9±3.6* | 76.3±3.5* | 75.9±3.2* | 56.0±5.2* | 56.0±5.7* | 73.7±2.7* | 73.2±2.8* | 65.1±1.5* | 65.3±1.7* | 80.6±1.5* | 80.4±1.6* | 78.8±1.9* | 80.3±1.4* |
| DTS | 55.6±3.0* | 44.0±1.9* | 61.2±2.0* | 60.1±2.0* | 41.8±4.7* | 35.7±2.3* | 72.8±3.3* | 70.1±2.6* | 64.4±2.3* | 64.9±1.5* | 79.8±2.4* | 79.7±1.7* | 67.0±2.2* | 67.4±1.5* |
| IDE-VC | 54.2±4.0* | 49.1±2.9* | 58.7±3.8* | 57.1±4.3* | 49.1±2.2* | 45.7±2.0* | 73.6±3.1* | 73.2±3.4* | 65.2±1.3* | 65.0±1.7* | 80.7±2.0* | 80.6±1.4* | 74.1±1.8* | 76.3±1.1* |
| MI | 56.9±4.0* | 45.2±2.8* | 62.1±3.0* | 60.7±3.0* | 52.3±4.8* | 51.0±4.4* | 74.9±2.1* | 74.5±2.7* | 66.0±1.8* | 65.5±1.6* | 80.9±1.7* | 80.7±2.1* | 76.3±1.2* | 77.6±1.6* |
| A-CMI | 58.5±2.3* | 41.2±5.9* | 61.8±5.3* | 60.0±6.5* | 53.0±8.2* | 52.8±8.1* | 71.4±3.4* | 70.0±3.0* | 65.4±1.5* | 65.5±1.2* | 80.2±1.8* | 80.3±2.3* | 78.8±1.4* | 79.8±0.7* |
| HFS | 66.5±2.1* | 64.9±2.1* | 66.2±3.6* | 65.3±3.1* | 46.8±3.7* | 46.2±4.1* | 67.1±3.5* | 65.1±4.0* | 48.9±1.8* | 39.8±1.5* | 78.2±1.2* | 78.3±1.5* | 75.4±1.7* | 71.0±1.3* |
| CODA | 69.8±1.9* | 68.3±2.3* | 72.5±1.5* | 71.3±1.9* | 43.9±2.2* | 43.9±2.3* | 71.1±4.2* | 70.4±5.0* | 65.3±0.7* | 66.7±0.6* | 77.9±1.7* | 77.6±1.8* | 62.5±0.8* | 54.6±1.2* |
| ID-FaceVC | 58.1±2.2* | 56.2±2.4* | 63.6±1.4* | 62.5±1.7* | 42.1±6.2* | 41.3±6.7* | 68.0±8.1* | 68.0±9.3* | 65.2±1.5* | 66.2±1.8* | 78.6±1.2* | 77.8±1.3* | 71.1±1.9* | 71.6±1.6* |
| DIOSC | 73.4±0.7* | 72.6±1.0* | 73.2±1.6* | 72.8±1.4* | 42.5±5.4* | 42.0±5.3* | 74.6±3.0* | 74.5±3.7* | 65.2±1.3* | 66.3±1.2* | 79.0±1.4* | 78.6±1.5* | 68.9±1.6* | 67.9±1.4* |
| CoDID | **85.2±2.7** | **84.6±3.0** | **84.1±3.2** | **83.5±3.3** | **68.3±5.2** | **68.2±5.7** | **86.2±0.5** | **86.7±0.6** | **70.9±1.3** | **71.7±1.0** | **88.4±1.1** | **88.3±1.1** | **90.7±1.7** | **91.9±1.7** |
| **Improvement** | +3.5 % | +3.7 % | +4.8 % | +4.6 % | +12.3 % | +12.2 % | +11.3 % | +12.2 % | +4.9 % | +5.0 % | +5.9 % | +5.8 % | +11.9 % | +11.6 % |

*Table 2.* Variant comparison (mean±std.). The notations follow Table 1.

| Method | $\mathcal{L}_{mp}$ | $\mathcal{L}_d$ | IC | IN | $\mathcal{L}_{wd}$ | $\mathcal{L}_{sc}$ | UP | CMNIST Acc. | CMNIST Mac. F1 | CFashion-MNIST Acc. | CFashion-MNIST Mac. F1 | Canine-BG Acc. | Canine-BG Mac. F1 | UCI-HAR Acc. | UCI-HAR Mac. F1 | Realworld Acc. | Realworld Mac. F1 | HHAR Acc. | HHAR Mac. F1 | MFD Acc. | MFD Mac. F1 |
|---|---|---|---|---|---|---|---|---|---|---|---|---|---|---|---|---|---|---|---|---|---|
| BASE | - | - | - | - | - | - | - | 79.2±0.9* | 78.3±0.9* | 77.8±1.5* | 77.1±1.5* | 55.6±2.0* | 54.6±2.3* | 71.2±2.8* | 69.7±3.6* | 64.6±1.4* | 65.4±1.4* | 80.8±1.6* | 80.9±2.0* | 72.7±1.6* | 76.3±0.9* |
| CoDID-km-MP | ✓ | ✗ | ✗ | ✗ | ✗ | ✗ | ✗ | 74.2±1.9* | 73.6±1.9* | 74.2±2.0* | 73.7±2.1* | 63.2±4.6* | 62.2±4.2* | 77.6±5.3* | 77.5±4.5* | 63.7±0.9* | 63.3±0.8* | 83.5±1.2* | 83.4±1.2* | 78.4±2.6* | 80.1±2.5* |
| CoDID-km-MC | ✗ | ✓ | ✗ | ✗ | ✗ | ✗ | ✗ | 73.0±1.4* | 72.1±1.4* | 71.6±1.5* | 70.2±1.3* | 60.9±3.2* | 60.2±3.4* | 80.2±3.3* | 79.7±4.4* | 68.4±0.8 | 68.0±0.9 | 83.8±1.4* | 83.2±1.5* | 81.1±2.1* | 80.2±2.3* |
| CoDID-km-ID | ✓ | ✓ | ✗ | ✗ | ✗ | ✗ | ✗ | 77.1±1.0* | 76.7±1.2* | 73.6±1.4* | 72.8±1.3* | 62.6±4.9* | 62.5±4.7* | 77.6±1.8* | 76.8±2.3* | 68.3±1.2* | 67.8±1.1* | 77.2±1.9* | 75.5±1.5* | 80.6±1.7* | 80.9±1.2* |
| CoDID-km-SD | ✓ | ✓ | ✗ | ✗ | ✗ | ✗ | ✗ | 76.3±1.7* | 75.6±1.5* | 72.9±1.6* | 72.3±1.6* | 61.5±3.5* | 61.4±3.6* | 77.4±1.5* | 76.8±1.8* | 66.2±1.3* | 66.6±1.8* | 81.0±2.4* | 81.2±1.7* | 79.2±1.8* | 79.2±1.3* |
| CoDID-km | ✓ | ✓ | ✗ | ✗ | ✗ | ✗ | ✗ | 79.0±1.8* | 78.3±1.6* | 77.4±2.3* | 76.9±2.2* | 64.9±2.3* | 64.5±2.2* | 83.0±3.0* | 83.3±3.6* | 69.8±1.9 | 69.9±1.4 | 84.5±2.3* | 84.2±1.5* | 82.5±2.0* | 82.5±1.5* |
| CoDID-itkm | ✓ | ✓ | ✓ | ✗ | ✗ | ✗ | ✗ | 76.5±1.0* | 75.1±1.2* | 76.6±1.5* | 76.0±1.5* | 60.5±5.5* | 60.0±5.6* | 79.4±1.6* | 79.1±1.6* | 66.1±1.8* | 65.2±1.8* | 80.5±0.7* | 80.4±0.8* | 79.6±0.9* | 80.0±0.5* |
| CoDID-itdpm | ✓ | ✓ | ✓ | ✓ | ✗ | ✗ | ✓ | 77.6±2.0* | 77.0±2.1* | 77.4±1.3* | 76.8±1.4* | 61.4±3.2* | 61.1±3.5* | 83.2±0.5* | 83.7±0.6* | 65.7±1.3* | 66.0±1.6* | 83.8±0.3* | 83.5±0.3* | 82.2±0.9* | 84.5±1.2* |
| CoDID-w/o-SC | ✓ | ✓ | ✓ | ✓ | ✓ | ✗ | ✓ | 79.1±0.8* | 78.4±0.7* | 78.4±1.6* | 77.9±1.6* | 64.5±3.7* | 64.4±3.7* | 83.5±1.9 | 84.0±0.4* | 65.4±1.9* | 65.3±1.9* | 85.3±0.8* | 85.0±0.8* | 82.9±1.3* | 85.1±1.3* |
| CoDID-UA | ✓ | ✓ | ✓ | ✓ | ✓ | ✓ | ✗ | 82.0±0.7* | 81.3±0.5* | 79.4±1.9* | 78.9±1.9* | 66.2±3.9* | 65.9±3.4* | 84.2±3.2* | 84.6±3.6* | 69.6±1.4 | 70.9±1.5 | 86.0±1.2* | 85.7±1.3* | 85.8±2.4* | 86.9±2.0* |
| CoDID | ✓ | ✓ | ✓ | ✓ | ✓ | ✓ | ✓ | **85.2±2.7** | **84.6±3.0** | **84.1±3.2** | **83.5±3.3** | **68.3±5.2** | **68.2±5.7** | **86.2±0.5** | **86.7±0.6** | **70.9±1.3** | **71.7±1.0** | **88.4±1.1** | **88.3±1.1** | **90.7±1.7** | **91.9±1.7** |

**Notes.** "✓" indicates the design choice is adopted, and "✗" indicates the alternative setting:
- $\mathcal{L}_{mp}$ and $\mathcal{L}_d$ denote these losses, where $\mathcal{L}_d$ denotes the unweighted discrimination loss that removes the weights from $\mathcal{L}_{wd}$.
- IC and IN denote iterative refinement of cluster assignments and the number of clusters, respectively; otherwise, these are fixed after pre-training.
- $\mathcal{L}_{wd}$ and $\mathcal{L}_{sc}$ denote adopting these losses; otherwise, unweighted discrimination loss and DPGMM subcluster probability-based cluster split are adopted instead, respectively.
- PR denotes partial network reconfiguration when the number of clusters changes (only the related input/output neurons); otherwise, all layers are reconfigured or re-initialized.

methods do not consider correlations and enforce independence or invariance constraints on representations, which might degrade representation informativeness under hidden correlations. A-CMI addresses attribute correlations but overlooks modes and hidden correlations under attribute values, thus losing mode information. The disadvantage of these baselines is pronounced on CMNIST, CFashion-MNIST, and Canine-BG, where train-test correlation shifts are prominent. IRM and REx enforce invariant predictions rather than invariant representations, and HFS deals with general correlations. Yet, they lack explicit mode modeling.

### 4.3. Comparison with Variants

We compare with two sets of variants to systematically analyze our design: The CoDID-km variants perform k-means (parametric clustering) on pre-trained representations, which investigate the design of disentanglement module, including the mode-related losses and the discriminator architectures; The other variants perform iterative clustering, which investigate the design related to iterations and

coordinations. **CoDID-km-ID/SD** use individual discriminators and one shared discriminator for different modes, respectively, while other variants shares discriminator parameters among the modes under the same attribute value. **CoDID-itkm** performs iterative k-means during DRL, and **CoDID-itdpm** performs iterative DPGMM (non-parametric clustering) during DRL. On image datasets, the ground-truth number of modes are provided to parametric clustering methods as a prior knowledge, which is unknown for non-parametric methods. The results and other variant descriptions are in Table 2, which show that:

(1) CoDID-km-MC/MP consistently underperform CoDID-km, showing that both discrimination loss and mode prediction loss are crucial for disentanglement. Mode prediction loss guides the representations to encode mode information, and discrimination loss enforces conditional independence to remove redundant information about other attributes.

(2) CoDID-km-ID/SD consistently underperform CoDID-km, indicating that proper discriminator parameter sharing between different modes benefits mode-based CMI mini-

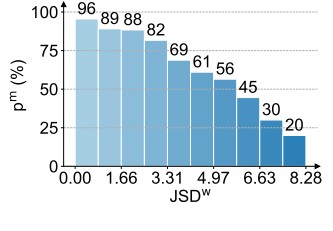

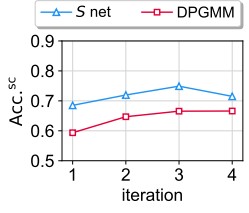

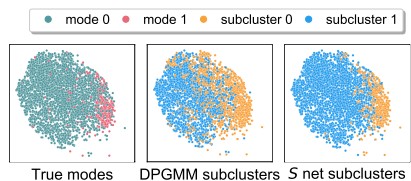

*(a)* Weight patterns

*(b)* Subclustering accuracy

*(c)* Subclustering visualization

*Figure 4.* Weight and subclustering analysis on CMNIST. The cluster $k$ in (c) exhibits statistically significant within-cluster correlations based on the $\chi^2$-test ($p = 4.22 \times 10^{-15}$), and DPGMM and $S$ net achieve subcluster accuracies of 0.678 and 0.892, respectively.

mization. Excessive parameter sharing may limit the ability to capture distinctions among modes, and removing parameter sharing may fail to leverage their commonality.

(3) The iterative CoDID-itkm/itdpm generally underperform the non-iterative CoDID-km, regardless of whether the number of clusters is updated. This is likely due to error amplification, i.e., clustering errors hinder representation disentanglement, which in turn makes clustering more error-prone, leading to more errors than a fixed clustering.

(4) CoDID-w/o-SC outperforms CoDID-dpm, showing the effectiveness of introducing weighted CMI minimization, which can preserve the informativeness of representations under within-cluster correlations. Further, CoDID outperforms CoDID-w/o-SC, demonstrating the benefits of introducing the refined subcluster probabilities for cluster splits, which can mitigate within-cluster correlations.

(5) CoDID outperforms CoDID-UA, indicating that adapting to the evolving number of clusters while reducing network reconfiguration facilitates better disentanglement, likely because avoiding abrupt changes enables stable training. On image datasets, CoDID outperforms CoDID-km and CoDID-itkm, despite their access to the ground-truth number of modes, demonstrating the strong end-to-end performance of CoDID without prior knowledge.

## 5. Model Investigation

**Toy Decision Boundary.** We investigate the effect of mode-based and attribute-based CMI minimization on toy data, where CoDID-T uses the ground-truth mode labels for DRL. The decision boundaries and $a_1$ prediction accuracy are shown in Figure 5, where: (1) The upper right boundaries of BASE surround the clusters at $\boldsymbol{x}_2 = 1$, and its performance decreases as the correlation shift enlarges from test 1 to 3, indicating that BASE over-encodes $a_2$ and lacks generalization ability. (2) The decision boundaries of A-CMI span across the clusters at $\boldsymbol{x}_2 = 0, 1$ without excluding either value, but fail to separate interleaving clusters of different modes. The performance is low but robust across 3 test sets, indicating that A-CMI does not over-encode $a_2$, but loses important mode information. (3) The decision boundaries

of CoDID-T conform to vertical lines $\boldsymbol{x}_1 = b, b \in [0, 5]$ that distinguish interleaving clusters of different modes, and CoDID-T shows robustness and superiority across 3 test sets, indicating that CoDID-T can learn mode information about $a_1$, and exclude irrelevant information about $a_2$.

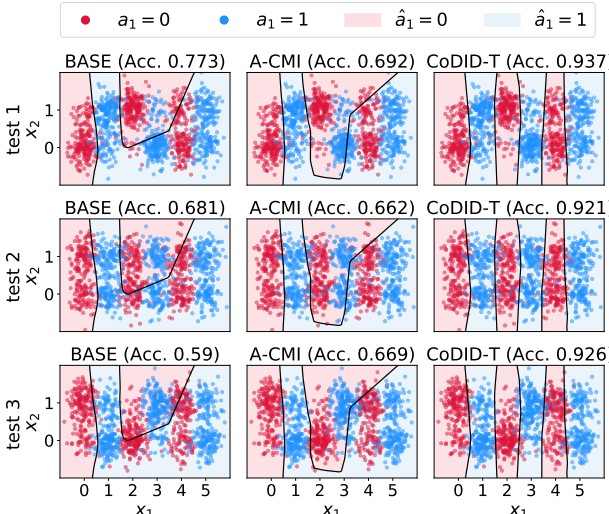

*Figure 5.* Toy decision boundary. Attributes $a_1, a_2$ control data dimensions $\boldsymbol{x}_1, \boldsymbol{x}_2$, respectively. Clusters centered at $\boldsymbol{x}_1 = 0, 2, 4$ and $1, 3, 5$ represent modes under $a_1 = 0$ and $1$, respectively.

**Weight pattern analysis.** We examine whether the learned weights encode mode-specific patterns by analyzing the association between weight divergence and the ground-truth mode labels of $\boldsymbol{z}_1$. The weight divergence of a pair of $\boldsymbol{z}_1$ samples $(\boldsymbol{z}_{1,i}, \boldsymbol{z}_{1,j})$ is denoted as $\mathrm{JSD}^{\mathrm{w}} = \bar{\mathrm{h}}_{\mathrm{jsd}}(\mathcal{P}_i^{\mathrm{w}}, \mathcal{P}_j^{\mathrm{w}})$ (Equation 11). We randomly sample within-cluster $\boldsymbol{z}_1$ pairs on CMNIST, group them by weight divergence into equal-width bins, and calculate the proportion of pairs from the same modes, denoted as $p^{\mathrm{m}} = p(m_{1,i} = m_{1,j})$. As shown in Figure 4(a), $p^{\mathrm{m}}$ decreases as $\mathrm{JSD}^{\mathrm{w}}$ increases, indicating that representations with a larger weight divergence are less likely to originate from the same modes, exhibiting mode-identifying weight patterns.

**Subclustering analysis.** We evaluate subclustering on CM-NIST using the average per-cluster subclustering accuracy (Acc.$^{\mathrm{sc}}$). As shown in Figure 4(b), subcluster net $S$ consistently yields better subclusterings than DPGMM. This

indicates that weight alignment loss $\mathcal{L}_{\mathrm{wa}}$, which encourages representations with larger weight divergence to be assigned to different subclusters, can effectively utilize the mode-identifying weight patterns to better distinguish modes under within-cluster correlations. Figure 4(c) visualizes the subclusters from a cluster selected for splitting with t-SNE. DPGMM fails to separate ambiguous modes solely based on the representation distribution, while subcluster net can identify the underlying modes using weight patterns and guide a split that mitigates within-cluster correlation.

**Evolution of Representation Distributions.** We visualize the training dynamics of representation distributions using t-SNE, focusing on the two modes of digits 3 and 9 under parity "odd" on CMNIST. Figure 6 shows that: (1) During training, the clustering gradually achieves inter-mode separation and intra-mode compactness, indicating effective mode discovery and preservation of mode information by avoiding error amplification. (2) At epoch 20, the representations of different $a_2$ values are clearly separable, suggesting that $z_1$ over-encodes $a_2$ after pre-training. This separation diminishes as training proceeds, indicating the removal of the information of $a_2$. (3) Under hidden correlations, the information of $a_2$ is partially shared with $m_1$, and the conditional distribution $p(a_2 \mid m_1)$ differs across modes ($a_2 = 0, 1$ at the probability of $(90\%, 10\%)$ in mode 0, and $(10\%, 90\%)$ in mode 1 in this case). CoDID only removes the redundant information of $a_2$, while preserving correlation-induced shared information, thus is able to keep $p(a_2 \mid c_1)$ close to $p(a_2 \mid m_1)$. In contrast, the variants of CoDID exhibit error amplification (see Appendix A.5).

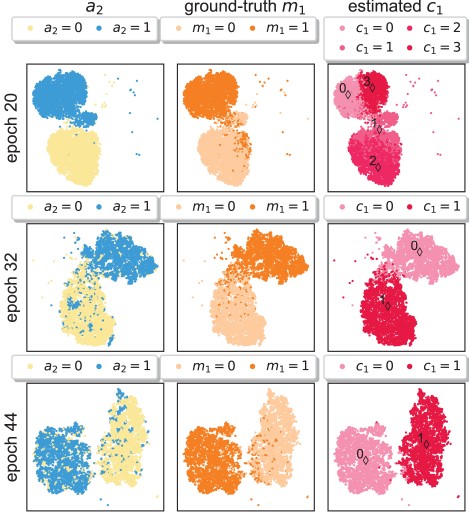

*Figure 6.* Representation distributions of $z_1$ in CoDID. At epoch 20, pre-training is completed, and clustering is initialized. Colors indicate labels $a_2$, $m_1$, or $c_1$. Black-edged diamond markers with numbers indicate the cluster centroids and their indices.

**Computational Complexity** We compare the number of parameters and the training duration of CoDID with the

baselines on CFashion-MNIST: (1) Figure 7(a) shows that CoDID contains the largest number of parameters due to its comprehensive architecture, including decoder, discriminator, weight net, and subclustering net. These components are essential for achieving disentanglement under hidden correlations, while the parameter increase compared to methods that also use decoders and discriminators (e.g., CODA) remains moderate. Subclustering net is a simple classifier; DPGMM mainly includes the means and variances of each Gaussian component; weight net is a light MLP that only occupies 0.07% of the total parameters. (2) Figure 7(b) shows that A-CMI and CoDID have the longest training durations, mainly due to the neural estimation and minimization of CMI (Belghazi et al., 2018). DPGMM clustering is computationally efficient (0.57% of the training duration), because each iteration only requires a few computation steps using GPU vectorization. The training efficiency of CoDID benefits from the following design choices: discriminator parameter sharing across the modes under the same attribute value, vectorized parallel forward computation across all modes, and the partial network reconfiguration corresponding to cluster merge/split (See details in Appendix F.4).

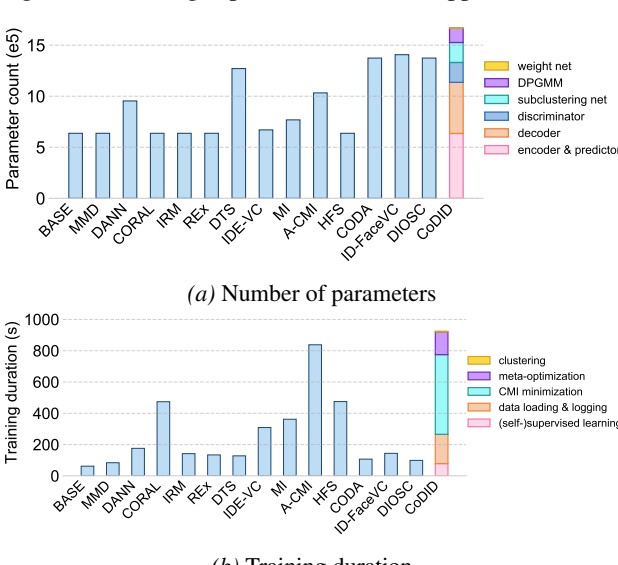

*(a)* Number of parameters

*(b)* Training duration

*Figure 7.* Computational complexity comparison.

# 6. Conclusions

In this work, we propose CoDID, a supervised disentanglement method under hidden correlations, which uses an iterative framework of disentanglement and mode discovery modules to learn the disentangled representations of a certain attribute with underlying modes. Theoretically, we reveal the cause of error amplification in this iterative framework, and design a meta-coordination mechanism to promote mutual enhancement between the two modules. Comprehensive experiments show the superiority of CoDID under correlation shifts and OOD tasks.

## Impact Statement

This research focuses on disentanglement representation learning, aiming to encode one single data attribute in each representation subspace, which holds great promise in enhancing generalization to unseen scenarios and enabling controllable generative modeling. It is essential to note that our work focuses solely on scientific issues, and we also ensure that ethical considerations are carefully taken into account. All the used datasets are publicly available for scientific research. Thus, we believe that there is no ethical risk associated with our research.

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

# A. Additional Experiments and Results

## A.1. Clustering Evaluation with Ground-Truth Mode Labels

We evaluate mode discovery using the average clustering accuracy (Acc.$^{\mathrm{c}}$), adjusted rand index (ARI), and Normalized Mutual Information (NMI) under each $a_1$ value, as summarized in Table 3. Iterative variants (CoDID-itkm, CoDID-itdpm) underperform CoDID-km, indicating error amplification in mode discovery, which is consistent with the trend in attribute prediction (Table 2). CoDID-w/o-SC offers little improvement over CoDID-itdpm, whereas CoDID achieves a substantial gain by refining subclustering, showing that weighted CMI alone preserves informativeness, but does not improve clustering without subclustering refinement using the weight patterns. Moreover, CoDID outperforms CoDID-UA, suggesting that maintaining training stability benefits mode discovery as well. For completeness, we report the clustering metrics at each iteration in Appendix A.2, which shows that all iterative variants degrade over iterations, except CoDID and CoDID-UA, which steadily improve and converge. This indicates the key role of subclustering refinement for enhancing mode discovery.

*Table 3.* Clustering metrics.

| Method | CMNIST | | | CFashion-MNIST | | | Canine-BG | | |
|---|---|---|---|---|---|---|---|---|---|
| | Acc.$^{\mathrm{c}}$ | ARI | NMI | Acc.$^{\mathrm{c}}$ | ARI | NMI | Acc.$^{\mathrm{c}}$ | ARI | NMI |
| CoDID-km | 0.714 | 0.339 | 0.379 | 0.700 | 0.417 | 0.449 | 0.173 | 0.065 | 0.146 |
| CoDID-itkm | 0.435 | 0.002 | 0.004 | 0.342 | 0.014 | 0.034 | 0.087 | 0.032 | 0.075 |
| CoDID-itdpm | 0.563 | 0.256 | 0.253 | 0.602 | 0.448 | 0.482 | 0.129 | 0.054 | 0.103 |
| CoDID-w/o-SC | 0.583 | 0.276 | 0.263 | 0.602 | 0.451 | 0.490 | 0.113 | 0.050 | 0.094 |
| CoDID-UA | 0.622 | 0.395 | 0.388 | 0.649 | 0.462 | 0.497 | 0.204 | 0.137 | 0.186 |
| CoDID | **0.819** | **0.594** | **0.545** | **0.739** | **0.534** | **0.537** | **0.256** | **0.153** | **0.235** |

## A.2. Convergence

We analyze training dynamics on CMNIST using learning curves for CoDID and variants that consider hidden correlations. Figure 8(a) and (b) plot clustering metrics for mode discovery and the prediction loss of $a_1$ for disentangled representation learning, respectively. The following tendencies can be observed:

(1) Figure 8(a) shows that only CoDID and CoDID-UA improve and converge over iterations, while iterative variants without subcluster refinement degrade, indicating that subcluster-refined cluster splits are key to improving mode discovery.

(2) Figure 8(b) shows that iterative methods exhibit abrupt changes due to cluster refinement at certain iterations. CoDID shows the smoothest trajectory toward the end of training, which is more stable than the non-iterative CoDID-km and indicates superior convergence.

We attribute the convergence of CoDID to the mutual enhancement between disentanglement and mode discovery, which drives gradual improvement and thus steady convergence, whereas other methods accumulate irreducible errors that cause the two modules to interfere with each other, hindering convergence.

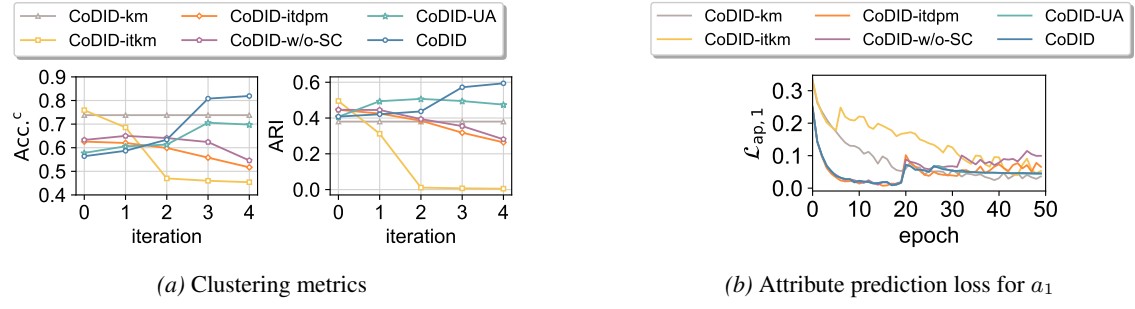

*(a)* Clustering metrics        *(b)* Attribute prediction loss for $a_1$

*Figure 8.* Learning curves on CMNIST.

## A.3. The Number of Clusters

We summarize the number of clusters for CoDID and variants in Table 4. For CoDID-km and CoDID-itkm with parametric k-means, we set the number of clusters to the ground-truth on CMNIST and CFashion-MNIST, and use hyperparameter

tuning on datasets with unknown modes. For non-parametric DPGMM-based methods, the number of clusters is inferred from data. The results show that: On CMNIST, CFashion-MNIST, and Canine-BG, the inferred mode count of CoDID is closest to the ground truth, indicating effective mode discovery. Other methods generally overestimate the number of modes, likely due to the lack of compactness in intra-mode representation distributions.

*Table 4.* Comparison with variants on the total number of modes (mean±std.).

| Method | Dataset | | | | | | |
|---|---|---|---|---|---|---|---|
| | CMNIST | CFashion-MNIST | Canine-BG | UCI-HAR | RealWorld | HHAR | MFD |
| Ground-truth | 5 | 7 | 20 | - | - | - | - |
| **Parametric clustering** (pre-specified with ground-truth value or hyperparameter tuning) | | | | | | | |
| CoDID-km | 5 | 7 | 20 | 48 | 24 | 12 | 6 |
| CoDID-itkm | 5 | 7 | 20 | 48 | 24 | 12 | 6 |
| **Non-parametric clustering** (inferred from data, reporting mean±std.) | | | | | | | |
| CoDID-itdpm | 7.8±1.6 | 8.0±1.0 | 25.9±3.2 | 54.4±3.6 | 25.0±2.8 | 32.3±0.5 | 13.3±1.2 |
| CoDID-w/o-SC | 8.0±1.6 | 9.4±0.9 | 26.5±3.1 | 58.6±4.3 | 20.8±2.0 | 29.7±1.7 | 12.7±0.9 |
| CoDID-UA | 7.4±1.5 | 11.4±0.9 | 16.4±2.7 | 55.2±3.9 | 15.0±1.2 | 31.3±0.5 | 14.0±1.4 |
| CoDID | 5.8±1.9 | 8.0±1.7 | 22.4±2.8 | 51.0±3.2 | 15.0±3.0 | 32.7±1.7 | 13.0±1.4 |

## A.4. Disentanglement Evaluation under Hidden Correlations

We train on correlated data and evaluate MI and DCI-I (Eastwood & Williams, 2018) on uncorrelated test sets, following (Funke et al., 2022). The results are reported in Table 5. We observe that methods with non-mode-conditioned independence constraints achieve low MI but exhibit high DCI-I due to the loss of mode information under hidden correlations. In contrast, methods that avoid improper independence constraints (CODA, HFS, DIOSC) yield lower DCI-I by preserving informativeness, but exhibit higher MI for retaining redundant information. CoDID attains the lowest DCI-I while keeping a relatively low MI, demonstrating disentanglement by striking a better balance between enforcing independence and preserving informativeness, likely due to the mode-conditioned independence constraint and the coordination of mode discovery and disentanglement.

*Table 5.* Disentanglement metrics.

| Method | MNIST | | CFashion-MNIST | | Canine-BG | |
|---|---|---|---|---|---|---|
| | MI $\downarrow$ | DCI-I $\downarrow$ | MI $\downarrow$ | DCI-I $\downarrow$ | MI $\downarrow$ | DCI-I $\downarrow$ |
| BASE | 0.406 | 0.119 | 0.599 | 0.133 | 0.344 | **0.296** |
| MMD | **0.160** | 0.345 | 0.256 | 0.213 | 0.387 | 0.464 |
| DTS | 0.236 | 0.369 | 0.276 | 0.268 | **0.224** | 0.501 |
| IDEVC | 0.230 | 0.381 | **0.200** | 0.246 | 0.278 | 0.507 |
| MI | 0.224 | 0.377 | 0.294 | 0.223 | 0.251 | 0.326 |
| A-CMI | 0.237 | 0.350 | 0.235 | 0.231 | 0.279 | 0.492 |
| HFS | 0.880 | 0.211 | 0.882 | 0.200 | 0.845 | 0.339 |
| CODA | 0.555 | 0.187 | 0.843 | 0.163 | 0.413 | 0.500 |
| ID-FaceVC | 0.341 | 0.268 | 0.336 | 0.215 | 0.386 | 0.486 |
| DIOSC | 0.560 | 0.169 | 0.851 | 0.162 | 0.548 | 0.495 |
| CoDID | 0.297 | **0.101** | 0.219 | **0.106** | 0.267 | 0.317 |

## A.5. Visualization of Representation Distributions and Time Series Modes

**Representation distributions for OOD tasks with unknown modes.** We use t-SNE visualization to examine the training and test representation distribution on MFD dataset, calculate the silhouette score (Sil.) to evaluate the cluster structure, and compute the $\mathcal{A}$-distance ($d_{\mathcal{A}}$) to measure the discrepancy between training and test representation distributions. Figure 9 shows that:

(1) For CoDID-km and CoDID-itkm, training representations are separated in several clusters with relatively low silhouette scores, and training and test representations lie in disjoint regions with large $d_{\mathcal{A}}$, indicating that the discovered modes are

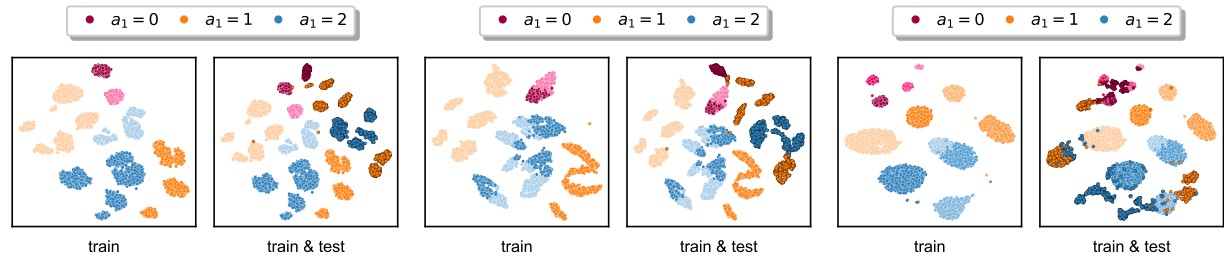

*(a)* CoDID-km (Sil. = 0.574, $d_{\mathcal{A}}$ = 1.99)    *(b)* CoDID-itkm (Sil. = 0.401, $d_{\mathcal{A}}$ = 1.97)    *(c)* CoDID (Sil. = 0.629, $d_{\mathcal{A}}$ = 1.78)

*Figure 9.* Machine fault representation distributions on MFD. Different colors, i.e., red, orange, and blue, indicate different machine fault types $a_1$, i.e., healthy, inner-bearing damage, and outer-bearing damage, respectively. Different shades of a color indicate different modes under the $a_1$ value. Points with white edges indicate training data, and points with black edges indicate test data.

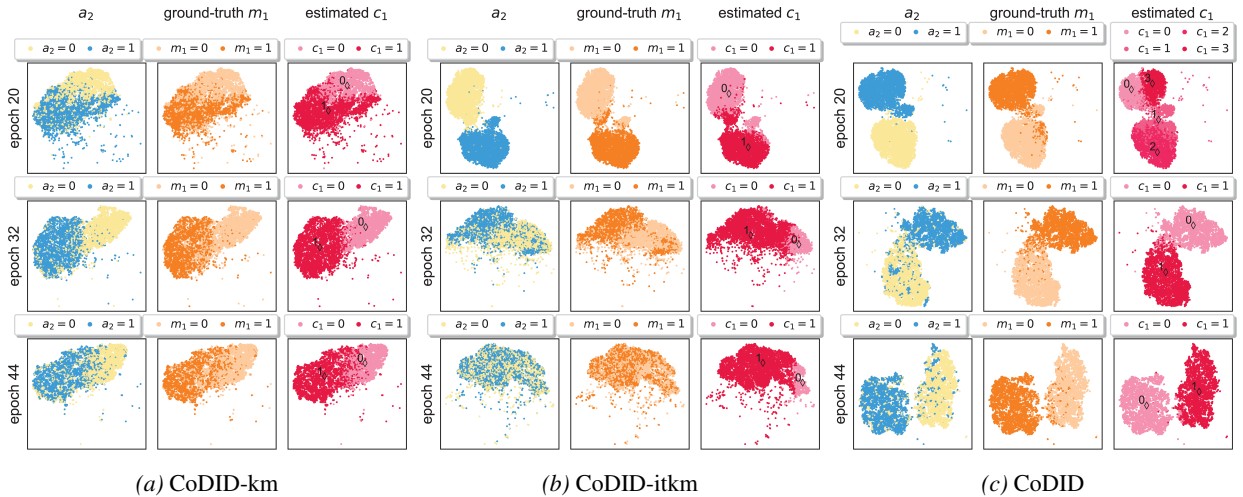

*(a)* CoDID-km    *(b)* CoDID-itkm    *(c)* CoDID

*Figure 10.* Digit representation distributions of $z_1$ for the two modes (digit 3 and 9) under $a_1 = 1$ (parity "odd") on CMNIST during training. CoDID-itkm and CoDID complete pre-training and conduct initial clustering at epoch 20. CoDID-km is trained from scratch at epoch 0 using mode labels estimated by a pre-trained BASE model. Colors indicate labels $a_2$, $m_1$, or $c_1$. Black-edged diamond markers with numbers indicate the cluster centroids and their indices.

misaligned with the data and fail to generalize, probably due to the inaccurate initial mode discovery and error amplification.

(2) In contrast, CoDID discovers compact mode structures on the training set with the highest silhouette score, and train and test representations largely overlap with the smallest $d_{\mathcal{A}}$, suggesting that the discovered modes well reflect the data distribution structure and generalize across distribution shifts. This is probably due to the flexible, data-driven mode discovery module and the effective mechanism for preventing error amplification.

**Evolution of Representation Distributions with Ground-Truth Mode Labels.** We additionally visualize the training dynamics of representation distributions for the variants of CoDID, as an extension of Figure 6. Figure 10 shows that:

(1) Although the ground-truth number of modes is provided, CoDID-km and CoDID-itkm representations of different modes become increasingly mixed during training (columns 2, 3 of Figure 10(a)(b)), indicating loss of mode information and mode estimation errors. CoDID-km suffers from the constant clustering errors on pre-trained representations, and CoDID-itkm is affected by the amplified clustering errors over iterations, hindering disentanglement and causing loss of mode information.

(2) The representations of different $a_2$ values are clearly separable for CoDID and CoDID-itkm at epoch 20 (column 1 of Figure 10(a)(c)), indicating the over-encoding of $a_2$ in representations $z_1$ at the end of pre-training. As training proceeds, $a_2$ values become harder to distinguish for all methods, indicating the removal of the information of $a_2$. In contrast to CoDID, CoDID-km and CoDID-itkm remove too much information of $a_2$, including the part shared with $m_1$, which disrupts the mode-specific conditional distribution and corrupts the mode information in $z_1$.

**Visualizations of the Discovered Modes.** We visualize the data of estimated modes on the training set of RealWorld. The results in Figure 11 show that the signals of the three walking modes differ in mean values and volatility, possibly due to

varying paces, strides, and postures in the walking activity. This justifies the presence of underlying modes within complex time series data.

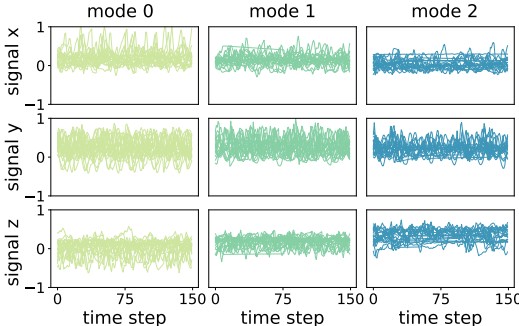

*Figure 11.* 3-channel accelerometer signals of three walking modes (20 random samples per mode with xyz channels). The x-axis indicates time steps, and the y-axis indicates normalized signals.

### A.6. Robustness Under Varying Noise Levels and Correlation Strengths

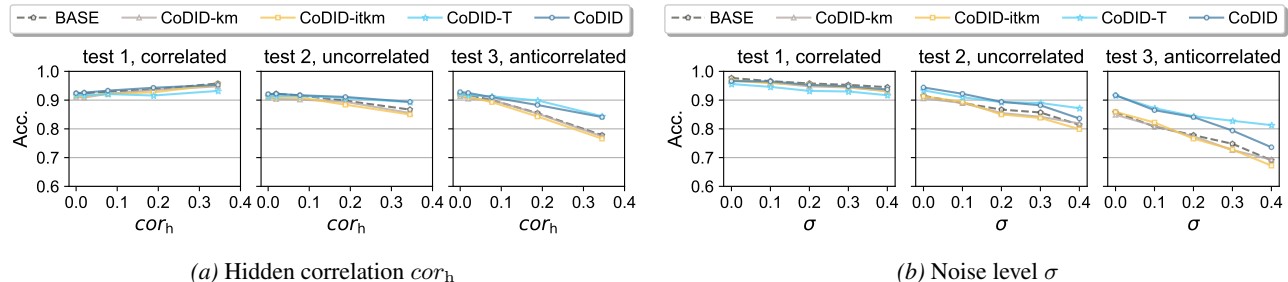

*(a)* Hidden correlation $cor_h$                    *(b)* Noise level $\sigma$

*Figure 12.* Comparison under varying correlation strengths and noise levels on CFashion-MNIST.

We compare different methods under varying train correlation strengths $cor_h = I(m_1; a_2|a_1)$ and noise levels $\sigma$ on CFashion-MNIST to investigate method robustness, as shown in Figure 12. For reference, we compare with BASE with only supervised learning, and CoDID-T, which uses the *ground-truth mode labels* for disentanglement. The following tendencies can be observed:

(1) Overall, the advantage of CoDID over CoDID-km and CoDID-itkm becomes more pronounced as the train–test correlation shift increases. This is reflected by the increasing performance gain across test sets, i.e., from test 1 to test 3 with increased correlation shift from the training set, and across correlation strengths, i.e., as $cor_h$ increases and induces larger correlation shifts on test 2, 3 in Figure 12(a). This indicates that CoDID achieves disentanglement better, being more robust under large correlation shifts.

(2) The advantage of CoDID over CoDID-km and CoDID-itkm remains stable as noise level increases, as shown in Figure 12(b), test 2, 3. Noises hinder mode discovery, making the initial clustering more error-prone and causing error amplification during iterations. These problems might cause the variants to only perform comparably to the BASE method, but could be mitigated by our meta-coordination mechanism, making CoDID more robust under large noises.

(3) CoDID performs comparably to CoDID-T with the ground-truth mode labels in most cases, which demonstrates the superiority of CoDID as a data-driven method without relying on prior knowledge. In some cases, CoDID outperforms CoDID-T, probably because the self-supervised learning with reconstruction tasks enhances the generalization ability. CoDID underperforms CoDID-T at high noise levels ($\sigma \geq 0.3$), where modes are unidentifiable and clustering is infeasible. This gap reflects an intrinsic limitation of the data that constrains all clustering methods.

### A.7. Hyperparameter Sensitivity

We analyze the hyperparameter sensitivity of the initial number of clusters $K^c$, and coefficients $\lambda_{mr}$ of the reconstruction loss during meta-optimization, $\lambda_{mp}$ of the mode prediction loss, $\lambda_{re}$ of the reconstruction loss, and $\lambda_{wa}$ of the weight alignment loss on CFashion-MNIST. The accuracy on test 3 is reported in Figure 13. We observe that:

(1) In Figure 13(a), the performance is stable with respect to $K^c$. This indicates that CoDID is robust to the initial choice of the number of clusters and can converge over a wide range of $K^c$. The slightly lower performance at $K^c = 2$ is probably because a small number of clusters increases the chance that true modes are mixed within a single cluster, which raises within-cluster correlations.

(2) Figures 13(b)-(e) show that performance peaks at intermediate values of the loss coefficients. This suggests that each loss term plays a meaningful role during learning, and choosing appropriate coefficients enhances DRL performance.

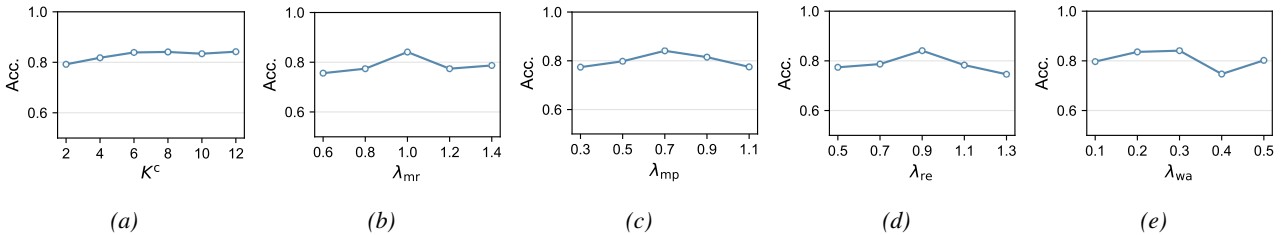

*Figure 13.* Hyperparameter sensitivity analysis.

## A.8. Experiments on Multiple Attributes with Hidden Correlations

We conduct experiments on a multi-attribute toy dataset to validate the effectiveness of CoDID in a more complex scenario.

**Data Construction.** We construct 4-dimensional toy data with 4 attributes ($a_i, 1 \leq i \leq 4$), where 2 attributes ($a_1, a_2$) exhibit underlying modes. This setting allows us to observe the impact on complex attributes with multi-modal distributions and simple attributes with uni-modal distributions. This dataset extends the simple toy dataset used in the main paper. Similarly, each data axis is controlled by one attribute, i.e., $x_i$ is affected by $a_i$ and unaffected by other attributes $a_{-i}$. Here, $a_1$ and $a_2$ with underlying modes are constructed the same as the $a_1$ in the simple toy data, with 3 modes under each attribute value; $a_3$ and $a_4$ are constructed the same as the $a_2$ in the simple toy data. The mappings from attribute/mode labels to the corresponding data axis remain the same.

*Table 6.* Prediction accuracy on toy data with multiple attributes.

| **BASE** | $a_1$ | $a_2$ | $a_3$ | $a_4$ |
|---|---|---|---|---|
| test 1 | 0.965 | 0.871 | 0.999 | 0.999 |
| test 2 | 0.757 | 0.697 | 0.997 | 0.998 |
| test 3 | 0.539 | 0.585 | 0.997 | 0.994 |
| **A-CMI** | $a_1$ | $a_2$ | $a_3$ | $a_4$ |
| test 1 | 0.510 | 0.593 | 0.998 | 0.525 |
| test 2 | 0.507 | 0.537 | 0.997 | 0.507 |
| test 3 | 0.507 | 0.518 | 0.996 | 0.476 |
| **CoDID-km** | $a_1$ | $a_2$ | $a_3$ | $a_4$ |
| test 1 | 0.994 | 0.797 | 0.998 | 0.999 |
| test 2 | 0.870 | 0.740 | 0.998 | 1.000 |
| test 3 | 0.752 | 0.741 | 0.997 | 0.999 |
| **CoDID** | $a_1$ | $a_2$ | $a_3$ | $a_4$ |
| test 1 | 0.994 | 0.997 | 0.996 | 0.999 |
| test 2 | 0.930 | 0.940 | 0.997 | 0.999 |
| test 3 | 0.872 | 0.881 | 0.997 | 0.999 |
| **CoDID-T** | $a_1$ | $a_2$ | $a_3$ | $a_4$ |
| test 1 | 0.999 | 0.997 | 0.997 | 1.000 |
| test 2 | 0.966 | 0.982 | 0.996 | 1.000 |
| test 3 | 0.928 | 0.964 | 0.998 | 0.999 |

**Experiment Settings.** We use the same settings as the simple toy data, training on correlated data and evaluating on three test sets: test 1 with the same correlations, test 2 without correlations, and test 3 with anticorrelations. Complex attributes and hidden correlations are introduced in the training data, e.g., $I(a_2; a_4) = 0.07, I(a_3; a_4) = 0.13, I(m_1; a_2|a_1) =$

$0.14, I(m_1; a_4|a_1) = 0.36, I(m_2; a_3|a_2) = 0.28$. The task is to learn disentangled representations for each attribute. We compare CoDID with BASE, A-CMI, CoDID-km, and CoDID-T. For CoDID(-km/-T), we use mode-based CMI minimization for $a_1$ and $a_2$ with underlying modes, and attribute-based CMI minimization for $a_3$ and $a_4$. For A-CMI, attribute-based CMI minimization is used for all attributes.

**Results and Discussions.** The attribute prediction accuracy is reported in Table 6, showing that:

(1) **A-CMI** performs poorly on $a_4$, even though $a_4$ does not exhibit underlying modes and is easily predicted by the BASE method. This is because, under hidden correlations $I(m_1; a_4|a_1)$, minimizing attribute-based CMI for $a_1$ might degrade the representation quality for the correlated $a_4$ as well, as indicated by Proposition 2.

(2) **CoDID-T** outperforms CoDID by a larger margin, likely because: The increased complexity with more attributes makes mode discovery harder, resulting in more errors in mode-based CMI and mode prediction losses. Reduced Informativeness in one representation may impact the disentanglement of others. Thus, mode estimation errors for $a_1$ and $a_2$ not only harm the quality of their own representations, but also affect the representations of each other when they are jointly disentangled.

(3) Still, **CoDID** generally shows superiority compared to BASE, A-CMI, and CoDID-km. For attributes $a_1, a_2$ with underlying modes, CoDID explicitly accounts for hidden correlations by discovering and leveraging modes, thus better preserving mode information. For the simple attributes $a_3, a_4$, considering their hidden correlations with the modes of other attributes, this also aids their disentanglement during the joint learning process. Due to the meta-learned weights, clustering errors would not harm representation informativeness, and multi-attribute joint disentanglement can be gradually achieved by removing redundant information while preserving mode information.

## B. Data Generation Process, Assumptions, and Scope of Applicability

**Data Generation Process.** We assume data are generated according to the causal process in Definition 1 (Figure 14) based on three key assumptions, as listed below.

**Definition 1.** (Disentangled Causal Process). *Consider a causal generative model $p(\boldsymbol{x}|\boldsymbol{a})$ for data $\boldsymbol{x}$ with $Q$ attributes $\boldsymbol{a} = (a_1, a_2, ..., a_Q)$. A certain attribute $a_q$ is associated with a categorical mode variable $m_q$. Attributes $\boldsymbol{a}$ are influenced by $L$ confounders $\boldsymbol{c}^{\mathrm{a}} = (c_1^{\mathrm{a}}, ..., c_L^{\mathrm{a}})$. Conditioned on $a_q$, mode variable $m_q$ and other attributes $\boldsymbol{a}_{-q}$ are influenced by $Q$ confounders $\boldsymbol{c}^{\mathrm{m}} = (c_1^{\mathrm{m}}, ..., c_Q^{\mathrm{m}})$. This causal model is called disentangled if and only if it follows a structural causal model (SCM) (Pearl, 2009) of the form:*

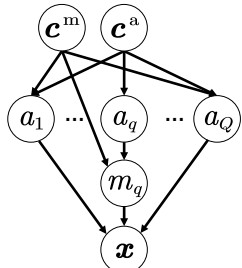

$$\boldsymbol{c}^{\mathrm{a}} \leftarrow \boldsymbol{n}^{\mathrm{ca}}, \boldsymbol{c}^{\mathrm{m}} \leftarrow \boldsymbol{n}^{\mathrm{cm}}$$
$$a_j \leftarrow h_j^{\mathrm{a}}(S_j^{\mathrm{a}}, S_j^{\mathrm{m}}, \boldsymbol{n}_j^{\mathrm{a}}), S_j^{\mathrm{a}} \subset \{c_1^{\mathrm{a}}, ..., c_L^{\mathrm{a}}\}, S_j^{\mathrm{m}} \subset \{c_1^{\mathrm{m}}, ..., c_Q^{\mathrm{m}}\}, j \neq q$$
$$a_q \leftarrow h_q^{\mathrm{a}}(S_q^{\mathrm{a}}, \boldsymbol{n}_q^{\mathrm{a}}), S_q^{\mathrm{a}} \subset \{c_1^{\mathrm{a}}, ..., c_L^{\mathrm{a}}\}, q \in \{1, ..., Q\} \qquad (12)$$
$$m_q \leftarrow h^{\mathrm{m}}(a_q, \boldsymbol{c}^{\mathrm{m}}, \boldsymbol{n}^{\mathrm{m}})$$
$$\boldsymbol{x} \leftarrow g(\boldsymbol{a}_{-q}, m_q, \boldsymbol{n}^{\mathrm{x}})$$

*Figure 14.* Causal graph of data generation under hidden correlations regarding a certain $a_q$.

*with functions $g, h_i^{\mathrm{a}}, h^{\mathrm{m}}$, jointly independent noises $\boldsymbol{n}^{\mathrm{ca}}, \boldsymbol{n}^{\mathrm{cm}}, \boldsymbol{n}_i^{\mathrm{a}}, \boldsymbol{n}^{\mathrm{m}}, \boldsymbol{n}^{\mathrm{x}}$, and confounder subsets $S_i^{\mathrm{a}}, S_j^{\mathrm{m}}$, for $i = 1, ..., Q, j = 1, ..., Q, j \neq q$. $-q$ denotes the set of attribute indices $\{j\}_{j \neq q}$.*

**Key Assumptions.** Among the three key assumptions below, the first is a standard assumption in DRL that must strictly hold (Suter et al., 2019; Wang & Jordan, 2024), while the others are specific to our method but can be relaxed: ❶ Each attribute is an *elementary ingredient* that has no causal effect on other attributes (Suter et al., 2019), i.e., interventions on one attribute do not influence others. ❷ For some value $\alpha$ of attribute $a_q$, $p(\boldsymbol{x}|a_q = \alpha)$ might be a *multi-modal distribution*. ❸ Correlations may arise from two confounder sets: $\boldsymbol{c}^{\mathrm{a}}$ induces *attribute correlations* $I(a_i; a_{i'}), i \neq i'$; $\boldsymbol{c}^{\mathrm{m}}$ induces *hidden correlations* $I(m_q; a_{-q}|a_q)$.

**Scope of Applicability.** Our theoretical results are based on the data generation process of Definition 1, relying on the causal structure (Assumption ❶, attributes as elementary ingredients) and not restricted to specific functional forms or parameterizations. They naturally extend to (1) *uni-modal distributions with attribute correlations*, where only one mode exists under each attribute value (relaxation of Assumption ❷), and mode-based CMI degrades to attribute-based CMI; and (2) *uncorrelated data* as correlation strengths vanish (relaxation of Assumption ❸). Although our results strictly rely on the elementary-ingredient assumption, they extend to arbitrary parameterizations, numbers of attributes/modes, and correlation

types/strengths.

## C. Propositions and Proofs about Mode-based CMI Minimization

We give the theoretical results using knowledge of mutual information and causal graphs. Note that we use formal definitions of mutual information, where separators **semicolon ";"** and **comma ","** should be distinguished from each other. Semicolon ";" separates groups of variables whose mutual information with respect to each other is being measured, while comma "," denotes the joint distribution of the listed variables.

### C.1. Proposition 2: A-CMI Fails Under Hidden Correlations

We show that enforcing attribute-based conditional independence (A-CMI), $I(z_1; z_2|a_1) = 0$, could hurt the predictive ability of representations, which is formalized in Proposition 2 as follows:

**Proposition 2.** *If $I(m_1; a_2|a_1) > 0$, then enforcing $I(z_1; z_2|a_1) = 0$ leads to at least one of $I(z_1; m_1) < H(m_1)$ and $I(z_2; a_2) < H(a_2)$.*

where $H(\cdot)$ denotes entropy, and the MI $I(\cdot; \cdot)$ between a representation and an attribute measures the amount of information the representation contains about the attribute. $I(z_1; m_1) < H(m_1)$ indicates that $z_1$ loses mode information about $m_1$, which is important for predicting $a_1$, while $I(z_2; a_2) < H(a_2)$ indicates that $z_2$ loses attribute information for predicting $a_2$. Thus, minimizing attribute-based CMI hurts the predictive ability of representations under hidden correlations.

**Relation to prior works.** This is an extension of Proposition 3.1 in (Funke et al., 2022), which proves that minimizing unconditional MI harms representation informativeness under attribute correlations. In short, unconditional MI minimization fails under attribute correlations. These results convey a common message: when attributes or modes are correlated, enforcing independence between the associated representations degrades representation informativeness.

**Proof.** We prove by contradiction. Assuming $I(z_1; m_1) = H(m_1)$ and $I(a_2; z_2) = H(a_2)$ both stand, we have $H(m_1|z_1) = 0$ and $H(a_2|z_2) = 0$.

Firstly, we prove that this leads to $I(m_1; a_2; z_1; z_2|a_1) > 0$ with (1)(2)(3).

(1) Since $H(m_1|z_1) = 0$ and $H(m_1|z_1) - H(m_1|a_1, z_1) = I(m_1; a_1|z_1) \geq 0$ by definition of conditional mutual information, we have $0 \leq H(m_1|a_1, z_1) \leq H(m_1|z_1) = 0$, we have $H(m_1|a_1, z_1) = 0$. By definition, $H(m_1|a_1, z_1) = H(m_1|a_1, a_2, z_1) + I(m_1; a_2|a_1, z_1) = 0$, which gives $I(m_1; a_2|a_1, z_1) = 0$, as both terms are non-negative. Therefore:

$$I(m_1; a_2; z_1|a_1) = I(m_1; a_2|a_1) - I(m_1; a_2|a_1, z_1)$$
$$= I(m_1; a_2|a_1) > 0$$

(2) Similar to (1), since $H(a_2|z_2) = 0$ and $0 \leq H(a_2|a_1, z_2) \leq H(a_2|z_2) = 0$, we have $H(a_2|a_1, z_2) = 0$. By definition, $H(a_2|a_1, z_2) = H(a_2|m_1, a_1, z_2) + I(m1; a2|a_1, z_2) = 0$, which gives $I(m_1; a_2|a_1, z_2) = 0$, as both terms are non-negative. Therefore:

$$I(m_1; a_2; z_2|a_1) = I(m_1; a_2|a_1) - I(m_1; a_2|a_1, z_2)$$
$$= I(m_1; a_2|a_1) > 0$$

(3) Given $H(m_1|z_1) = 0$, we have $H(m_1|z_1) = H(m_1|z_1, z_2) + I(m_1; z_2|z_1) = 0$ and thus $H(m_1|z_1, z_2) = 0$, as both terms are non-negative. Similar to (1) that yields $I(m_1; a_2; z_1|a_1) = I(m_1; a_2|a_1)$ from $H(m_1|z_1) = 0$, we can get $I(m_1; a_2; z_1|a_1, z_2) = I(m_1; a_2|a_1, z_2)$ from $H(m_1|z_1, z_2) = 0$ by additionally conditioning on $z_2$. Combined with $I(m_1; a_2; z_2|a_1) > 0$ in (2), we have:

$$I(m_1; a_2; z_1; z_2|a_1) = I(m_1; a_2; z_1|a_1) - I(m_1; a_2; z_1|a_1, z_2)$$
$$= I(m_1; a_2|a_1) - I(m_1; a_2|a_1, z_2)$$
$$= I(m_1; a_2; z_2|a_1) > 0$$

Secondly, we prove $I(m_1; a_2; z_1; z_2|a_1) \leq 0$ with (4)(5)(6).

(4) Given $H(m_1|a_1, \mathbf{z}_1) = 0$ in (1), we have $H(m_1|a_1, \mathbf{z}_1) = H(m_1|a_1, \mathbf{z}_1, \mathbf{z}_2) + I(m_1; \mathbf{z}_2|a_1, \mathbf{z}_1) = 0$ and followingly, $I(m_1; \mathbf{z}_2|a_1, \mathbf{z}_1) = 0$, as both terms are non-negative. Therefore:

$$I(m_1; \mathbf{z}_1; \mathbf{z}_2|a_1) = I(m_1; \mathbf{z}_2|a_1) - I(m_1; \mathbf{z}_2|a_1, \mathbf{z}_1)$$
$$= I(m_1; \mathbf{z}_2|a_1) \geq 0$$

(5) Since $I(\mathbf{z}_1; \mathbf{z}_2|a_1) = 0$, we have:

$$I(m_1; \mathbf{z}_1; \mathbf{z}_2|a_1) = I(\mathbf{z}_1; \mathbf{z}_2|a_1) - I(\mathbf{z}_1; \mathbf{z}_2|m_1, a_1)$$
$$= -I(\mathbf{z}_1; \mathbf{z}_2|m_1, a_1) \leq 0$$

(6) Combine $I(m_1; \mathbf{z}_1; \mathbf{z}_2|a_1) \geq 0$ in (4) and $I(m_1; \mathbf{z}_1; \mathbf{z}_2|a_1) \leq 0$ in (5), we have $I(m_1; \mathbf{z}_1; \mathbf{z}_2|a_1) = 0$. Given $H(m_1|a_1, \mathbf{z}_1) = 0$ in (1) and $H(m_1|a_1, \mathbf{z}_1) = H(m_1|a_1, \mathbf{z}_1, \mathbf{z}_2) + I(m_1; \mathbf{z}_2|a_1, \mathbf{z}_1)$, we have $H(m_1|a_1, \mathbf{z}_1, \mathbf{z}_2) = 0$ as both terms are non-negative. Similar to (4) that yields $I(m_1; \mathbf{z}_1; \mathbf{z}_2|a_1) = I(m_1; \mathbf{z}_2|a_1)$ from $H(m_1|a_1, \mathbf{z}_1) = 0$, we can get $I(m_1; \mathbf{z}_1; \mathbf{z}_2|a_1, a_2) = I(m_1; \mathbf{z}_2|a_1, a_2)$ from $H(m_1|a_1, \mathbf{z}_1, \mathbf{z}_2) = 0$ by additionally conditioning on $\mathbf{z}_2$. Therefore:

$$I(m_1; a_2; \mathbf{z}_1; \mathbf{z}_2|a_1) = I(m_1; \mathbf{z}_1; \mathbf{z}_2|a_1) - I(m_1; \mathbf{z}_1; \mathbf{z}_2|a_1, a_2)$$
$$= -I(m_1; \mathbf{z}_2|a_1, a_2) \leq 0$$

This is contradictory with $I(m_1; a_2; \mathbf{z}_1; \mathbf{z}_2|a_1) > 0$. Therefore, if $I(m_1; a_2|a_1) > 0$ and $I(\mathbf{z}_1; \mathbf{z}_2|a_1) = 0$, then at least one of $I(m_1; \mathbf{z}_1) < H(m_1)$ and $I(a_2; \mathbf{z}_2) < H(a_2)$ must hold.

### C.2. The Property of the True Latent Representations

We build the causal graphs of data generation with the true *latent* representations $\mathbf{z}_i^1, i = 1, 2$, where $\mathbf{c}^{\mathrm{a}}$ is the set of confounders that introduce attribute correlations between $a_1, a_2$, and $\mathbf{c}^{\mathrm{m}}$ is the set of confounders that introduce hidden correlations between $m_1, a_2$ given $a_1$. For example, on human activity data with activity attribute $a_1$, the latent $\mathbf{z}_1^1$ encodes the body movements that characterize activities, which are unaffected by changes in user behavior patterns, yet activity and user ID attributes may be correlated due to confounding. Since the disentangled $\mathbf{z}_i$ aims to recover the true latent $\mathbf{z}_i^1$, the learned representations should retain the properties of the true latent representations. We show that the true *latent* representations satisfy mode-based conditional independence, making it a necessary condition for disentanglement, and thus a proper independence constraint.

According to causal graph theorems (Pearl, 2009), two variables $X, Y$ are conditionally independent given a variable that blocks all *backdoor paths* between them, i.e., the paths that flow backward from $X$ or $Y$. In Figure 15(a), we consider only attribute correlations as A-CMI, where $a_1$ blocks the only *backdoor path* between $\mathbf{z}_1^1$ and $\mathbf{z}_2^1$. In comparison, we consider additional hidden correlations in Figure 15(b), where $m_1$ blocks all *backdoor paths*, yet $a_1$ fails to block the path via $\mathbf{c}^{\mathrm{m}}$ (consistent with the failure of A-CMI under hidden correlations). Thus, under hidden correlations and attribute correlations, the true latent representations are conditionally independent based on $m_1$, a property the learned disentangled representations must retain:

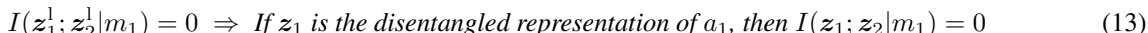

$$I(\mathbf{z}_1^1; \mathbf{z}_2^1|m_1) = 0 \implies \textit{If } \mathbf{z}_1 \textit{ is the disentangled representation of } a_1, \textit{ then } I(\mathbf{z}_1; \mathbf{z}_2|m_1) = 0 \tag{13}$$

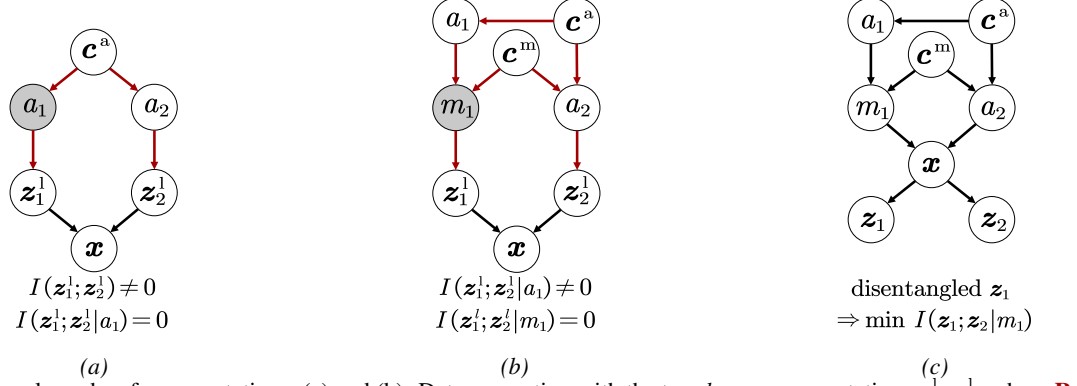

*(a)*        *(b)*        *(c)*

*Figure 15.* Causal graphs of representations. (a) and (b): Data generation with the true *latent* representations $\mathbf{z}_1^1, \mathbf{z}_2^1$, where **Red** arrows indicate the *backdoor paths* between them. (c): Representation learning that produces the *learned* representations $\mathbf{z}_1, \mathbf{z}_2$.

# D. Propositions, Proofs, and Case Studies about the Impact of Clustering Errors

To thoroughly analyze the impact of clustering errors in the iterative framework, we expand the discussion in Section 3.2 to elaborate on the two types of clustering errors, and to explain why the second type does not require special attention in our iterative framework.

Formally, clustering errors can be quantified by the Variation of Information (VI) (Meilă, 2007) between the ground-truth modes $m_1$ and learned clusters $c_1$:

$$\text{VI}(m_1, c_1) = H(m_1|c_1) + H(c_1|m_1), \tag{14}$$

where $H(m_1|c_1)$ measures cluster impurity caused by **Type 1 errors** (mixing modes in one cluster) and $H(c_1|m_1)$ measures mode incompleteness caused by **Type 2 errors** (splitting a mode into several clusters). The two terms are complementary components of the overall VI (neither implies the other), and they are non-exclusive and might co-occur.

In our iterative framework, Type 1 errors induce within-cluster correlations that lead to error amplification (Proposition 1; discussed in Section 3.2; proof in Section D.1), while Type 2 errors can be mitigated through iterative refinement (Proposition 3 and case study in Section D.2). Table 7 provides a systematic comparison.

*Table 7.* The two types of clustering errors.

| Type 1 | |
| --- | --- |
| Description | Data from multiple modes are mixed into one cluster. |
| VI terms | $H(m_1 \mid c_1)$: reduced within-cluster purity. |
| Impact on subsequent DRL | Hinders disentanglement by hurting ***Informativeness***. |
| Impact over iterations | Induces error amplifications. |
| Solution | *Cluster split*: split mixed clusters. |
| Solution Realization | Not promoted by subsequent DRL; requires explicit intervention. |
| **Type 2** | |
| Description | Data from one mode are divided into multiple clusters. |
| VI terms | $H(c_1 \mid m_1)$: reduced mode completeness. |
| Impact on subsequent DRL | Promotes disentanglement by approaching but not fully achieving ***Independence***. |
| Impact over iterations | Facilitates mutual enhancement. |
| Solution | *Cluster merge*: merge clusters that originate from the same mode. |
| Solution Realization | Promoted by subsequent DRL; emerges spontaneously. |

## D.1. Proposition 1: The Negative Effects of Type 1 Clustering Error

**Proposition 1.** *If $I(m_1; a_2|c_1) > 0$, then enforcing $I(z_1; z_2|c_1) = 0$ leads to at least one of $I(z_1; m_1) < H(m_1)$ or $I(z_2; a_2) < H(a_2)$.*

***Proof.*** Proposition 1 can be proved by replacing $a_1$ with $c_1$ in Proposition 2 in Appendix C.1. The derivation is otherwise identical and is omitted for brevity.

## D.2. Non-Amplifying Type 2 Clustering Errors That Vanish Over Iterations

As discussed in Section 3.2, disentanglement relies on a proper independence constraint that achieves ***Independence*** while preserving ***Informativeness*** (Funke et al., 2022). Type 1 clustering error induces within-cluster correlations, thus hurts informativeness in DRL, and causes error amplification. In contrast, under only Type 2 clustering error, each cluster only contains the data from one mode, leaving no variation of $m_1$ within clusters, and within-cluster correlations regarding $m_1$ are naturally absent. Therefore, informativeness is not harmed, and errors will not be amplified.

The limitation under Type 2 clustering error is that *Independence* cannot be fully achieved, i.e., a part of redundant information about $a_2$ remains in representation $z_1$, as shown in Proposition 3 in Appendix D.2.1. We show that this redundancy can be gradually reduced over iterations via a case study in Section D.2.2.

### D.2.1. PROPOSITION 3: BENEFITS AND LIMITATIONS OF TYPE 2 CLUSTERING ERROR

We prove that, without Type 1 error, i.e., $H(m_1|c_1) = 0$, potential Type 2 error guides CMI minimization to promote disentanglement by approaching but not fully achieving *Independence*, as formalized in Proposition 3.

**Proposition 3.** If $H(m_1|c_1) = 0$, $I(z_1; c_1) = H(c_1)$, and $I(z_2; a_2) = H(a_2)$, then enforcing $I(z_1; z_2|c_1) = 0$ leads to $I(z_1; a_2|c_1) = 0$ and $I(z_1; a_2) = I(m_1; a_2) + I(c_1; a_2|m_1)$.

**Interpretations.** The conclusion $I(z_1; a_2|c_1) = 0$ indicates that, knowing the cluster label $c_1$, $z_1$ contains no additional information about $a_2$. Thus, minimizing the cluster-based CMI removes cluster-conditioned redundant information about $a_2$ and brings $z_1$ closer to disentanglement.

$I(z_1; a_2)$ should be able to account for hidden correlations $I(m_1; a_2)$, while we have $I(z_1; a_2) = I(m_1; a_2) + I(c_1; a_2|m_1)$. Here, $I(c_1; a_2|m_1)$ is the residual information about $a_2$ in $z_1$, which quantifies the redundancy induced by Type 2 clustering error: When a single mode $m_1 = k$ is split into multiple clusters, cross-cluster variations $c_1|m_1 = k$ within this mode may carry extra information about $a_2$, i.e., $I(c_1; a_2|m_1 = k)$. Since CMI minimization is applied within each cluster, it cannot eliminate this dependence across clusters, thereby preventing full disentanglement.

*Proof.* The two conclusions are reached with separate proofs as given below.

*(1) Proof of* $I(z_1; a_2|c_1) = 0$. To reach this conclusion, we first prove that $I(z_1; z_2|c_1) \geq I(z_1; a_2|c_1)$.

By definition of mutual information, we have $H(z_1|c_1, z_2) = H(z_1|c_1, a_2, z_2) + I(z_1; a_2|c_1, z_2)$. Then, we have $I(z_1; a_2|c_1, z_2) = H(a_2|z_2, c_1) - H(a_2|z_1, z_2, c_1) = 0 - 0 = 0$, which is based on $H(a_2|z_2) = H(a_2) - I(z_2; a_2) = 0$, $H(a_2|z_2) \geq H(a_2|z_2, c_1)$, and $H(a_2|z_2) \geq H(a_2|z_1, z_2, c_1)$. Therefore, we have (i) $H(z_1|c_1, z_2) = H(z_1|c_1, a_2, z_2)$.

By definition of mutual information, we have:

$$
\begin{aligned}
I(z_1; z_2|c_1) - I(z_1; a_2|c_1) &= [H(z_1|c_1) - H(z_1|c_1, z_2)] - [H(z_1|c_1) - H(z_1|c_1, a_2)] \\
&= H(z_1|c_1, a_2) - H(z_1|c_1, z_2) \\
&= H(z_1|c_1, a_2) - H(z_1|c_1, a_2, z_2) \quad \text{(based on (i))} \\
&\geq 0
\end{aligned}
$$

Therefore, we have $I(z_1; z_2|c_1) \geq I(z_1; a_2|c_1)$. By $I(z_1; z_2|c_1) = 0$, we conclude that $I(z_1; a_2|c_1) = 0$.

*(2) Proof of* $I(c_1; a_2) = I(m_1; a_2) + I(c_1; a_2|m_1)$. First, we prove that $I(z_1; a_2) = I(c_1; a_2)$ with (2.1-2.4), where (2.1)(2.2) proves $I(c_1; a_2) \geq I(z_1; a_2)$, (2.3) shows $I(z_1; a_2) \geq I(c_1; a_2)$, and (2.4) concludes the proof.

*(2.1)* Since $H(a_2|z_2) = 0$, we have $H(a_2|z_2) = H(a_2|z_1, z_2) + I(z_1; a_2|z_2) = 0$, and followingly $I(z_1; a_2|z_2) = 0$, as both terms are non-negative. Therefore, by definition of interaction information, we have $I(z_1; z_2; a_2) = I(z_1; a_2) - I(z_1; a_2|z_2) = I(z_1; a_2)$. Since $I(z_1; z_2|c_1) = 0$, we have $I(z_1; z_2; a_2|c_1) = I(z_1; z_2|c_1) - I(z_1; z_2|c_1, a_2) = -I(z_1; z_2|c_1, a_2)$. Therefore:

$$
\begin{aligned}
I(z_1; z_2; c_1; a_2) &= I(z_1; z_2; a_2) - I(z_1; z_2; a_2|c_1) \\
&= I(z_1; a_2) + I(z_1; z_2|c_1, a_2) \\
&\geq I(z_1; a_2)
\end{aligned}
$$

*(2.2)* i. Since $H(a_2|z_2) = 0$, we have $H(a_2|z_2) = H(a_2|c_1, z_2) + I(c_1; a_2|z_2) = 0$, and followingly $I(c_1; a_2|z_2) = 0$, as both terms are non-negative.

ii. Since $H(c_1|z_1) = 0$, we have $H(c_1|z_1) = H(c_1|z_1, z_2) + I(c_1; z_2|z_1) = 0$, and followingly $H(c_1|z_1, z_2) = 0$, as both terms are non-negative. Therefore, $H(c_1|z_1, z_2) = H(c_1|z_1, z_2, a_2) + I(c_1; a_2|z_1, z_2) = 0$, and followingly $I(c_1; a_2|z_1, z_2) = 0$, as both terms are non-negative.

iii. Given $I(c_1; a_2|z_2) = 0$ in i. and $I(c_1; a_2|z_1, z_2) = 0$ in ii. as shown above, we have $I(c_1; a_2; z_1|z_2) = I(c_1; a_2|z_2) - I(c_1; a_2|z_1, z_2) = 0$.

iv. Since $H(c_1|z_1) = 0$, by definition of conditional mutual information, we have $H(c_1|z_1) = H(c_1|z_1, a_2) + I(c_1; a_2|z_1) = 0$, and followingly $I(c_1; a_2|z_1) = 0$, as both terms are non-negative. Thus $I(c_1; a_2; z_1) = I(c_1; a_2) - I(c_1; a_2|z_1) = I(c_1; a_2)$.

Given $I(c_1; a_2; z_1) = I(c_1; a_2)$ in iv. and $I(c_1; a_2; z_1|z_2) = 0$ in iii., we have:

$$
\begin{aligned}
I(z_1; z_2; c_1; a_2) &= I(c_1; a_2; z_1) - I(c_1; a_2; z_1|z_2) \\
&= I(c_1; a_2)
\end{aligned}
$$

Given (2.1)(2.2), we have $I(c_1; a_2) = I(\boldsymbol{z}_1; \boldsymbol{z}_2; c_1; a_2) \geq I(\boldsymbol{z}_1; a_2)$

**(2.3)** We prove $I(\boldsymbol{z}_1; a_2) \geq I(c_1; a_2)$ as follows.

i. Since $H(c_1|\boldsymbol{z}_1) = 0$, we have $H(c_1|\boldsymbol{z}_1) = H(c_1|\boldsymbol{z}_1, a_2) + I(c_1; a_2|\boldsymbol{z}_1) = 0$, and followingly $I(c_1; a_2|\boldsymbol{z}_1) = 0$, as both terms are non-negative. Thus, by chain rule of mutual information, we have:

$$I(c_1, \boldsymbol{z}_1; a_2) = I(\boldsymbol{z}_1; a_2) + I(c_1; a_2|\boldsymbol{z}_1)$$
$$= I(\boldsymbol{z}_1; a_2)$$

ii. We also have:

$$I(c_1, \boldsymbol{z}_1; a_2) = I(c_1; a_2) + I(\boldsymbol{z}_1; a_2|c_1)$$
$$\geq I(c_1; a_2)$$

Given $I(c_1, \boldsymbol{z}_1; a_2) = I(\boldsymbol{z}_1; a_2)$ in i. and , $I(c_1, \boldsymbol{z}_1; a_2) \geq I(c_1; a_2)$ in ii., we have $I(\boldsymbol{z}_1; a_2) \geq I(c_1; a_2)$.

**(2.4)** Finally, given $I(c_1; a_2) \geq I(\boldsymbol{z}_1; a_2)$ with (2.1)(2.2) and $I(\boldsymbol{z}_1; a_2) \geq I(c_1; a_2)$ in (2.3), the equality must hold that $I(\boldsymbol{z}_1; a_2) = I(c_1; a_2)$.

Followingly, by definition of mutual information, we have $H(a_2|c_1) = I(a_2; m_1|c_1) + H(a_2|c_1, m_1)$. Then, we have $I(a_2; m_1|c_1) = H(m_1|c_1) - H(m_1|a_2, c_1) = 0 - 0 = 0$, based on $H(m_1|c_1) = 0$ and $H(m_1|c_1) \geq H(m_1|a_2, c_1)$. Therefore, we have (iii) $H(a_2|c_1) = H(a_2|c_1, m_1)$.

By definition of mutual information, we have:

$$I(c_1; a_2) = H(a_2) - H(a_2|c_1)$$
$$= H(a_2) - H(a_2|c_1, m_1) \quad \text{(based on (iii))}$$
$$= H(a_2) - H(a_2|m_1) + H(a_2|m_1) - H(a_2|c_1, m_1)$$
$$= I(m_1; a_2) + I(c_1; a_2|m_1)$$

Finally, by (ii) $I(\boldsymbol{z}_1; a_2) = I(c_1; a_2)$ and $I(c_1; a_2) = I(m_1; a_2) + I(c_1; a_2|m_1)$, we have $I(\boldsymbol{z}_1; a_2) = I(m_1; a_2) + I(c_1; a_2|m_1)$.

### D.2.2. CASE STUDY: THE MITIGATION OF TYPE 2 CLUSTERING ERROR

To isolate the impact of Type 2 clustering error, we construct a toy dataset where the true latent representations for $a_1$ are well separated across different modes, so that the clustering method is unlikely to assign data from different modes to one cluster, i.e., avoiding Type 1 error.

**Toy data construction.** As shown in Figure 16, we construct two-dimensional toy data with two attributes. Attribute $a_1$ takes three values $0, 1, 2$ and exhibits a varying number of modes under each value. There are 2, 3, 4 modes under $a_1 = 0, 1, 2$, respectively, which are indexed from 0 to 8 in the order of $a_1 = 0 \sim 2$. Attribute $a_2$ takes two values $0, 1$. Data are generated by linearly mapping $(m_1, a_2)$ to $\mathbb{R}^2$ and adding Gaussian noise $\boldsymbol{n} \sim \mathcal{N}(\boldsymbol{0}, \sigma^2 \boldsymbol{I})$ with noise level $\sigma = 0.02$:

$$\boldsymbol{x} = \boldsymbol{m}_1 \cdot \boldsymbol{W}_{\mathrm{m}} + \boldsymbol{a}_2 \cdot \boldsymbol{W}_{\mathrm{a}} + \boldsymbol{n}, \ \boldsymbol{W}_m = \begin{bmatrix} 0 & 2 & 4 & 6 & 8 & 1 & 3 & 5 & 7 \\ 0 & 0 & 0 & 0 & 0 & 0 & 0 & 0 & 0 \end{bmatrix}, \ \boldsymbol{W}_a = \begin{bmatrix} 0 & 0 \\ 0 & 1 \end{bmatrix}. \tag{15}$$

where $\boldsymbol{m}_1$ and $\boldsymbol{a}_2$ are one-hot encodings of $m_1$ and $a_2$. The matrices $\boldsymbol{W}_{\mathrm{m}}$ and $\boldsymbol{W}_{\mathrm{a}}$ have all zero entries in different dimensions, so that $m_1$ and $a_2$ control data dimensions $x_1, x_2$, respectively.

We introduce hidden corelations between $m_1$ and $a_2$ by specifying the conditional distribution $p(a_2 \mid m_1)$ in matrix $\boldsymbol{P}_{a_2|m_1}$, where the $(i, j)$-th component corresponds to $p(a_2 = i \mid m_1 = j)$:

$$\boldsymbol{P}_{a_2|m_1} = \begin{bmatrix} 0.8 & 0.2 & 0.8 & 0.1 & 0.6 & 0.3 & 0.8 & 0.2 & 0.7 \\ 0.2 & 0.8 & 0.2 & 0.9 & 0.4 & 0.7 & 0.2 & 0.8 & 0.3 \end{bmatrix} \tag{16}$$

**Experimental results of a simple iterative framework.** We evaluate **CoDID-dpm** (the variant that uses DPGMM for iterative clustering without meta-coordination mechanism) on toy data. The results are reported in Table 8. The representation distribution of $a_1$ at different iterations are visualized with t-SNE in Figure 17. The following tendencies can be observed:

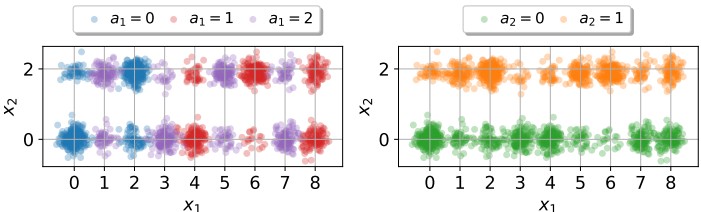

*Figure 16.* Toy data with various modes and hidden correlations under the values of attribute $a_1$.

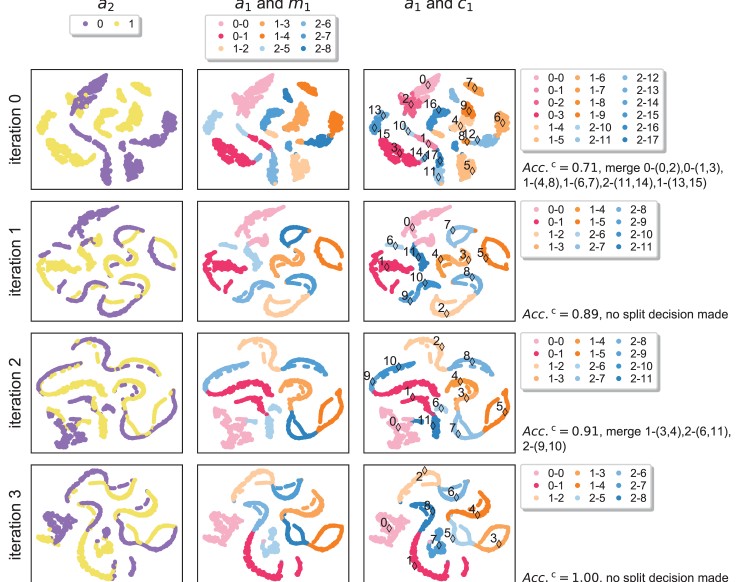

*Figure 17.* Representation distribution of attribute $a_1$ on toy data at clustering iterations. Different colors indicate different attribute values of $a_1$ and $a_2$. Different shades of a color indicate different modes or clusters under the $a_1$ value. "$\alpha - k$" indicates mode or cluster $k$ under $a_1$ value $\alpha$. "merge $\alpha - (k, l)$" indicates merging clusters $k, l$ under $a_1$ value $\alpha$ into a new cluster. Black-edged diamond markers with numbers indicate the cluster centroids and their indices.

(1) In Table 8, as train-test correlation shift enlarges from test 1 to test 3, BASE (with supervised learning only) exhibits performance degradation, due to entanglement under hidden correlations; CoDID-itdpm exhibits robust and high accuracies that are close to 1, indicating disentanglement. CoDID-itdpm also achieves a perfect clustering with all clustering metrics reaching their optimal. This demonstrates an ideal iterative process that achieves both disentanglement and mode discovery.

*Table 8.* Performance on toy data.

| Method | Attribute Prediction Acc. | | | Clustering metrics | | |
|---|---|---|---|---|---|---|
| | test 1 | test 2 | test 3 | Acc.$^c$ | ARI | NMI |
| BASE | 0.835 | 0.797 | 0.742 | – | – | – |
| CoDID-itdpm | **0.974** | **0.969** | **0.964** | **1.000** | **1.000** | **1.000** |

(2) Figure 17 shows that, at iteration 0 (the end of pre-training), representation $z_1$ of one mode may be scattered and assigned into two clusters with $a_2$ values $0, 1$, e.g., the two lightest blue clusters of mode 5 under $a_1 = 2$ in the left-middle area. This indicates over-encoding of $a_2$. As training proceeds, the representations of the same mode become closer, e.g., the two clusters of mode 5 move closer from iteration 0 to 2, get merged at iteration 2, and become more compact by iteration 3. This illustrates CMI-minimization under Type 2 error: Although *Independence* cannot be fully achieved at first, *Informativeness* can be preserved and redundancy about $a_2$ can be reduced, pulling clusters closer and enabling merges within modes. After merging, CMI minimization can fully remove redundancy and achieve disentanglement.

## E. Theoretical Extension to Multiple Attributes

Our theoretical results, including Proposition 1 in Section 3.2, Proposition 2 in Appendix C.1, the property of the true latent representations in Appendix C.2, and Proposition 3 in Appendix D.2.1 can be generalized to multiple attributes. The extension mainly involves replacing $m_1, z_1$ with $m_q, z_q$, and replacing $a_2, z_2$ with the joint $a_{-q}, z_{-q}$, as the properties of mutual information and causal graphs remain the same for joint variables.

**Proposition Extensions.** Corollaries 1.1, 2.1, 3.1 extend Propositions 1, 2, 3 to multiple attribues:

**Corollary 1.1** *If $I(m_q; a_{-q}|c_q) > 0$, then enforcing $I(z_q; z_{-q}|c_q) = 0$ leads to at least one of $I(z_q; m_q) < H(m_q)$ or $I(z_{-q}; a_{-q}) < H(a_{-q})$.*

**Corollary 2.1** *If $I(m_q; a_{-q}|a_q) > 0$, then enforcing $I(z_q; z_{-q}|a_q) = 0$ leads to at least one of $I(z_q; m_q) < H(m_q)$ and $I(z_{-q}; a_{-q}) < H(a_{-q})$.*

**Corollary 3.1** *If $H(m_q|c_q) = 0$, $I(z_q; c_q) = H(c_q)$, and $I(z_{-q}; a_{-q}) = H(a_{-q})$, then enforcing $I(z_q; z_{-q}|c_q) = 0$ leads to $I(z_q; a_{-q}|c_q) = 0$ and $I(z_q; a_{-q}) = I(m_q; a_{-q}) + I(c_q; a_{-q}|m_q)$.*

**The Extended Property of the True Latent Representations.** Figure 15(a)(b) with two attributes is extended to Figure 18(a)(b) with $K$ attributes, where the true latent representations satisfy the conditional independence as follows, yielding the necessary condition for disentanglement under hidden correlations and attribute correlations:

$$I(z_q^1; z_{-q}^1|m_q) = 0 \implies \textit{If } z_q \textit{ is the disentangled representation of } a_q, \textit{ then } I(z_q; z_{-q}|m_q) = 0 \tag{17}$$

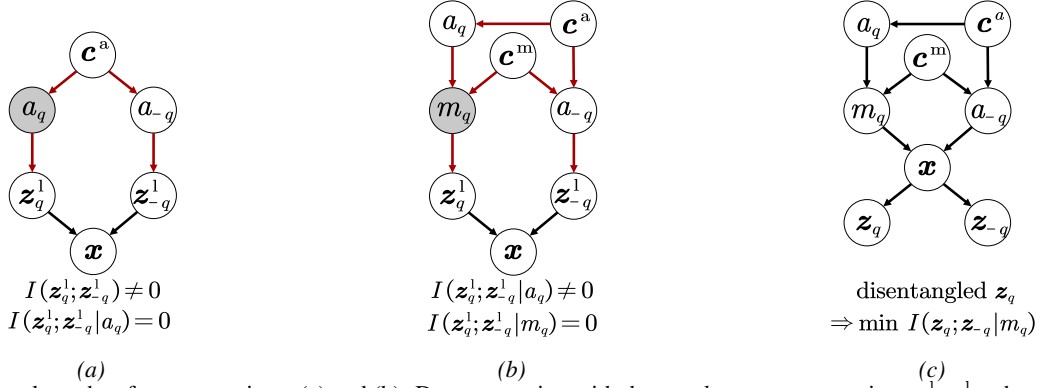

$$I(z_q^1; z_{-q}^1) \neq 0$$
$$I(z_q^1; z_{-q}^1|a_q) = 0$$

$$I(z_q^1; z_{-q}^1|a_q) \neq 0$$
$$I(z_q^1; z_{-q}^1|m_q) = 0$$

disentangled $z_q$
$$\implies \min I(z_q; z_{-q}|m_q)$$

*(a)*          *(b)*          *(c)*

*Figure 18.* Causal graphs of representations. (a) and (b): Data generation with the true *latent* representations $z_1^1, z_2^1$, where **Red** arrows indicate the *backdoor paths* between them. (c): Representation learning that produces the *learned* representations $z_1, z_2$.

# F. Experimental Setup and Implementation Details

## F.1. Datasets

The detailed information of the datasets is summarized in Table 9. The datasets are described as follows:

**Colored image datasets.** Colored MNIST (**CMNIST**) is constructed by coloring and occluding a subset of MNIST (Arjovsky et al., 2019a). We define $a_1$ as the parity of digits with $a_1 = 0, 1$ representing "even" and "odd", and $a_2$ as the color of digits with $a_2 = 0, 1$ representing "red" and "blue". There are 3 and 2 modes under $a_1 = 0, 1$, respectively, i.e., digits 8, 4, and 2 under parity "even", and digits 3 and 9 under parity "odd". Colored Fashion-MNIST (**CFashion-MNIST**) is constructed based on Fashion-MNIST (Xiao et al., 2017). We define $a_1$ as the clothing style with $a_1 = 0, 1$ representing "sporty" and "chic", and $a_2$ as the color of clothing with $a_2 = 0, 1$ representing "red" and "blue". There are 3 and 4 modes under $a_1 = 0, 1$, respectively, i.e., "sneaker", "pullover", and "shirt" under style "sporty", and "sandal", "dress", "bag", "ankle boot" under style "chic". Noise is introduced in two forms to images: a random occlusion mask is applied to each image with occlusion ratio $\sigma$ as the noise level (Chai et al., 2021); a random scalar multiplier is applied to the RGB channels of the digit/clothing foreground as the coloring noise. The generated data are shown in Figure 19.

**Canine-BG.** Canine-BG is constructed by combining canine images from ImageNet (Deng et al., 2009) with environmental backgrounds from the Places dataset (Zhou et al., 2018) following (Sagawa et al., 2020), as shown in Figure 20. Canines are combined with environments by the following procedure: First, SAM (Kirillov et al., 2023) is used to obtain segmentation masks of the canine image, and CLIP (Radford et al., 2021) is used to select the mask that best fits the semantics of the dog breeds with the prompt "a photo of a {*dog breed*}"; then, the canine foregrounds are combined with environment background to generate the images. Attribute $a_1$ is defined as the functional categories of canines, i.e., $a_1 = 0, 1$ indicates "working dog" and "pet dog"; attribute $a_2$ is defined as the environmental backgrounds, i.e., $a_2 = 0, 1$ indicates "indoors" ("living room" environment from Places dataset) and "outdoors" ("forest" environment from Places dataset). $a_1$ has 10 modes under each attribute value, i.e., "Australian terrier", "Briard", "Welsh springer spaniel", "Kelpie", "Cairn terrier",

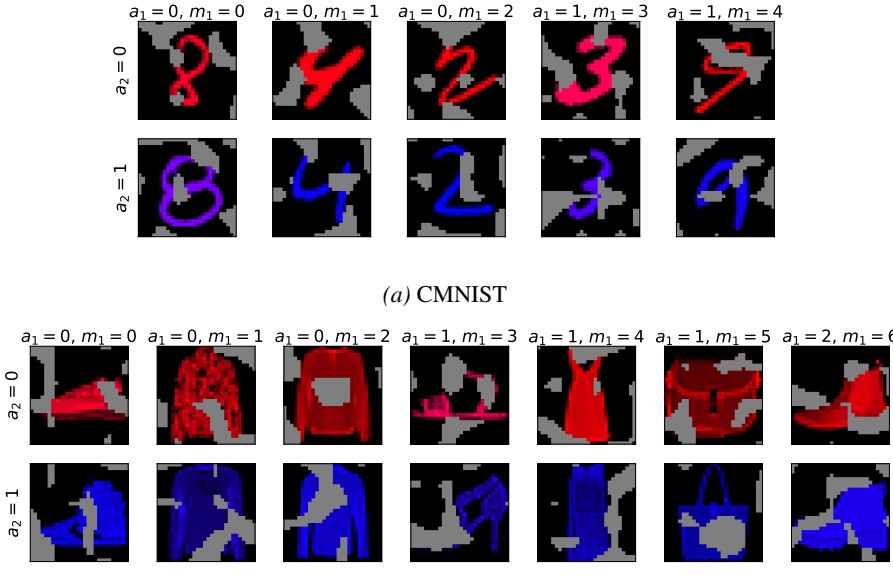

*(a)* CMNIST

*(b)* CFashion-MNIST

*Figure 19.* CMNIST and CFashion-MNIST data with occlusion and coloring noises.

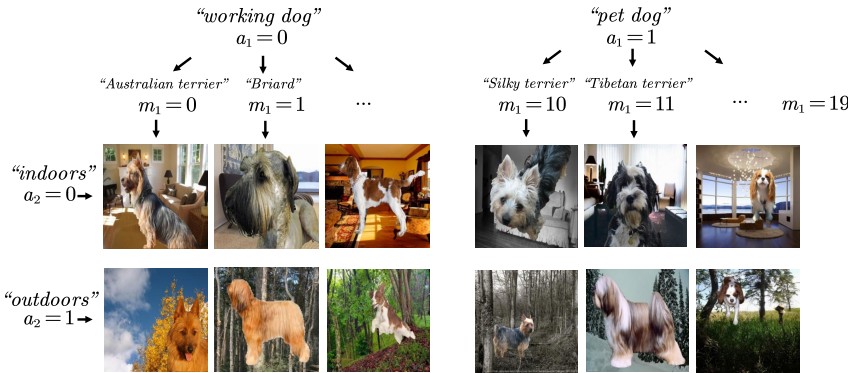

*Figure 20.* Canine-BG data with environment backgrounds.

"Bedlington terrier", "American Staffordshire terrier", "Border collie", "Bullmastiff", "Irish water spaniel" under category "working dog", and "Silky terrier", "Tibetan terrier", "Blenheim spaniel", "Toy terrier", "West Highland white terrier", "Standard poodle", "French bulldog", "Cardigan Welsh corgi", "Pug", "Miniature poodle" under category "pet dog". As a natural scenario, the environment of canines is often correlated to the functional categories and the specific canine breeds, e.g., working dogs are more likely to be outdoors.

**Time series datasets.** For wearable human activity recognition (WHAR) datasets **UCI-HAR** (Anguita et al., 2013), **RealWorld** (Sztyler & Stuckenschmidt, 2016), and **HHAR** (Stisen et al., 2015), we use 3-axis acceleration data from smartphones placed on the waist (UCI-HAR and HHAR) or chest (RealWorld) of users. We define $a_1$ as the human activity and $a_2$ as the user ID. For the machine fault diagnosis dataset **MFD** (Lessmeier et al., 2016), we use vibration signals from bearing sensors. We define $a_1$ as the fault type and $a_2$ as the operating condition. Pre-processing includes removing invalid values, performing channel-wise normalization that scales signals to [-1, 1], and segmenting time series data using sliding windows.

*Table 9.* Dataset descriptions.

| Dataset | CMNIST | CFashion-MNIST | Canine-BG | UCI-HAR | RealWorld | HHAR | MFD |
|---|---|---|---|---|---|---|---|
| $a_1$ | digit parity | fashion style | functional category | activity | activity | activity | incipient fault type |
| # values of $a_1$ | 2 | 2 | 2 | 6 | 8 | 6 | 3 |
| values of $a_1$ | even, odd | sporty, chic | work, pet | walking, walking upstairs, walking downstairs, sitting, standing, laying | climbing stairs up, climbing stairs down, jumping, lying, standing, sitting, running, walking | biking, sitting, standing, walking, stair up, stair down | healthy, inner-bearing damage, outer-bearing damage |
| $a_2$ | digit color | clothing color | background environment | user | user | user | operating condition |
| # values of $a_2$ | 2 | 2 | 2 | 30 | 15 | 9 | 4 |
| $m_1$ | digit | clothing | dog breed | activity modes | activity modes | activity modes | fault modes |
| # values of $m_1$ | 5 | 7 | 20 | unknown | unknown | unknown | unknown |
| values of $m_1$ | digit 8, 4, 2, 3, 9 | sneaker, pullover, shirt, sandal, dress, bag, ankle boot | Australian terrier, Briard, Welsh springer spaniel, Kelpie, Cairn terrier, Bedlington terrier, American Staffordshire terrier, Border collie, Bullmastiff, Irish water spaniel, and Silky terrier, Tibetan terrier, Blenheim spaniel, Toy terrier, West Highland white terrier, Standard poodle, French bulldog, Cardigan Welsh corgi, Pug, Miniature poodle | unknown | unknown | unknown | unknown |
| # samples | 35000 | 49000 | 11884 | 11711 | 36980 | 14772 | 10916 |
| # of groups | - | - | - | 5 | 5 | 3 | 4 |
| # channels | - | - | - | 3 | 3 | 3 | 1 |
| window length | - | - | - | 128 | 150 | 128 | 5120 |

## F.2. Evaluation Protocol

We introduce correlations to image datasets with constructed modes by sampling data with various correlations, and explore natural correlations on time series datasets with unknown modes by leave-one-group-out validation.

**Image datasets.** The number of samples for each mode is set to be equal, and hidden correlations are introduced by generating data with different conditional probabilities $p(a_2|m_1)$ across modes. For the main comparison with baselines and variants, noise level $\sigma$ is set to 0.2, and the conditional distribution $p(a_2 \mid m_1)$ on the training data is set according to $\boldsymbol{P}_{a_2|m_1}$, where the $(i,j)$-th component corresponds to $p(a_2 = i \mid m_1 = j)$:

$$\boldsymbol{P}_{a_2|m_1} = \begin{bmatrix} 0.1 & 0.9 & 0.1 & 0.9 & 0.1 \\ 0.9 & 0.1 & 0.9 & 0.1 & 0.9 \end{bmatrix} \text{ (CMNIST)} \tag{18}$$

$$\boldsymbol{P}_{a_2|m_1} = \begin{bmatrix} 0.9 & 0.1 & 0.1 & 0.1 & 0.9 & 0.9 & 0.1 \\ 0.1 & 0.9 & 0.9 & 0.9 & 0.1 & 0.1 & 0.9 \end{bmatrix} \text{ (CFahion-MNIST)} \tag{19}$$

For Canine-BG, $\boldsymbol{P}_{a_2|m_1} = [[0.1, 0.2, 0.3, 0.4, 0.1, 0.2, 0.2, 0.4, 0.3, 0.1, 0.9, 0.8, 0.7, 0.6, 0.9, 0.8, 0.8, 0.6, 0.7, 0.9], [0.9, 0.8, 0.7, 0.6, 0.9, 0.8, 0.8, 0.6, 0.7, 0.9, 0.1, 0.2, 0.3, 0.4, 0.1, 0.2, 0.2, 0.4, 0.3, 0.1]]$.

Test set 1 (correlated) follows the $\boldsymbol{P}_{a_2|m_1}$ configuration above; test set 2 (uncorrelated) uses $p(a_2 = i \mid m_1 = j) = 0.5$ for all $i, j$; and test 3 (anticorrelated) uses $1 - \boldsymbol{P}_{a_2|m_1}$ to configure $p(a_2 \mid m_1)$.

**Time series datasets.** To construct out-of-distribution tasks, we perform leave-one-group-out validation. The data are grouped by attribute $a_2$, with each group containing an equal number of $a_2$ values. We loop over the groups: each group is selected as the test data once, and the rest groups serve as training and validation data with a ratio of 0.8:0.2.

*Table 10.* Network architectures. "Conv(out, k, s)" denotes a 1D convolution layer with its output channels, kernel size, and stride. "FC(·)" denotes a fully connected layer with its output dimension. "LeakyReLU(·)" denotes a LeakyReLU activation with its scale. "Reshape(·, ·)" transforms vectors into 2D matrices for convolutional inputs.

| Component | Dataset | Architectures |
|---|---|---|
| Encoder subnetwork | Colored image | FC(128), BN → FC(128), BN |
| Decoder | Colored image | FC(256), BN → FC(3 × 28 × 28) |
| Encoder subnetwork | Canine-BG | ResNet18 |
| Decoder | Canine-BG | Transposed convolutional network (https://github.com/Horizon2333/imagenet-autoencoder) |
| Encoder subnetwork | WHAR | Conv(out=128, k=8, s=2), BN → Conv(out=256, k=5, s=2), BN → Conv(out=128, k=3, s=1), BN, MaxPool(2) |
| Decoder | WHAR | FC($c_{dec}$ × 32), BN, LeakyReLU(0.2) → Reshape($c_{dec}$, 32) → Conv(out=16, k=3, s=1), BN, LeakyReLU(0.2), Upsample(2) → Conv(out=8, k=3, s=1), BN, LeakyReLU(0.2), Upsample(2) → Conv(out=3, k=3, s=1), Tanh ($c_{dec}$ = 41 for RealWorld and 35 for UCI-HAR and HHAR) |
| Encoder subnetwork | MFD | Conv(out=64, k=32, s=6), BN → Conv(out=128, k=8, s=2), BN → Conv(out=128, k=8, s=2), BN, MaxPool(2) |
| Decoder | MFD | FC(128 × 214), BN, LeakyReLU(0.2) → Reshape(128, 214) → Upsample(2), Conv(out=32, k=8, s=1), BN, LeakyReLU(0.2) → Upsample(2), Conv(out=8, k=8, s=1), BN, LeakyReLU(0.2) → Upsample(6), Conv(out=1, k=33, s=1), BN, LeakyReLU(0.2) |
| Predictor | All | FC(·), Softmax |
| Discriminator subnetwork | All | FC(512), ReLu → FC(1), Sigmoid |
| Weight Net subnetwork | All | FC(32), ReLu → FC(1), Sigmoid |
| Subclustering Net subnetwork | All | FC(256), ReLu → FC(2) |

### F.3. Computing Environment

All methods are implemented in PyTorch 2 (Paszke et al., 2019) with Python 3.10. Model optimization is performed with Adam (Kingma & Ba, 2015). Experiments are conducted on Linux servers with the following hardware configurations: (1) Intel(R) Core(TM) i9-12900K CPU and NVIDIA RTX 3090 GPU, (2) AMD Ryzen 9 7950X3D 16-Core Processor CPU and NVIDIA GeForce RTX 4090 D GPU, and (3) AMD EPYC 9654 96-Core Processors with NVIDIA H100 HBM3 GPU.

### F.4. Network Architectures

We summarize the network architectures in Table 10. Encoder and decoder are constructed as multi-layer perceptrons (MLPs) for image datasets CMNIST and CFashion-MNIST, as ResNet18 architectures for Canine-BG, and as convolutional networks for time series datasets. Encoder employs an individual subnetwork to learn the representations of each attribute, and decoder fuses the representations of different attributes to reconstruct data. Predictors, discriminator, weight net, and subclustering Net are constructed as MLPs. Discriminator, weight net, and subclustering Net employ an individual subnetwork for each attribute value, which receives the cluster labels under this attribute value as the conditional input, i.e., these networks use parameter sharing between the clusters under the same attribute value. Image datasets CMNIST and CFashion-MNIST share the same architectures, and wearable human activity recognition (WHAR) datasets UCI-HAR, RealWorld, and HHAR share similar architectures.

**Architecture changes after clustering iterations.** When the number of clusters changes during iterative clustering, only the neurons in four layers are adjusted by adding neurons that correspond to the emerging clusters after splitting, or removing neurons that correspond to the vanished clusters after merging. The adjusted layers are: (1) the output layer of mode predictor, (2) the output layer of subclustering net, (3) the conditional input layer in discriminator, and (4) the conditional input layer in weight net. Other network parameters remain unchanged, facilitating stable training.

---

**Algorithm 1** The training process of CoDID

---

1: **Input:** Training data $\boldsymbol{x}$ with attributes labels $a_1, a_2$, the number of epochs $N_{\text{ptr}}, N_{\text{tr}}, N_{\text{cl}}, N_{\text{it}}$, and the number of steps $N_{\text{d}}$ for discriminator updates, initial number of clusters $K^{\text{c}}(\alpha)^{(t=0)}$

2: Initialize encoder $F$, attribute predictors $C_1, C_2$, mode predictor $C_1^{\text{m}}$, decoder $R$, discriminator $D$, weight net $W$, and sublcustering net $S$

3: **for** $epoch = 1$ **to** $N_{\text{ptr}}$ **do**

4:     **for** mini-batch $(\boldsymbol{x}, a_1, a_2)$ **do**

5:         Update encoder $F$, predictors $C_1, C_2, C_1^{\text{m}}$, and decoder $R$ by minimizing $\mathcal{L}_{\text{info}}$ in Equation 4

6:     **end for**

7: **end for**

8: Under each value $\alpha$ of $a_1$, initialize iteration $t = 0$, initialize DPGMM clustering with the number of clusters $K^{\text{c}}(\alpha)^{(t=0)}$ on the output representations $\boldsymbol{z}_1$ of the pre-trained encoder, and get the cluster labels $c_1^{(t=0)}$

9: **for** $epoch = 1$ **to** $N_{\text{tr}}$ **do**

10:     **if** $epoch \mod N_{\text{cl}} = 0$ and $epoch < N_{\text{it}}$ **then**

11:         $t = t + 1$

12:         Refine DPGMM clustering to get the cluster labels $c_1^{(t)}$ and number of clusters $K^{\text{c}}(\alpha)^{(t)}, \alpha \in \mathcal{A}_1$, make merge proposals at $t = 0, 2, 4, \ldots$ and split proposals at $t = 1, 3, 5, \ldots$

13:         Update the network architectures of predictor $C_1^{\text{m}}$, discriminator $D$, weight net $W$, and subclustering net $S$ if any split or merge decisions are made

14:     **end if**

15:     **for** mini-batch $(\boldsymbol{x}, a_1, a_2, c_1^{(t)})$ **do**

16:         Update encoder $F$ and predictors $C_1, C_2, C_1^{\text{m}}$, and decoder $R$ by minimizing $\mathcal{L}_{\text{info}}$ in Equation 4, update subclustering net $S$ by minimizing $\mathcal{L}_{\text{sc}}$ in Equation 11

17:         **for** $step = 1$ **to** $N_{\text{d}}$ **do**

18:             Update discriminator $D$ by minimizing $\mathcal{L}_{\text{wd}}$ in Equation 5 with equal weights $w' = 1$

19:         **end for**

20:         Update weight net $W$ by minimizing $\mathcal{L}_{\text{meta}}$ in Equation 8

21:         Update encoder $F$ by maximizing $\mathcal{L}_{\text{wd}}$ in Equation 5 with weights $w$ from weight net

22:     **end for**

23: **end for**

24: **Output:** Encoder $F$ and predictor $C_1$

---

### F.5. Training Process

We summarize the training process of CoDID in Algorithm 1. Alternative optimization steps are performed to minimize different losses.

**Extension to multiple attributes.** We focus on disentangling a certain attribute $a_q, q = 1$ with underlying modes to facilitate downstream attribute prediction tasks, e.g., activity recognition that aims to predict the activity of users. Although we only include one other attribute $a_2$ in our formulations, our model is capable of handling multiple other attributes $a_{-q} = \{a_j\}_{j \neq q}$. Our model can be extended to multiple attributes by adding the corresponding attribute predictors for each attribute in $a_{-q}$, using joint representations $\boldsymbol{z}_{-q}$ in reconstruction, and using the joint representations of $\boldsymbol{z}_{-q}$ in mode-based CMI minimization, i.e., minimizing $I(\boldsymbol{z}_q; \boldsymbol{z}_{-q} | c_q)$ instead (Funke et al., 2022). If the disentangled representations of some other attribute $a_j, j \neq q$ are required for the downstream task, our model can be extended by adding a conditional discriminator that minimizes the attribute/mode-based CMI between $a_j$ and $a_{-j}$, depending on whether underlying modes exist under the values of this attribute.

### F.6. Hyperparameter Settings

We specify the hyperparameter settings as follows. For all datasets, the mini-batch size is set to 128. For CMNIST, CFashion-MNIST, and time series datasets, the number of dimensions $d_{\text{z}}$ for representations $\boldsymbol{z}$ is set to 128. For Canine-BG, the number of dimensions $d_{\text{z}}$ for representations $\boldsymbol{z}$ is set to 512, corresponding to ResNet18. The number of epochs for pre-training, $N_{\text{ptr}}$, the number of epochs for supervised DRL, $N_{\text{tr}}$, and the number of epochs before stopping clustering refinement, $N_{\text{it}}$, are set to 20, 30, 30 on image datasets, 50, 150, 150 on UCI-HAR and MFD, 50, 100, 100 on RealWorld,

*Table 11.* Hyperparameter search spaces and NNI settings.

| Hyperparameter | Search space |
|---|---|
| $\lambda_{\mathrm{mp}}$ | $\{10^{-4}, 10^{-3}, 10^{-2}, 10^{-1}\}$ or $\{0.1, 0.3, \ldots, 0.9\}$ |
| $\lambda_{\mathrm{re}}$ | $\{0.6, 0.7, \ldots, 1.4\}$ |
| $\lambda_{\mathrm{mr}}$ | $\{0.6, 0.8, \ldots, 1.4\}$ |
| $\lambda_{\mathrm{wa}}$ | $\{0.1, 0.2, \ldots, 0.5\}$ |
| $K^c(\cdot)^{(0)}$ | $\{2, 3, \ldots, 10\}$ |
| $l_{\mathrm{d}}$ | $\{10^{-4}, 3{\times}10^{-4}, 5{\times}10^{-4}, 7{\times}10^{-4}, 10^{-3}\}$ |
| $N_{\mathrm{d}}$ | $\{1, 3, \ldots, 17\}$ |

**Notes.** For $\lambda_{\mathrm{mp}}$, we observe that smaller values perform better on MFD and thus use the choice $\{10^{-4}, 10^{-3}, 10^{-2}, 10^{-1}\}$; on other datasets, we use the choice $\{0.1, 0.2, \ldots, 0.9\}$.

and 50, 150, 100 on HHAR, respectively.

We use Neural Network Intelligence (NNI)[2] to tune a subset of hyperparameters with Tree-structured Parzen Estimator (TPE). The search spaces of the parameters are summarized in Table 11, and the selected values are as follows: The loss coefficients $(\lambda_{\mathrm{mp}}, \lambda_{\mathrm{re}}, \lambda_{\mathrm{mr}}, \lambda_{\mathrm{wa}})$ are set to (0.3, 1.1, 1, 0.3) on CMNIST, (0.7, 0.9, 1, 0.3) on CFashion-MNIST, (0.5, 1.2, 1, 0.1) on UCI-HAR, (0.1, 1, 1, 0.2) on RealWorld, (0.1, 0.8, 1, 0.2) on HHAR, and (0.01, 1, 1, 0.2) on MFD, respectively. The initial number of clusters $K^c(\alpha)^{(0)}$ under each value $\alpha$ of $a_1$ is set to (6,4), (6,8) on CMNIST and CFashion-MNIST under $\alpha = 0, 1$ (tuned separately under each $\alpha$), and set to 6, 2, 3, and 4 on UCI-HAR, RealWorld, HHAR, and MFD under all $a_1$ values (use a single shared value for all $\alpha$ and tuned as one parameter). The initial learning rate of Adam for discriminator, $l_{\mathrm{d}}$, is set to 0.0003, 0.0003, 0.0007, 0.0007, 0.0005, 0.001 on CMNIST, CFashion-MNIST, UCI-HAR, RealWorld, HHAR, and MFD, respectively. The initial learning rates for other networks are set to 0.001 on all datasets. The number of update steps $N_{\mathrm{d}}$ is set to 15, 15, 7, 7, 1, and 1 on CMNIST, CFashion-MNIST, UCI-HAR, RealWorld, HHAR, and MFD, respectively.

### F.7. Baselines

For fairness, we implement all baselines with the same encoder–decoder backbone as CoDID, with appropriate adaptations for variational methods, i.e., Gaussian parameter outputs and reparameterization tricks. The descriptions of the baselines for DRL on attributes $a_1, a_2$ are summarized below:

- **MMD** (Lin et al., 2020) learns the generalized representations of attribute $a_1$ by aligning its representation distribution across different values of $a_2$.

- **DANN** (Ganin et al., 2016) adversarially trains an attribute predictor to make $a_2$ unpredictable from the representations of $a_1$.

- **CORAL** (Sun & Saenko, 2016) aligns the second-order statistics of the representations of $a_1$ computed from data with any two distinct values of $a_2$.

- **IRM** (Arjovsky et al., 2019b) learns invariant representations of $a_1$ by ensuring its optimal predictor is simultaneously optimal for all values of $a_2$.

- **REx** (Krueger et al., 2021) reduces the variance of the prediction risk of $a_1$ across different values of $a_2$.

- **DTS** (Li et al., 2022) adversarially trains attribute predictors to make $a_1$ unpredictable from the representations of $a_2$, and vice versa.

- **IDE-VC** (Yuan et al., 2021) minimizes the MI between the representations of $a_1, a_2$ by adversarially training representation estimators to make the representations of $a_2$ unpredictable from the representations of $a_1$.

- **MI** (Cheng et al., 2022) minimizes the MI between the representations of $a_1, a_2$ by adversarially training a discriminator to align the distributions of the joint and marginal representations.

---

[2]https://github.com/microsoft/nni

- **A-CMI** (Funke et al., 2022) minimizes the attribute-based CMI between the representations of $a_1, a_2$ by adversarially training two conditional discriminators, each conditioned on one attribute, to align the conditional distributions of the joint and marginal representations.

- **HFS** (Roth et al., 2023) minimizes the Hausdorff distance between two representation sets to factorize the supports of the representations of $a_1, a_2$, where we use Euclidean distance as the distance measure between representations with reference to (Oublal et al., 2024b).

- **CODA** (Ou & Zhang, 2024) generates synthetic samples by decoding the combined representations from different instances, and enforces the model to deliver consistent predictions on the original and synthesized samples.

- **ID-FaceVC** (Rong & Liu, 2025) minimizes the MI between the representations of $a_1, a_2$ using a variational upper bound technique.

- **DIOSC** (Oublal et al., 2024a) leverages weak contrastive learning to minimize the information overlap between representations and their negatives, while compelling similarity between representations and their augmented representations.

## G. Preliminaries: Non-parametric DPGMM Clustering with Split/Merge Operations

We adopt an existing Dirichlet Process Gaussian Mixture Model (DPGMM) clustering model as the backbone of our mode-discovery module. A concise overview of the DPGMM we used is provided below. For full algorithmic details and the split/merge framework, please refer to the prior works (Ronen et al., 2022; Jain & Neal, 2004). We describe a DPGMM for generic input data $x$ in this section. The notations here are standalone and independent of those used in our method.

**The DPGMM model.** DPGMM is a Bayesian non-parametric extension of the classic Gaussian Mixture Model (GMM). DPGMM uses a Dirichlet process prior, supporting an infinite mixture with weight $\pi_k$ and Gaussian parameters $\theta_k = (\boldsymbol{\mu}_k, \boldsymbol{\Sigma}_k)$ for the $k$-th component:

$$p(\boldsymbol{x}|(\pi_k, \boldsymbol{\mu}_k, \boldsymbol{\Sigma}_k)_{k=1}^{\infty}) = \sum_{k=1}^{\infty} \pi_k \mathcal{N}(\boldsymbol{x}; \boldsymbol{\mu}_k, \boldsymbol{\Sigma}_k) \tag{20}$$

For $N$ data points, $\mathcal{X} = (\boldsymbol{x}_i)_{i=1}^{N}, \boldsymbol{x}_i \in \mathbb{R}^d$, let $c_i$ denote the cluster label of $\boldsymbol{x}_i$. We specify a DPGMM via the following generative process:

$$\pi = (\pi_k)_{k=1}^{\infty} \sim GEM(1, \alpha^{\star}), \tag{21}$$

$$\boldsymbol{\theta}_k = (\boldsymbol{\mu}_k, \boldsymbol{\Sigma}_k) \overset{iid}{\sim} NIW(\boldsymbol{\mu}^{\star}, \kappa^{\star}, \boldsymbol{\Psi}^{\star}, \nu^{\star}), \quad k \in \{1, ...\}, \tag{22}$$

$$c_i \sim \mathrm{Cat}(\boldsymbol{\pi}), \quad i \in \{1, ..., N\} \tag{23}$$

$$\boldsymbol{x}_i|c_i \sim \mathcal{N}(\boldsymbol{x}_i; \boldsymbol{\theta}_{c_i}), \quad i \in \{1, ..., N\} \tag{24}$$

The weights $\boldsymbol{\pi} = (\pi_k)_{k=1}^{\infty}$ are drawn from the Griffiths–Engen–McCloskey stick-breaking process (GEM) (Pitman, 2002) with a concentration parameter $\alpha^{\star} > 0$. The parameters $\boldsymbol{\theta} = (\boldsymbol{\theta}_k)_{k=1}^{\infty}$ are independent and identically distributed (i.i.d.) draws from a Normal–Inverse Wishart (NIW) distribution. For data $\boldsymbol{x} \in \mathbb{R}^d$, NIW hyperparameters are denoted as $\lambda = (\boldsymbol{\mu}^{\star}, \kappa^{\star}, \boldsymbol{\Psi}^{\star}, \nu^{\star})$: a vector $\boldsymbol{\mu}^{\star} \in \mathbb{R}^d$, a positive real number $\kappa^{\star} > 0$, a symmetric positive-definite (SPD) matrix $\boldsymbol{\Psi}^{\star} \in \mathbb{R}^{d \times d}$, and a positive real number $\nu^{\star} > d - 1$. Roughly speaking, the higher $\nu^{\star}$ and $\kappa^{\star}$ are, the more the distribution is peaked around $\boldsymbol{\Psi}^{\star}$ and $\boldsymbol{\mu}^{\star}$, respectively.

While there may be infinite-many components, the number of clusters is bounded by the number of samples on a finite dataset. Thus, a truncated version with a finite cluster count is typically used in practice. DPGMM seeks to infer the cluster labels, involving the number of clusters $K$ and parameters $(\pi_k, \boldsymbol{\theta}_k)_{k=1}^{K}$. The inference is affected by the data as well as GEM and NIW hyperparameters.

**Split/merge framework.** Split/merge operations allow the change of $K$ via the Metropolis-Hastings framework (Hastings, 1970): *split* proposes to divide a cluster into its two subclusters ($K \leftarrow K + 1$); *merge* proposes to combine two clusters into one ($K \leftarrow K - 1$).

For cluster split, each component $(\pi_k, \boldsymbol{\theta}_k)$ is augmented with two subcomponents ($\tilde{\boldsymbol{\theta}} = (\tilde{\boldsymbol{\theta}}_{k,1}, \tilde{\boldsymbol{\theta}}_{k,2}), \tilde{\boldsymbol{\pi}}_k = (\tilde{\pi}_{k,1}, \tilde{\pi}_{k,2})$ ), forming a 2-component GMM; each cluster label $c_i$ is augmented with a subcluster label $\tilde{c}_i \in \{1, 2\}$. A split proposal for splitting a cluster $k$ is accepted with probability $\min(1, H_s)$, where the Hastings ratio $H_s$ is computed as:

$$H_s = \frac{\alpha\,\Gamma(N_{k,1})\,f_x(\mathcal{X}_{k,1}; \lambda)\,\Gamma(N_{k,2})\,f_x(\mathcal{X}_{k,2}; \lambda)}{\Gamma(N_k)\,f_x(\mathcal{X}_k; \lambda)} \tag{2}$$

where $\Gamma$ is the Gamma function, $\mathcal{X}_k = (x_i)_{i:c_i=k}$ denotes the points in cluster $k$, $N_k = |\mathcal{X}_k|$ denotes the number of points in cluster $k$, $\mathcal{X}_{k,j} = (x_i)_{i:(c_i, \tilde{c}_i)=(k,j)}, j \in \{1, 2\}$ denotes the points in subcluster $j$, $N_{k,j} = |\mathcal{X}_{k,j}|$, and $f_x(\cdot\,; \lambda)$ denotes the marginal likelihood under NIW hyperparameters $\lambda$ (defined later). The ratio $H_s$ can be interpreted as comparing the marginal likelihood of the data under the two subclusters with its marginal likelihood under the cluster.

Similarly, a merge proposal for merging two clusters $k_1$ and $k_2$ is accepted with probability $(1, H_m)$, where the Hastings ratio $H_m$ is computed as:

$$H_m = \frac{1}{H_s} = \frac{\Gamma(N_{k_1} + N_{k_2})\,f_x(\mathcal{X}_{\{k_1, k_2\}}; \lambda)}{\alpha\,\Gamma(N_{k_1})\,f_x(\mathcal{X}_{k_1}; \lambda)\,\Gamma(N_{k_2})\,f_x(\mathcal{X}_{k_2}; \lambda)}$$

where $\mathcal{X}_{k_j} = (x_i)_{i:c_i=k_j}, j \in \{1, 2\}$ denotes the points in each cluster $k_1, k_2$, $N_{k_j} = |\mathcal{X}_{k_j}|$, and $\mathcal{X}_{\{k_1, k_2\}} = (x_i)_{i:\,c_i \in \{k_1, k_2\}}$ denotes the points in clusters $k_1$ and $k_2$.

The *marginal likelihood* of the data $(\boldsymbol{x}_i)_{i=1}^{N_k}$ in a cluster $k$ is obtained by marginalizing over the Gaussian parameters given the hyperparameters $\lambda$ of the NIW prior:

$$f_x\Big( (\boldsymbol{x}_i)_{i=1}^{N_k}; \lambda \Big) = \int \prod_{i=1}^{N_k} \mathcal{N}(\boldsymbol{x}_i \mid \boldsymbol{\mu}_k, \boldsymbol{\Sigma}_k)\; NIW(\boldsymbol{\mu}_k, \boldsymbol{\Sigma}_k; \lambda)\; d(\boldsymbol{\mu}_k, \boldsymbol{\Sigma}_k)\,.$$

**Inferring process.** In every iteration, cluster parameters are updated using the EM-algorithm with Maximum-a-Posteriori (MAP) estimates. Split and merge are proposed alternately every certain number of iterations. The marginal likelihood of data can be computed in closed form with the posterior NIW hyperparameters. For the computational details, please refer to (Ronen et al., 2022).

## H. Full Results on Three Test Sets of CMNIST and CFashion-MNIST

The full results on all three test sets for baseline comparisons on CMNIST and CFashion-MNIST are reported in Table 12, 13, 14, respectively. We observe that the advantage of CoDID becomes more prominent from test 1 to 3 with enlarging train-test correlation shifts, demonstrating the robustness of CoDID.

*Table 12.* Full comparison with baselines on CMNIST (mean±std., in percentage). The notations follow Table 1.

| Method | Test 1 (correlated) | | Test 2 (uncorrelated) | | Test 3 (anticorrelated) | |
| --- | --- | --- | --- | --- | --- | --- |
| | Acc. | Mac. F1 | Acc. | Mac. F1 | Acc. | Mac. F1 |
| BASE | 95.4±0.3 | 95.2±0.3 | 88.1±0.3 | 87.6±0.3 | 79.2±0.9* | 78.3±0.9* |
| MMD | 73.6±1.2* | 61.8±2.0* | 65.5±2.8* | 52.4±4.7* | 57.3±2.7* | 41.2±7.0* |
| DANN | 85.1±0.6* | 84.5±0.6* | 70.7±1.5* | 69.8±1.3* | 58.3±2.9* | 56.9±2.6* |
| CORAL | 74.7±0.4* | 74.4±0.4* | 66.0±2.0* | 65.4±1.7* | 58.6±3.2* | 57.6±3.0* |
| IRM | 95.5±0.8 | 95.3±0.8 | 88.3±2.3 | 87.8±2.0 | 81.0±3.5* | 80.3±3.2* |
| REx | **96.0**±0.5 | **95.8**±0.5 | 88.3±1.4 | 87.8±1.1 | 81.7±3.3* | 80.9±3.6* |
| DTS | 70.8±1.6* | 63.1±1.5* | 63.1±2.9* | 53.6±7.4* | 55.6±3.0* | 44.0±1.9* |
| IDEVC | 72.0±2.1* | 68.8±2.4* | 61.9±1.5* | 57.4±1.0* | 54.2±4.0* | 49.1±2.9* |
| MI | 67.8±1.5* | 60.4±1.7* | 62.3±1.9* | 52.8±5.4* | 56.9±4.0* | 45.2±2.8* |
| A-CMI | 70.9±1.1* | 62.5±1.7* | 65.0±4.6* | 52.9±6.4* | 58.5±2.3* | 41.2±5.9* |
| HFS | 91.2±0.5* | 90.8±0.4* | 78.9±1.6* | 77.9±1.6* | 66.5±2.1* | 64.9±2.1* |
| CODA | 92.7±0.3* | 92.4±0.4* | 81.3±1.1* | 80.5±1.2* | 69.8±1.9* | 68.3±2.3* |
| ID-FaceVC | 87.9±0.3* | 87.4±0.4* | 73.2±1.3* | 72.0±1.3* | 58.1±2.7* | 56.2±2.4* |
| DIOSC | 93.0±0.4 | 92.7±0.3 | 83.1±0.3* | 82.6±0.3* | 73.4±0.7* | 72.6±1.0* |
| CoDID | 95.4±0.9 | 95.2±0.9 | **89.9**±1.2 | **89.6**±1.3 | **85.2**±2.7 | **84.6**±3.0 |
| **Improvement** | -0.6 % | -0.6 % | +1.6 % | +1.8 % | +3.5 % | +3.7 % |

*Table 13.* Full comparison with baselines on CFashion-MNIST (mean±std., in percentage). The notations follow Table 1.

| Method | Test 1 (correlated) | | Test 2 (uncorrelated) | | Test 3 (anticorrelated) | |
|---|---|---|---|---|---|---|
| | Acc. | Mac. F1 | Acc. | Mac. F1 | Acc. | Mac. F1 |
| BASE | **95.8**±0.3 | **95.7**±0.3 | 86.7±0.9* | 86.4±0.9* | 77.8±1.5* | 77.1±1.5* |
| MMD | 94.3±0.3 | 94.1±0.2 | 78.7±1.8* | 78.3±1.9* | 63.5±3.6* | 62.7±3.7* |
| DANN | 90.2±0.6* | 90.0±0.6* | 76.7±1.7* | 76.0±1.8* | 63.8±2.5* | 62.5±2.8* |
| CORAL | 95.4±0.4 | 95.3±0.4 | 81.7±2.0* | 81.4±1.9* | 68.0±3.4* | 67.3±3.0* |
| IRM | 95.7±0.8 | **95.7**±0.8 | 87.4±2.3 | 87.2±2.2 | 79.3±2.6* | 78.9±2.5* |
| REx | **95.8**±0.5 | **95.7**±0.5 | 86.1±1.3* | 85.9±1.1* | 76.3±3.5* | 75.9±3.2* |
| DTS | 85.3±0.9* | 85.1±0.9* | 73.2±1.6* | 72.6±1.6* | 61.2±2.0* | 60.1±2.0* |
| IDEVC | 92.4±0.3* | 92.2±0.3* | 75.4±2.3* | 74.6±2.5* | 58.7±3.8* | 57.1±4.3* |
| MI | 92.0±1.0* | 91.8±1.1* | 77.7±1.8* | 77.0±2.0* | 62.1±3.0* | 60.7±3.0* |
| A-Cmi | 91.9±1.6* | 91.7±1.7* | 76.9±3.3* | 76.1±3.7* | 61.8±5.3* | 60.0±6.5* |
| HFS | 93.9±0.7 | 93.8±0.7 | 80.0±1.5* | 79.6±1.2* | 66.2±3.6* | 65.3±3.1* |
| CODA | 94.6±0.5 | 94.4±0.5 | 83.7±0.8* | 83.2±1.0* | 72.5±1.5* | 71.3±1.9* |
| ID-FaceVC | 93.5±0.2 | 93.3±0.2 | 78.5±0.4* | 77.9±0.6* | 63.6±1.4* | 62.5±1.7* |
| DIOSC | 94.7±0.4 | 94.6±0.4 | 83.8±0.8* | 83.5±0.7* | 73.2±1.6* | 72.8±1.4* |
| CoDID | 95.3±0.4 | 95.2±0.4 | **89.4**±1.3 | **89.1**±1.3 | **84.1**±3.2 | **83.5**±3.3 |
| **Improvement** | -0.5 % | -0.5 % | +2.0 % | +1.9 % | +4.8 % | +4.6 % |

*Table 14.* Full comparison with baselines on Canine-BG (mean±std., in percentage). The notations follow Table 1.

| Method | Test 1 (correlated) | | Test 2 (uncorrelated) | | Test 3 (anticorrelated) | |
|---|---|---|---|---|---|---|
| | Acc. | Mac. F1 | Acc. | Mac. F1 | Acc. | Mac. F1 |
| BASE | 81.0±1.9* | 80.7±2.2* | 68.6±1.5* | 68.1±1.9* | 55.6±2.0* | 54.6±2.3* |
| MMD | 51.1±5.5* | 49.0±7.4* | 52.6±5.9* | 50.7±8.2* | 54.1±6.7* | 52.1±9.4* |
| DANN | 71.3±4.6* | 71.2±6.6* | 50.8±4.7* | 50.7±6.4* | 29.8±4.4* | 29.6±4.8* |
| CORAL | 50.7±2.4* | 33.7±2.4* | 51.2±2.3* | 33.9±3.9* | 51.0±3.5* | 33.8±3.2* |
| IRM | 83.9±3.8 | 83.9±3.8 | 69.1±4.5* | 69.1±3.1* | 54.3±5.6* | 54.3±5.1* |
| REx | 84.1±4.5 | 84.1±5.5 | 70.2±4.6* | 70.2±3.3* | 56.0±5.2* | 56.0±5.7* |
| DTS | 59.9±4.6* | 59.9±6.2* | 51.9±0.8* | 46.9±1.9* | 41.8±4.7* | 35.7±2.3* |
| IDE-VC | 51.0±2.5* | 47.6±2.6* | 50.1±0.7* | 46.7±5.9* | 49.1±2.2* | 45.7±2.0* |
| MI | 72.9±4.2* | 72.3±4.9* | 62.9±3.0* | 62.0±3.4* | 52.3±4.8* | 51.0±4.4* |
| A-CMI | 73.0±2.6* | 72.9±2.6* | 63.8±5.2* | 63.6±5.2* | 53.0±8.2* | 52.8±8.1* |
| HFS | 79.8±1.6* | 79.7±1.8* | 63.9±2.3* | 63.6±2.5* | 46.8±3.7* | 46.2±4.1* |
| CODA | 59.2±1.3* | 59.2±1.3* | 51.9±0.8* | 51.9±0.8* | 43.9±2.2* | 43.9±2.3* |
| ID-FaceVC | 59.9±6.6* | 59.4±6.6* | 49.4±1.3* | 49.1±1.8* | 42.1±6.2* | 41.3±6.7* |
| DIOSC | 59.4±5.2* | 58.9±5.4* | 51.8±1.0* | 51.1±1.3* | 42.5±5.4* | 42.0±5.3* |
| CoDID | **85.3**±4.8 | **85.3**±5.0 | **75.8**±3.2 | **75.7**±3.5 | **68.3**±5.2 | **68.2**±5.7 |
| **Improvement** | +1.2 % | +1.2 % | +5.6 % | +5.5 % | +12.3 % | +12.2 % |

