# OpenReview forum: "Coordinated Disentanglement with Iterative Mode Discovery Under Hidden Correlations"
_ICML.cc/2026/Conference — ICML 2026 regular_

### Official Review · Reviewer_pmwH · 2026-03-08

**Soundness:** 3
**Presentation:** 3
**Significance:** 3
**Originality:** 3
**Overall Recommendation:** 6
**Confidence:** 3

**Summary:**

The paper proposes a method for disentangled representation learning, in which we are given training examples labeled along multiple attributes and must learn representations for each attribute that are ideally not influenced by the other attributes. This is important because the attributes’ distributions and correlations could shift between train and test time. Prior methods have done this by minimizing mutual information between the representations, conditioned on the attributes to remove the impact of attribute-attribute correlation. However, these methods fail to guard against hidden correlations that might arise when modes within a value of an attribute correlate with the values of other attributes. The paper shows theoretically that these hidden correlations can degrade performance and that conditioning on the modes would address the problem, but only if the modes are fully accounted for. The paper proposes method CoDID, which uses DPGMM to find modes, adversarial learning to minimize mutual information conditioned on them, and sample weighting to mitigate the effects of possible underclustering. CoDID is benchmarked on multiple synthetic image and real-world time-series datasets and outperforms existing baselines by a significant margin.

**Compliance With Llm Reviewing Policy:**

Affirmed.

**Final Justification:**

I initially rated this paper as a 5, as I felt the work was solid but could be strengthened by more in-depth computational cost analysis, analysis on bigger backbones, and analysis of convergence reliability given the use of adversarial methods. Another reviewer also raised a concern I hadn't initially thought of, which was missing comparison to OOD-generalization methods. The authors addressed all of these concerns convincingly, making for a stronger paper, so my final recommendation is a 6.

**Key Questions For Authors:**

See weaknesses, particularly the questions in items 1, 2, 4, and 6. I am open to increasing my score depending on how the questions are addressed in the rebuttal and what the other reviewers say.

**Limitations:**

yes

**Strengths And Weaknesses:**

**Strengths**:

1. The paper proposes an interesting idea that’s nicely connected to the theory.

2. The method is evaluated on a decent number of benchmarks, both image and time series, and appears to get a solid performance gain.

3. Analysis seems very thorough. It includes ablations, hyperparameter sensitivity, compute cost, and explorations of the modes found in the datasets.

4. Source code is provided, which is a plus.

**Weaknesses**:

1. CoDID adds clustering and meta-optimization modules to the existing CMI method, which could impact its computational performance. I know that compute cost is studied in Appendix A.7, but it would be nice to break it down by component and also get a sense of whether the DPGMM component makes the method significantly CPU-bound.

2. CoDID uses adversarial learning, which can sometimes have convergence issues. Does the addition of clustering have a positive or negative impact on the reliability of convergence?

3. CoDID assumes that correlation is due to discrete modes. How well would CoDID perform in scenarios where correlation was caused by relationships between continuous variables?

4. Architectures used are maybe somewhat limited. For example, the largest model used for image datasets is ResNet18. It would be interesting to see if the performance gains extend to larger encoders, even if it was just on one or two datasets against one or two baselines.

5. The image datasets are all synthetically constructed and have synthetically created correlations. For me this is mostly mitigated by the fact that the time-series datasets have real-world data and correlations, but it would have been nice to include real-world image data as well.

6. Presentation/clarity could be improved a bit, especially for audiences not as familiar with disentangled representation learning. Some suggestions and questions:
    * It would help to make it clearer in Fig. 1 how the correlation illustrated leads to incorrect predictions at test time.
    * Maybe consider using concrete examples of attribute values and modes in Section 3.1.
    * I’m not sure I entirely understand how $L_{meta}$ incentivizes weights that would align the marginal with the joint (as described in “Our Solution” paragraph). Do the gradients from $L_{sc}$ encourage $W$ to produce those weights, or is there some other mechanism?
    * How do the weights and subcluster net influence the mode-finding process?
    * Minor point: Alg. 1 does not mention $L_{sc}$ at all.

---

> ### Author Rebuttal · Authors · 2026-03-31
>
> Thank you very much for your positive evaluation. We highly value your constructive comments and address each point below.
>
> ### W1. Computational cost
> We further break down the training time and parameter counts by component (***Table 1&2 at https://anonymous.4open.science/api/repo/CoDID-B038/file/supplementary.pdf?v=baeedeaa***). For training time, the dominant cost is still **adversarial training**, especially the discriminator learning; **DPGMM clustering only takes up a very small fraction** of the total time. This is because DPGMM is updated after every several disentanglement epochs (when the representation distribution has evolved), rather than at every epoch, and each update requires only a few computation steps. We run this computation using GPU vectorization operations, so CoDID is **not significantly CPU-bound**.
> ### W2. Clustering impact on convergence
> Convergence is analyzed in ***Figure 6, Appendix A.2, page 12***. Figure 6(b) shows that abrupt changes appear only when clustering is initialized; afterwards, the losses converge stably. When the number of clusters changes, we only reconfigure layers that take cluster labels as inputs or output cluster-related probabilities, while keeping the rest of the network fixed (***left line 197, page 4***). This avoids abrupt changes and ensures training stability. Thus, clustering would not affect convergence in practice.
> ### W3. Continuous setting
> Thank you for this interesting point. In practice, discretization is a common way to handle continuous variables, so our mode formulation may still be useful to some extent: correlations between continuous variables can be viewed as correlations between discretized value ranges, and modeling these ranges as modes may preserve meaningful range-level information. Fully continuous extensions are beyond the scope of this work.
> ### W4. Larger backbones beyond ResNet18
> We additionally test a larger encoder (**ResNet50**) on Canine-BG. The full results are in ***Table 3 at the link mentioned above***. We observe that:
> * The improvement of **BASE** suggests that a stronger encoder can better capture image structures.
> * **Methods that do not address hidden correlations do not improve consistently**, likely because a larger encoder may also encode more correlation-induced shared information, thus the representation independence constraints of these methods may remove more mode-related shared information.
> * **CoDID improves and remains the best-performing method**, suggesting that a stronger encoder helps CoDID better capture mode structure and thus better handle hidden correlations.
>
> ||Test 3 Acc (ResNet50)|Test 3 Acc (ResNet18)|
> |-|-|-|
> |BASE|55.6|62.5|
> |MMD|54.1|54.9|
> |DTS|41.8|41.8|
> |IDE-VC|49.1|41.9|
> |MI|52.3|46.0|
> |A-CMI|53.0|50.6|
> |HFS|46.8|44.5|
> |CoDID|68.3|79.6|
>
> ### W5. Additional real-world image data
> We appreciate your recognition that real-world time-series datasets provide important evidence beyond synthetic settings. We consider real-world image data as a meaningful step for future work.
> ### W6. Clarity
> * **How meta-learned weights align marginal with joint**: This is achieved by the nested meta-optimization with $L_{meta}$ in ***Equations 7-8***.
>   * In the inner loop, the weight net outputs weights, and these weights are used in weighted CMI minimization; as shown in ***Equation 7***, the optimization step of the encoder is computed while **saving the computation graph**, so the updated encoder parameters remain associated with the weights.
>   * In the outer loop, the meta-loss evaluates whether the **representations from the updated encoder** still preserve attribute/mode information. Therefore, gradients from $L_{meta}$ can flow back through the updated encoder to the weight net.
>   * By ***Proposition 1***, mismatched marginal/joint label distributions harm informativeness, so minimizing the meta-loss automatically encourages weights to reduce this mismatch and achieve alignment.
> * **How weights and subcluster net influence mode finding**: As shown in ***Figure 2, page 4***, the learned weights reflect mode-specific conditional probabilities under within-cluster correlations. The subcluster net first learns to reproduce the DPGMM subclustering assignments, then aligns the output subclustering probabilities with summarized weight patterns to produce more meaningful subclusters. These refined subclustering probabilities are then fed back into DPGMM for split proposals, helping to separate correlated modes into different subclusters.
>
> * **Figure 1/Section 3.1/Algorithm 1**: Thank you for pointing these out. As suggested, we intend to revise the figure and add concrete examples to improve clarity, and add the optimization of $L_{sc}$ at line 16 of Algorithm 1, while mentioning the subclustering output to DPGMM.
>
> We are sincerely grateful for your thoughtful comments and openness to further increasing your score. We are committed to enhancing clarity and including the additional experiments in our paper.

---

> > ### Author Rebuttal · Reviewer_pmwH · 2026-04-03
> >
> > I appreciate the authors addressing my concerns and questions in the rebuttal. All of my concerns have been addressed, so I will raise my score to a 6.

---

> > > ### Author Response · Authors · 2026-04-04
> > >
> > > Dear Reviewer pmwH,
> > >
> > > Thank you for raising your score to **6**! We truly appreciate your time and substantial effort in reviewing our paper and reading our rebuttal.
> > >
> > > We also deeply value your feedback. Your comments are constructive, helpful, and also inspired meaningful discussions and guided us toward improving the work in several key aspects:
> > >
> > > - **Computational cost breakdown:** This has helped us identify the dominant sources of computation costs and validate our design choices regarding acceleration and parameter sharing.
> > > - **Continuous setting:** We find this direction particularly inspiring. We believe it is highly relevant for practical applications and could be a promising topic worth exploring in future work.
> > > - **Larger backbone experiments:** This suggestion has guided us to evaluate our models from the new perspective of model capacities, and we obtained meaningful findings.
> > > - **Other notes and questions:** Other thoughtful comments have helped us improve the clarity and presentation of the paper.
> > >
> > > We sincerely appreciate your careful evaluation, and we are committed to incorporating these points to further strengthen the manuscript.
> > >
> > > Best regards,
> > >
> > > The Authors

---

### Official Review · Reviewer_yMgg · 2026-03-12

**Soundness:** 3
**Presentation:** 3
**Significance:** 3
**Originality:** 3
**Overall Recommendation:** 5
**Confidence:** 4

**Summary:**

The paper tackles the underexplored issue of hidden correlations in supervised Disentangled Representation Learning. The authors propose CoDID, an end-to-end framework that couples iterative mode discovery with conditional independence enforcement. To prevent the cycle of error amplification between clustering and representation learning, the framework introduces a meta-coordination mechanism that reweights marginal representations and refines subclusters.

**Compliance With Llm Reviewing Policy:**

Affirmed.

**Final Justification:**

The rebuttal strengthened the paper by adding comparisons with standard OOD and invariant learning baselines, clarifying its relationship to DEAR, CausalVAE, and GEM, and providing additional evidence that the discovered modes on real data can correspond to meaningful underlying structures. Overall, I find the work original, empirically strong, and likely to be useful to researchers studying disentanglement under hidden correlations and robustness under distribution shift.

**Key Questions For Authors:**

* Can you provide formal convergence guarantees or empirical bounds for the alternating optimization between the non-parametric clustering and the representation learning modules?
* How does CoDID perform against standard out-of-distribution (OOD) or invariant learning baselines (rather than strictly DRL baselines) on the evaluated datasets?
* How sensitive is the meta-coordination mechanism to a collapsed or highly entangled state during the initial pre-training phase?
* Recent works have explored modeling factor correlations using causal graph models or GNNs (e.g., DEAR[1], CausalVAE[2], and GEM [3]). How does the proposed CoDID framework fundamentally differ from these structure-driven approaches in terms of assumptions, problem setting, and methodology?
* Methods like GEM and DEAR typically report quantitative generative metrics, such as FID, to evaluate the quality of disentanglement in the image space. Given that CoDID also employs a decoder module in its architecture, could you provide similar metrics or qualitative image generation/manipulation results to facilitate a more comprehensive comparison?


[1] Shen, X., Liu, F., Dong, H., Lian, Q., Chen, Z., & Zhang, T. (2022). Weakly supervised disentangled generative causal representation learning. Journal of Machine Learning Research, 23(241), 1-55.

[2] Yang, M., Liu, F., Chen, Z., Shen, X., Hao, J., & Wang, J. (2021). Causalvae: Disentangled representation learning via neural structural causal models. In Proceedings of the IEEE/CVF conference on computer vision and pattern recognition (pp. 9593-9602).

[3] Xie, B., Chen, Q., Wang, Y., Zhang, Z., Jin, X., & Zeng, W. (2024). Graph-based unsupervised disentangled representation learning via multimodal large language models. Advances in Neural Information Processing Systems, 37, 103101-103130.

**Limitations:**

yes

**Strengths And Weaknesses:**

**Strengths**

* The paper provides solid theoretical analysis to back its architectural choices. It rigorously proves that attribute-based conditional mutual information minimization fails under hidden correlations and demonstrates how within-cluster correlations degrade representation informativeness.

* The meta-coordination mechanism is a clever and theoretically sound solution. Using a meta-optimized weight network to handle within-cluster correlations and translating those learned weight patterns to guide DPGMM cluster splits effectively closes the loop between clustering and representation learning.

* The empirical validation is thorough. The method demonstrates a significant performance margin over strong baselines, and extensive ablation studies carefully dissect the contributions of each module.

**Weaknesses**

* The propositions serve mainly as motivational justifications, lacking rigorous mathematical guarantees for alternating optimization convergence, error amplification bounds, and training stability during dynamic split/merge operations.
* Despite targeting robust out-of-distribution prediction, evaluations strictly compare against disentanglement methods, omitting general robustness baselines like invariant learning. Furthermore, hidden modes in real-world datasets are explicitly assumed, lacking direct evidence that they correspond to genuinely meaningful physical or semantic sub-distributions.
* The tightly coupled multi-network architecture raises the barrier for reproducibility and incurs significant computational overhead. Crucially, the closed-loop system is highly sensitive to initialization; if early pre-trained representations are entangled, the iterative clustering fails at the outset, destabilizing the entire meta-coordination mechanism.

---

> ### Author Rebuttal · Authors · 2026-03-31
>
> Thank you very much for your thoughtful review and positive assessment. We address your comments below.
>
> ### W2&Q2: Additional OOD/invariant learning baselines
> We additionally compare with **DANN, CORAL, IRM, and REx**, along with **BASE (ERM)** and **MMD** from our paper. The full results are in ***Table 6 at https://anonymous.4open.science/api/repo/CoDID-B038/file/supplementary.pdf?v=baeedeaa***. The results show that **CoDID** performs the best due to the explicitly modeling of underlying modes and hidden correlations:
> * Under strong correlation shifts on CMNIST, CFashion-MNIST, and Canine-BG, invariant representation learning methods **DANN, CORAL, and MMD** perform poorly, due to not accounting for correlations and enforcing representation invariance, which may harm representation informativeness.
> * Invariant prediction methods **IRM and REx** do not enforce independence between representations, and thus maintain a modest advantage over BASE (ERM) in some cases. However, they underperform CoDID due to lack of explicit mode modeling.
>
> |Method|CMNIST|CFashion|Canine-BG|UCI-HAR|Realworld|HHAR|MFD|
> |-|-|-|-|-|-|-|-|
> |BASE(ERM)|79.2|77.8|55.6|71.2|64.6|80.8|72.2|
> |MMD|57.3|63.5|54.1|70.3|66.0|80.9|78.2|
> |DANN|58.3|63.8|29.8|67.8|66.0|77.1|74.0|
> |CORAL|58.6|68.0|51.0|74.4|64.8|81.0|77.3|
> |IRM|81.0|79.3|54.3|70.9|65.4|82.5|78.4|
> |REx|81.7|76.3|56.0|73.7|65.1|80.6|78.8|
>
> ### W2: Evidence for meaningful underlying modes on real data
> We visualize the discovered modes on real data in ***Figure 7 and 9, Appendix A.5, page 14-15***. Figure 7 shows representation distributions with inter-mode separation, intra-mode compactness, and between-class distinctiveness in machine-fault data. Figure 9 shows raw time series signals, where the discovered walking modes differ in mean values and volatility, plausibly corresponding to different paces/strides/postures. This supports that the discovered modes can reflect meaningful underlying structures on real data.
>
> ### W3&Q3: Implementations
> * **Reproducibility**: Our codes are provided in the anonymous Github link in the abstract and will be made publicly available to support reproducibility.
> * **Computational overload**: As analyzed in ***Appendix A.7, page 16***, the additional cost over existing mutual information (MI) minimization methods is moderate: the extra modules do not introduce substantial parameter overload, and the training time remains comparable to existing MI-based methods. These modules are necessary because they explicitly model the underlying modes and hidden correlations ignored by the baselines, and their benefits are empirically verified.
> * **Sensitivity to entangled early pre-training**: ***Figure 8(c), Appendix A.5, page 14*** shows CoDID can gradually recover the true mode structures from highly entangled initial representations. Entanglement means the over-encoding of other attributes, most likely those correlated with the target attribute and its modes. This leads to clustering errors and within-cluster correlations, which is the key problem that our meta-coordination mechanism addresses (***Appendix C***).
>
> ### Q4: Comparison to DEAR, CausalVAE, and GEM
> * **Assumptions**: These methods and CoDID are based on Structural Causal Model (***Appendix B.2, page 18***), but the assumptions differ: DEAR/CausalVAE/GEM allow causal effects between attributes/factors, whereas CoDID follows [1] and treats attributes as elementary ingredients without direct causal effects, while introducing confounders to model correlations, and mode variable to model underlying structures.
> * **Problem setting**: DEAR and CausalVAE are generative causal disentanglement methods requiring **additional supervision** and aiming at controllable generation. GEM is an **unsupervised** structural DRL framework for discovering factor interrelations. CoDID addresses supervised DRL for **robust attribute prediction**, where attribute labels are known but **mode labels are unknown**.
> * **Methodology**: These methods encode inter-factor structure in the latent space through a graph prior and optimize a generative objective. CoDID instead uses iterative mode discovery and disentanglement: nonparametric clustering infers modes, **clustering-based CMI minimization** performs disentanglement, and **meta-coordination** prevents error amplification. Underlying mode structures are not explicitly modeled by these methods.
>
> [1] Robustly disentangled causal mechanisms: Validating deep representations for interventional robustness, ICML 2019.
>
> ### W1&Q1&Q5: Theoretical bounds and generative evaluation
> * Formal proofs are not included due to space constraints, but **can be provided as a follow-up after reviewer response**. Empirical convergence evidence is in ***Figure 6, page 12, Figure 8, page 14, and Figure 15, page 24 in Appendix A, C***.
> * For generative evaluation, please see ***Figure 2 at the link mentioned above***.
>
> We hope these clarifications and additional results address your concerns.

---

> > ### Author Rebuttal · Reviewer_yMgg · 2026-04-02
> >
> > Thank you for the detailed rebuttal. The additional comparisons with DANN, CORAL, IRM, and REx are useful and directly address one of my earlier concerns regarding the lack of standard OOD/invariant learning baselines. The clarification on the relationship to DEAR, CausalVAE, and GEM is also helpful, and the supplementary visualizations of discovered modes on real data make the empirical story more convincing. I will raise my score, as the paper offers strong motivating propositions and useful empirical evidence for stable optimization behavior. That said, some important issues remain unresolved, most notably the lack of a formal convergence guarantee and the absence of explicit bounds on error amplification for the alternating optimization procedure with dynamic split/merge operations. I hope the authors will investigate these questions more thoroughly in future work, as doing so would further strengthen the theoretical foundation of the proposed framework.

---

> > > ### Author Response · Authors · 2026-04-05
> > >
> > > Dear Reviewer yMgg,
> > >
> > > Thank you for raising your score! We truly appreciate your time and substantial effort in reviewing our paper and reading our rebuttal. Your feedback is invaluable to us.
> > >
> > > Thank you for your early reply and for your patience. We agree that theoretical guarantees would strengthen our paper. After careful considerations over the past few days, we are now able to provide the following results about the **components of CoDID** with reasonable assumptions. Convergence of the whole alternating optimization would require additional assumptions, which we leave as future work.
> > > ### 1. Only ideal weights (***right line 157, page 3***) may allow intact representation informativeness
> > > **Notations**: In cluster $k$, denote the normalized weighted marginal label probability function as $q_k^w(\mu,\alpha):=C_k w(\mu,\alpha,k)p(m_1=\mu \mid c_1=k)p(a_2=\alpha \mid c_1=k)$, where $C_k$ is a normalization constant, and denote the weighted marginal representation distribution as:
> > > $$\tilde p_k^w(z_1,z_2):=\sum_{\mu',\alpha'}q_k^w(\mu',\alpha')p(z_1 \mid m_1=\mu', c_1=k)p(z_2 \mid a_2=\alpha', c_1=k)$$
> > > **Proposition R1**: For $c_1 = k$, if the weighted marginal label probability function does not match the joint label probability function, i.e., $q_k^w(\mu,\alpha)\neq p(m_1=\mu,a_2=\alpha \mid c_1=k)$,
> > > and exact weighted alignment is enforced on the learned representations, i.e., $\tilde p_k^w(z_1,z_2)=p(z_1,z_2 \mid c_1=k)$, then at least one of $I(z_1;m_1 \mid c_1=k)<H(m_1 \mid c_1=k)$ and $I(z_2;a_2 \mid c_1=k) < H(a_2 \mid c_1=k)$ must hold, i.e., informativeness is compromised.
> > > ### 2. Meta-optimization converges
> > > We use **one-step meta-optimization with second-order gradients** to learn the weight net parameters $\phi_{w}$. By ***Theorem 4.12 [1]***, suppose the standard smoothness, bounded-gradient-variance, and step-size assumptions are satisfied, the meta-optimization **converges to an $\varepsilon$-first-order stationary point (FOSP)** of the objective $\mathcal{L}\_{\rm{meta}}(\phi_w)$ (i.e., $\|\nabla \mathcal{L}_{\rm{meta}}(\phi_w)\|\le \varepsilon$) after at most $O(1/\varepsilon^2)$ iterations.
> > >
> > > [1] On the Convergence Theory of Gradient-Based Model-Agnostic Meta-Learning Algorithms. AISTATS 2020.
> > >
> > > Based on **proposition R1** above, it is safe to conclude that **non-ideal weights** harm informativeness, assume $\mathcal{L}_{\rm{meta}}$ is an appropriate substitution for mode prediction loss (with similar minimizer and gradients), these non-ideal weights **cannot be the global minimizer of $\mathcal{L}_{\rm{meta}}$**. This would likely **eliminate some ill-posed weight functions**.
> > > ### 3. DPGMM with detailed balance
> > > We use Dirichlet Process Gaussian Mixture Model (DPGMM) to discover modes, where the split-merge Markov chain Monte Carlo (MCMC) procedure follows the Metropolis-Hastings (MH) framework [3]. In this procedure, split/merge are proposed as transitions of the chain, and Hastings ratio is calculated as the **acceptance probability** of the proposed transition. For a correctly specified MH step, the induced transition satisfies **detailed balance**, i.e., reversible w.r.t. the posterior over the chain’s state (cluster assignments), which implies that the posterior is **stationary** [2]. Although formal **convergence rates** are typically not established, appropriate parameter tuning can usually lead to convergence [3]. Stationarity is a desirable theoretical property supporting this behavior.
> > >
> > > [2] K. W Hastings. Monte Carlo sampling methods using Markov chains and their applications. 1970.
> > >
> > > [3] A split-merge Markov chain Monte Carlo procedure for the Dirichlet process mixture
> > > model. 2004.
> > > ### 4. Corrected subclustering
> > > We prove that the subclustering probablities learned by alignment loss $\mathcal{L}\_{\mathrm{wa}}$ (***Equation 10, page 6***) can distinguish different modes, assuming **close-to-ideal** weights, **monotone** weight summaries $\mathcal{P}^{\mathrm{w}}$ and subclustering probabilities $\hat{r}^{\rm{sc}}=(\hat{r}^{\rm{sc}}\_1, \hat{r}^{\rm{sc}}\_2), \hat{r}^{\rm{sc}}\_1+\hat{r}^{\rm{sc}}\_2=1$ with **regularizations** (monotone encouraged by $\mathcal{L}_{\rm{wa}}$).
> > >
> > > **Proposition R2**: In cluster $k$, for representations $z_i, z_j$ from modes $\mu,\nu (\mu \neq \nu)$ under within-cluster correlations, respectively, denote $D\_{\rm{w}}(i,j)=\bar{h}\_{\rm{jsd}}(\mathcal{P}^{\rm{w}}\_i,\mathcal{P}^{\rm{w}}\_j)$. For certain $\kappa_{\mu\nu}>0$, $p_{\min}\in(0,\tfrac12)$, non-decreasing function $\omega(\cdot)$, if $D_{\rm{w}}(i,j) \ge \kappa_{\mu\nu}$, then $\bigl|\hat{r}^{sc}\_{i,1}-\hat{r}^{sc}\_{j,1}\bigr|\ge \sqrt{2}p\_{\min}(1-p\_{\min})\omega(\kappa\_{\mu\nu})$.
> > >
> > > This likely assigns different modes to different subclusters and reduces clustering error.
> > >
> > > Thank you for inspiring this meaningful discussion. We are committed to incorporating the detailed assumptions and proofs in the paper. We have benefited a great deal from your valuable insights.
> > >
> > > Sincerely,
> > >
> > > The Authors

---

### Official Review · Reviewer_cMGy · 2026-03-13

**Soundness:** 3
**Presentation:** 3
**Significance:** 3
**Originality:** 3
**Overall Recommendation:** 5
**Confidence:** 3

**Summary:**

This paper identifies the degradation of disentanglement and information loss caused by hidden correlations in supervised disentangled representation learning. Hidden correlations can arise when an attribute exhibits multi-mode structure with different correlations across modes (i.e., when attribute labels are not sufficiently detailed). To address this, the paper proposes using clustering on representations as conditioning (psuedo) labels instead of simply using supervised attribute labels, thereby handling multiple mode structure. While a naive application may not work well due to the negative synergy between inaccurate clustering-based labels and representation learning based on them, a meta-coordination mechanism is introduced to address this issue. The proposed method was evaluated on seven real-world datasets and achieved robust attribute prediction performance outperforming existing methods.

**Compliance With Llm Reviewing Policy:**

Affirmed.

**Final Justification:**

Through the Rebuttal and Reply Rebuttal Comment, the authors have mostly addressed my concerns.
While the proposed method may seem complicated in some part, it does not seem to be overly complex or incur excessive costs compared to other methods, and the extension to multi-attribute settings appears feasible.
The paper addresses what seems to be an important and novel issue, and the proposed approach seems reasonable to me.
Thus, I raise my recommended score to 5.

**Key Questions For Authors:**

- Address the weaknesses above.

**Limitations:**

yes.

**Strengths And Weaknesses:**

### Strengths
 - The hidden correlation problem in supervised disentangled representation learning identified in this paper seems to be an important and novel issue.
- The proposed clustering-based approach seems reasonable to me (in the absence of detailed labels), and the meta-coordination mechanism that controls the potential error amplification seems convincing.
- The method achieves strong performance with large margins over baselines across diverse datasets.
- Detailed ablation studies and analyses are provided.

### Weaknesses
- As far as i understand, the paper focuses on the two-attribute setting in both its formulation and experiments. Given that the proposed method is highly dependent on clustering quality, it is questionable whether it would work well as the number of attributes increases. Experimental validation of scalability in multi-attribute (>2) settings seems necessary.
- Meta-learning, iterative clustering, and other components are expected to incur significant training costs. Additionally, the complex architecture may pose optimization difficulties.

---

> ### Author Rebuttal · Authors · 2026-03-31
>
> Thank you for your constructive comments and favorable recognition. We address each point below.
>
> ### 1. Extension to multi-attribute settings
> Thank you for noting the scalability to multiple attributes. CoDID can be extended to multiple attributes $a_1, ..., a_k, ..., a_K$ as follows:
> * **Theoretical extension**: This is achieved by casting the disentanglement of **$a_1$ from a single $a_2$** onto the disentanglement of **$a_k$ from the joint $a_{-k} = \{a_j\}_{j \neq k}$**. Following [1], the extension mainly involves replacing corresponding variables, i.e., replacing $a_1, m_1, c_1, z_1$ with $a_k, m_k, c_k, z_k$, and replacing $a_2, z_2$ with $a_{-k}, z_{-k}$, as the **properties of mutual information and causal graphs remain the same for joint variables**.
>   * *Proposition 1 (extended).* If $I(m_k; a_{-k}|c_k) > 0$, then enforcing $I(z_k; z_{-k}|c_k) = 0$ leads to at least one of $I(z_k; m_k) < H(m_k)$ or $I(z_{-k}; a_{-k}) < H(a_{-k})$.
>   * *Proposition 2 (extended).* If $I(m_k; a_{-k}|a_k)>0$, then enforcing $I(z_k; z_{-k}|a_k)=0$ leads to at least one of $I(z_k;m_k) < H(m_k)$ or $I(z_{-k};a_{-k})<H(a_{-k})$.
>   * The extended causal graphs are in the ***Figure 1 at https://anonymous.4open.science/api/repo/CoDID-B038/file/supplementary.pdf?v=baeedeaa***.
> * **Methodological extension**:
>   * **Supervised learning and reconstruction**: build one predictor per attribute and reconstruct data from all attribute representations.
>   * **For disentangling attribute $a_k$ with underlying modes**: use $z_k$ for mode discovery; enforce clustering-based conditional independence between $z_k$ and the joint $z_{-k}$ with a discriminator [1]; in meta-coordination, weight net receives the concatenated one-hot labels and prediction losses of each attribute in $a_{-k}$, and subcluster net uses weight summaries w.r.t. the joint $a_{-k}$, with the weight distribution fitted under each unique attribute value combination of $\{a_j\}_{j \neq k}$.
>   * **For disentangling attribute $a_j, j\neq k$ without underlying modes**: enforce attribute-based conditional independence between $z_j$ and the joint $z_{-j}$ with a discriminator [1].
>
> [1] Disentanglement and generalization under correlation shifts, CoLLAs, 2022.
>
> * **Experimental extension**: We add multi-attribute experiments on toy data, the results in ***Table 5 at the link mentioned above*** show that CoDID outperforms baselines by discovering and leveraging modes for multi-modal attributes $a_1,a_2$. Due to the meta-learned weights, clustering errors would not harm representation informativeness, and multi-attribute joint disentanglement can be gradually achieved by removing redundant information while preserving mode information (***Figure 8, page 14, Appendix A.5 & Figure 15, page 24, Appendix C.2.2***). This in turn helps disentangle single-modal attributes $a_3,a_4$, which are correlated with $a_1,a_2$ and their modes.
> ### W2. Training cost
> **Computational cost** is analyzed in ***Appendix A.7, page 16*** with component breakdown in ***Table 1&2 at the link mentioned above***, showing:
> * **Moderate parameter increase over baselines**: This is due to the lightweight structure of additional components:
>   * Weight net and subcluster net are small multi-layer perceptrons.
>   * DPGMM is a statistical method without additional neural backbones.
>   * Conditional discriminator uses parameter sharing across modes under the same attribute value.
> * **Comparable training-time versus other mutual information (MI) based methods**: The training duration is mainly due to the adversarial training for minimizing conditional mutual information (CMI). CoDID stays in a comparable range to other MI-based approaches. This is due to our acceleration choices, including partially shared discriminators, vectorized mode parallelization, and stabilized training dynamics as stated below, which leads to fast convergence. Meta-learning only takes up 15\% of the training time.
> ### W2. Optimization
> We adopt customized designs to ensure stable optimization:
> * **Meta-coordination to avoid error amplification**: This mechanism uses meta-learning to ensure the mutual enhancement of mode discovery and disentanglement modules, thus facilitating stable optimization. Weight net preserves representation informativeness against clustering errors during disentanglement, while subcluster net refines DPGMM subclusterings using weight summaries, guiding cluster splits to ensure clustering is correctly updated. The effectiveness of such meta-learned weights is validated in ***Figure 4, page 8***.
> * **Targeted reconfiguration for clustering update**: When the number of clusters changes, only cluster-related input/output layers are updated, while the rest of the network stays fixed, which avoids abrupt changes and ensures training stability. Details are in ***left line 196, page 4*** and ***Appendix D.4***.
>
> We hope our responses address your concerns. We are committed to including the additional analysis in our paper.

---

> > ### Author Rebuttal · Reviewer_cMGy · 2026-04-03
> >
> > We thank the authors for their detailed rebuttal, which addressed most of my concerns. While the multi-attribute experiments are limited to toy data and real-world validation would have been more convincing, I understand this is reasonable given the short rebuttal period.
> >
> > One additional question: the computational cost analysis appears to be based on the two-attribute setting. Do the authors expect the computational cost to remain comparable to MI-based baselines in multi-attribute settings as well?
> >
> > ---
> > Thank you for the Reply Rebuttal Comment. It helps my understanding. I have reflected the Rebuttal and Reply Rebuttal Comment in the updated Overall Recommendation and Final Justification.

---

> > > ### Author Response · Authors · 2026-04-04
> > >
> > > Dear Reviewer cMGy,
> > >
> > > Thank you for thoughtfully reading our rebuttal and providing valuable feedback. Also, thank you for recognizing the additional multi-attribute toy experiments and for understanding our reasons for this setting. We consider real-world multi-attribute experiments as a meaningful extension for future work.
> > >
> > > We sincerely appreciate the opportunity to provide further clarifications. To answer the additional question, we do expect the computational cost to **remain comparable** to MI-based baselines in multi-attribute settings. We provide clarifications w.r.t. different network components and computational costs:
> > >
> > > * **For disentangling a certain attribute $a_k$ with underlying modes**: For this attribute, the extension of the MI objective is similar for CoDID and MI-based baselines, replacing the **independence constraint with $a_2$** to **independence constraint with the joint $a_{-k} = (a_j)_{j \neq k}$** [1], as summarized in the table below.
> > >   * All methods require expansions of the **discriminator** architecture to receive the joint $z_{-k}$ as inputs, with **proportional** parameter/computation costs.
> > >   * All methods require additional attribute **predictors** for each attribute in $a_{-k}$, with **the same** parameter/computation costs.
> > >   * The **clustering module and subclustering net** of CoDID remain **unchanged**, as they only operate on representation $z_k$.
> > >   * The **weight net** of CoDID requires architecture expansion to receive the concatenated one-hot labels and prediction losses of the joint $a_{-k}$. This induces **negligible parameter increase and computational cost**, as the weight net is a small MLP (only **0.07\%** of the total parameters) that takes up a small portion of the training time (**15.63\%** of the total training time), as shown in ***Table 1&2 in the PDF at https://anonymous.4open.science/api/repo/CoDID-B038/file/supplementary.pdf?v=baeedeaa***.
> > >
> > > |Method|Two-Attribute Objective|Multi-Attribute Objective|Handled Correlations|
> > > |:-|:-|:-|:-|
> > > |CoDID|$\min I(z_1;z_2\mid m_1)$|$\min I(z_k;z_{-k}\mid m_k)$|Hidden correlations|
> > > |A-CMI [1]|$\min I(z_1;z_2\mid a_1)$|$\min I(z_k;z_{-k}\mid a_k)$|Attribute correlations|
> > > |Conventional|$\min I(z_1;z_2)$|$\min I(z_k;z_{-k})$|None|
> > >
> > > [1] Disentanglement and generalization under correlation shifts, CoLLAs, 2022.
> > >
> > > * **For disentangling other attributes $a_j, j\neq k$**:
> > > One MI objective is required for disentangling the representation $z_j$ of this attribute from the joint representations of other attributes (similar to $z_k$ in the table above).
> > >   * **Training time**: Since the two-attribute case shows comparable training time between CoDID and MI-based baselines, which is **dominated** by adversarial MI minimization (51.35% of the total training time, as shown in ***Table 1 at the link above***), the extended training time would be proportional and thus **comparable** in the multi-attribute case.
> > >   * **Parameter count**: One **discriminator** would be constructed for the corresponding MI objective, with **proportional** parameter increase.
> > >
> > > We hope this clarification helps.
> > >
> > > Thank you for raising the constructive points regarding the multi-attribute case and computational cost. This has helped us strengthen the practical usage and scalability aspects of our paper. We are fully committed to refining our paper to incorporate these points.
> > >
> > > Best regards,
> > >
> > > The Authors

---

### Official Review · Reviewer_RWbM · 2026-03-20

**Soundness:** 2
**Presentation:** 2
**Significance:** 2
**Originality:** 2
**Overall Recommendation:** 3
**Confidence:** 3

**Summary:**

The submission studies supervised disentangled representation learning in the presence of attribute correlations and additional latent variation within attribute values.  The motivating example considers a binary activity label (walking vs. running), where the "walking" class contains unobserved sub-modes (e.g., slow vs. fast walking) that may correlate with other attributes such as user identity.  The goal is to learn representations that both disentangle labeled attributes and preserve such latent structure.
The proposed method alternates between clustering representations to infer latent modes and minimizing conditional mutual information between representation components conditioned on these clusters.  A meta-coordination mechanism is introduced to stabilize this iterative process. Experiments on synthetic and real datasets demonstrate improvements over prior disentanglement baselines.

**Compliance With Llm Reviewing Policy:**

Affirmed.

**Final Justification:**

The rebuttal and the other reviewers' comments helped me to better understand the method.  I still feel this is an overcomplicated solution to a structured problem, but I can see its merits.

**Key Questions For Authors:**

In the CMNIST setting, where digit identity serves as the latent "mode" within parity, how is variation such as handwriting style distributed across the learned representations?  More generally, when multiple sources of intra-class variation exist, what determines which attribute representation $z_q$ they are assigned to?

**Limitations:**

yes

**Strengths And Weaknesses:**

# Strengths
- The iterative combination of clustering with conditional mutual information minimization is a reasonable and potentially useful approach.

# Weaknesses

- The formulation implicitly assumes that each labeled attribute corresponds to an incomplete factor that should be augmented with latent "modes".  This departs from standard supervised disentanglement, where labels define the factors of interest, and instead introduces an additional latent variable whose association with a specific attribute is not justified.  In particular, since the "hidden correlations" arise from relationships between this latent variation and other attributes, it is unclear why the method assigns this variation exclusively to $z_q$​, rather than modeling it as a separate factor.  This makes the objective internally ambiguous.

- The method introduces substantial complexity (multiple interacting losses, clustering, and meta-optimization) for a setting that depends on strong structural assumptions (discrete clusterable modes within attributes).  It is unclear how broadly applicable this setting is in practice.

---

> ### Author Rebuttal · Authors · 2026-03-31
>
> Thank you very much for your thoughtful comments on our formulation, assumptions, and scope. We appreciate the opportunity to clarify these points.
>
> ### Q1&W1. Formulation of latent modes
> Thank you for raising this point. We clarify the formulation of modes as follows:
> * **Flexibility of the assumption**: Our assumption is that **certain attributes** may contain latent modes under **some values**. This is not imposed on every attribute or every value.
> * **Basic disentanglement assumption**: Disentangled representation learning typically treats labeled attributes as **elementary ingredients** in data generation, where each attribute captures one aspect of the data and can be **changed (by intervention) without affecting the others**, although confounding may exist[1]. For example, in human activity data, activity and user identity may be correlated because some users perform certain activities more often, but user identity does not directly determine which activity is performed, nor vice versa.
> * **Association of modes with a specific attribute**: For certain attributes, one attribute value may contain meaningful latent variations **intrinsic to that attribute**. For example, a simple activity **"walking"** may include modes such as **"stroll" and "brisk walk"** (***Figure 1, page 1***); a complex activity **"doing chores"** may include modes such as **"ironing clothes"** and **"sweeping"**.
>   * Modes ($m_1$) reflect the activity semantics and depend on the activity $a_1$, e.g., the mode "brisk walk" can only occur under activity "walking". Modes are not causally affected by user identity $a_2$; although they may be statistically correlated, e.g., a user tends to engage in some activity modes more frequently.
>   * **Difference with general intra-class variations**: In one activity class, variations intrinsic to activity semantics are treated as its modes, while variations induced by personal user patterns are treated as redundant information about the other attribute, which should be removed during disentanglement.
> * **Generative modeling**: We model the attribute and its modes through the path $a_1 \to m_1 \to z_1^l \to x$ in the data generation process (***Appendix B.2, Figure 13(b), page 19***). Intuitively, for human activity data, $a_1$ is the activity label (e.g., "walking"), $m_1$ is the activity mode label (e.g., the walking mode "stroll"), and $z_1^l$ captures the specific body-movement patterns characterizing that mode (e.g., the pace, stride, posture that identifies "stroll"). Therefore, $m_1$ is associated with $a_1$ rather than modeled as a separate factor.
>
> We agree that these aspects should be explained more clearly in the main text, and we will revise the paper accordingly.
>
> [1] Robustly disentangled causal mechanisms: Validating deep representations for interventional robustness, ICML, 2019.
>
> ### W2. Applicability
> Thank you for noting applicability. This is addressed from three aspects:
> * **Complexity and deployment**: As analyzed in ***Appendix A.7, page 16***, the additional cost over existing mutual information (MI) minimization methods is moderate: the extra modules do not introduce substantial parameter overload, and training time remains comparable to existing MI-based methods. We also provide a component-wise cost breakdown in ***Table 1&2 at https://anonymous.4open.science/api/repo/CoDID-B038/file/supplementary.pdf?v=baeedeaa***. In practice, the model can be **trained offline** with local computational resources. For online deployment, only the learned encoder and predictor are needed for inference.
> * **Validated real-world usage**: Our multi-modal assumption is validated on real data rather than only synthetic settings. We use **four real-world time-series datasets**: **UCI-HAR, RealWorld, HHAR**, where human activities may naturally contain latent modes, and **MFD**, where a machine fault type may include latent modes with different forms of damage/pitting/indentation. See ***Figure 7&9, Appendix A.5, page 14-15*** for visualization of discovered modes on real data.
> * **Adaptability**: In practice, one can **choose** to model latent mode structures only for target task-relevant attributes with inherent complexity. We further use **non-parametric clustering** to automatically estimate the number of modes under each attribute value, allowing **different numbers of modes under different attribute values**, including the single-modal case.
>
> ### Q1. CMNIST setting
>
> In CMNIST, we aim to discover digit under parity, and disentangle the digit representations from color. Variations such as handwriting style are not explicitly modeled, so they may be reflected in the learned representations as they appear in the data. If they were to be modeled, they would be treated as another attribute to disentangle from digit representations, as they are not caused by the digit semantics.
>
> We hope these responses address your concerns. We are committed to improving clarity in our paper as suggested.

---

> > ### Author Rebuttal · Reviewer_RWbM · 2026-04-03
> >
> > Thank you for the clarifications -- I have a better understanding of the formulation now.

---

> > > ### Author Response · Authors · 2026-04-06
> > >
> > > Dear Reviewer RWbM,
> > >
> > > Thank you for raising your score and for confirming your understanding of our formulation! We truly appreciate your time and effort in reading our rebuttal and reviewing our paper. Your feedback is invaluable to us.
> > >
> > > We also thank you for your patience. During the last few days, we have carefully reflected on the presentation w.r.t. the mode formulation. We acknowledge the need to give this more room in the main paper rather than Appendix, as "latent modes" is indeed a novel problem setting (**as you insightfully noted**). We hope to take this opportunity to **supplement additional points** and list **the revisions for addressing your concerns**.
> > > ### 1. Supplementing "formulation of latent modes"
> > > We plan to add the following content to expand ***Section 3***:
> > > * A formal structural causal model (SCM) about the **data generation process**, as given in **Definition R1** below. The causal graph is in ***Figure 1 at https://anonymous.4open.science/api/repo/CoDID-B038/file/supplementary_reply.pdf?v=9754bce4*** (corresponding to ***Figure 13, page 19***).
> > > * The **elementary ingredient assumption** mentioned in the rebuttal, which applies to the Definition R1 below.
> > >
> > > **Definition R1**. (Disentangled Causal Process). Consider a causal generative model $p(x|a)$ for data $x$ with attributes $a=(a_1, a_2)$. Attribute $a_1$ is associated with a categorical mode variable $m_1$. Attributes $a$ are influenced by $L$ confounders $c^{\rm{a}}=(c^{\rm{a}}_1, ..., c^{\rm{a}}_L)$. Conditioned on $a_1$, mode variable $m_1$ and the other attribute $a_2$ are influenced by $Q$ confounders $c^{\rm{m}}=(c^{\rm{m}}_1, ..., c^{\rm{m}}_Q)$. This causal model is called disentangled if and only if it follows a SCM of the form:
> > > $$c^{\rm{a}} \gets n^{\rm{ca}} , c^{\rm{m}} \gets n^{\rm{cm}}$$
> > > $$a_2\gets h^{\rm{a}}_2(c^{\rm{a}},c^{\rm{m}},n^{\rm{a}}_2)$$
> > > $$a_1 \gets h^{\rm{a}}_1(c^{\rm{a}},  n^{\rm{a}}_1)$$
> > > $$m_1 \gets h^{\rm{m}}(a_1, c^{\rm{m}},  n^{\rm{m}})$$
> > > $$x \gets g(a_2,  m_1, n^{\rm{x}})$$
> > > with functions $g$, $h^{\rm{a}}_i$, $h^{\rm{m}}$, jointly independent noises $n^{\rm{ca}}$, $n^{\rm{cm}}$, $n^{\rm{a}}_i$, $n^{\rm{m}}$, $ n^{\rm{x}}$.
> > > ### 2. Supplementing "Applicability"
> > > We believe the following materials supplemented for other reviewers can further support the applicability of CoDID:
> > > * **Extension to multi-attribute setting:** This is elaborated in our **response to Reviewer cMGy**, and relates to **your question about CMNIST**, where handwriting style can be viewed as an additional attribute. This includes:
> > >   * **Theoretical** extension via casting the disentanglement of **$a_1$ from the single $a_2$** onto the disentanglement of **$a_k$ from the joint $a_{-k}=(a_j)_{j \neq k}$**;
> > >   * **Methodological** extension via reconfigurations for the joint attribute $a_{-k}$;
> > >   * **Experimental** extension via multi-attribute toy experiments.
> > >   * **Complexity** analysis showing that comparable training cost to baselines.
> > > * **Theoretical guarantee:** In our **response to Reviewer yMgg**, we added theoretical analysis w.r.t. convergence and errors. The main results include:
> > >   * Ideal weights (***right line 157, page 3***) are the **only solution** that enables full representation informativeness.
> > >   * Meta-optimization **convergence** under standard assumptions.
> > >   * Mode discovery module has **detailed balance & stationarity** property.
> > >   * Refined subclustering assignment can **distinguish different modes**.
> > >
> > > For details, please refer to these rebuttals.
> > > ### 3. Supplementing "Data settings"
> > > **Synthetic data:** These data include constructed modes and introduced hidden correlations (**Figures 14, 16, 17, page 23, 25**).
> > >   * On **CMNIST** (mentioned in your question), Fashion-CMNIST, and the toy data, the modes are constructed to **simulate** the problem formulation, not to exactly match a realistic application.
> > >   * **Canine-BG** is the **closest to a realistic scenario**, where dogs can be divided by *functional category* ($a_1$) into *working dog* and *pet dog*, and the modes correspond to the specific *breeds* under each category, e.g., "standard poodle" under "pet dog". The background environment ($a_2$) includes *indoors* and *outdoors*. Correlations between $a_1$/$m_1$ and $a_2$ mirror natural confounding, e.g., working dogs are more likely to be outdoors, and certain breeds might be outdoors more frequently.
> > >
> > > **Real data:** To validate the **activity mode examples** in the previous rebuttal, we visualize the representation distribution on RealWorld dataset in ***Figure 2 at the link mentioned above***.
> > >
> > > ---
> > > As the mode formulation is established, our proposed framework further addresses error amplifications between the mutually dependent disentanglement and mode discovery modules, modeling complex mode structures under hidden correlations in a coordinated manner.
> > >
> > > Thank you again for your valuable comments. We remain committed to incorporating these points in our paper.
> > >
> > > Sincerely,
> > >
> > > The Authors

---

### Decision · Program_Chairs · 2026-04-30

**Decision:**

Accept (regular)

**Comment:**

This paper proposes a new framework to tackle disentangled representation learning. It aims to additionally capture the correlation between hidden modes of an attribute with other attributes, which is often overlooked by existing techniques. To this end, the framework jointly discovers hidden modes and enforces mode-based conditional independence in learned representation.  Experimental results show the new framework achieves significant performance improvement over several baseline methods.

The work carries some interesting ideas and the performance improvement seems significant. Three reviewers are on the positive side. Authors are encouraged to incorporate reviewers’ feedback in preparing the future version, especially the training cost.